# Characteristics of the 1979-2020 Antarctic firn layer simulated with IMAU-FDM v1.2A

Sanne B.M. Veldhuijsen[1], Willem Jan van de Berg[1], Max Brils[1], Peter Kuipers Munneke[1], and Michiel R. van den Broeke[1]

[1]Institute for Marine and Atmospheric Research Utrecht, Utrecht University, Utrecht, The Netherlands

**Correspondence:** Sanne B.M. Veldhuijsen (s.b.m.veldhuijsen@uu.nl)

**Abstract.** Firn simulations are essential for understanding Antarctic ice sheet mass change as they enable us to convert satellite altimetry observed volume changes to mass changes, column thickness to ice thickness, and to quantify the meltwater buffering capacity of firn. Here, we present and evaluate a simulation of the contemporary Antarctic firn layer using the updated semi-empirical firn model IMAU-FDM for the period 1979-2020. We have improved previous fresh snow density and firn compaction parameterizations, and used updated atmospheric forcing. In addition, the model has been calibrated and evaluated using 112 firn core density observations across the ice sheet. We found that 62 % of the seasonal and 67 % of the decadal surface height variability are due to variations in firn air content rather than firn mass. Comparison of simulated surface elevation change with a previously published multi-mission altimetry product for the period 2003-2015 shows that performance of the updated model has improved, notably in Dronning Maud Land and Wilkes Land. However, a substantial trend difference (>10 cm yr$^{-1}$) remains in the Antarctic Peninsula and Ellsworth Land, mainly caused by uncertainties in the spin-up forcing. By estimating previous climatic conditions from ice core data, these trend differences can be reduced by 38 %.

## 1 Introduction

The Antarctic ice sheet (AIS) is the largest freshwater reservoir on Earth, and has been losing mass since at least 2002 (Shepherd et al., 2018; Rignot et al., 2019), thereby contributing to around 10 % of global average sea level rise since 1993 (Oppenheimer et al., 2019). About 99 % of the AIS is covered by a firn layer (Winther et al., 2001), which represents the transition of fresh snow to glacier ice. The firn layer thickness typically ranges from 40 m in the coastal zone and can exceed 100 m in the interior (van den Broeke, 2008), and also fluctuates in time due to changes in accumulation, melt and firn compaction. A detailed understanding of the firn layer and its variability is important for constraining current and future AIS mass change for three key reasons. Firstly, firn depth and density estimates are required to convert altimetry observed volume changes to mass changes, which remains a major source of uncertainty in mass balance studies (Morris and Wingham, 2015; Verjans et al., 2021). Secondly, firn depth and density estimates are required to convert column thickness to ice mass, to calculate solid ice discharge over the grounding line (Rignot et al., 2019). Thirdly, firn provides pore space in which nearly all (>94 %) the surface meltwater refreezes or is retained, thereby buffering surface meltwater and preventing mass loss by runoff (Medley et al., 2022).

From both observational and modelling studies, firn layer depth and density are known to vary greatly across the AIS, resulting from the wide range of (near) surface climatic conditions (e.g., Keenan et al., 2021), i.e. temperature, wind speed, accumulation and melt. Low temperatures in the interior (Fig. 1) cause slow densification rates resulting in a firn layer thickness (defined here as the depth of the $830 \, \mathrm{kg \, m^{-3}}$ density level) greater than $100 \, \mathrm{m}$. On the other hand, on warm and dry ice shelves, densification and melt are enhanced, resulting in denser firn layers with a depth less than $50 \, \mathrm{m}$ (van den Broeke, 2008).

Changes of AIS surface elevation are an expression of changes in the firn layer and of dynamical change of the underlying ice and bedrock. The difficulty is that these seasonal cycles and decadal changes in elevation are not solely caused by fluctuations in mass, but also for a large part by fluctuations in the firn density (Arthern and Wingham, 1998; Ligtenberg et al., 2012; Medley et al., 2022). In altimetry studies, it is therefore important to accurately separate these changes in mass- and density-change components. The mass component can subsequently be separated in ice-dynamical, bedrock and surface mass balance components (Willen et al., 2021).

Another important characteristic of the firn layer is its capability to retain meltwater in pore spaces by capillary forces and refreezing. In a warmer future climate, enhanced firn compaction, melt, and refreezing can all potentially lead to firn air depletion, thereby limiting the meltwater storage capacity of the firn. This is especially important over ice shelves, where meltwater accumulation can lead to hydrofracturing-induced ice shelf collapse (Kuipers Munneke et al., 2014; Datta et al., 2019), thereby accelerating future AIS mass loss and sea level rise (Gilbert and Kittel, 2021). Accurate information about firn conditions is thus essential both for understanding the current AIS mass balance, and to predict its future.

Firn models can fill spatial and temporal gaps between the relatively sparse observations from firn cores and snow pits. While remote sensing products provide high temporal and spatial resolution information about surface height changes, and indirectly, firn properties of the top few meters (e.g., Davis and Poznyak, 1993; Medley et al., 2015; Schröder et al., 2019; Shepherd et al., 2019), they do not provide information about the deeper firn column. Firn models are, therefore, also important to obtain vertical variations in density, and can be used to perform future simulations of firn depth and density in response to climate change scenarios (Ligtenberg et al., 2012; Kuipers Munneke et al., 2014).

Firn models can roughly be divided in two classes: physics-based and semi-empirical models. Physics-based models describe densification using a constitutive relationship between stress and strain for snow, while semi-empirical models use physics-based densification equations in combination with parameters that are calibrated with observed density profiles. Semi-empirical models neglect some processes such as drifting-snow compaction (e.g., Sommer et al., 2018) and are calibrated with observations representative of the past or present climate, but these conditions might be not representative of a future climate. Nevertheless, semi-empirical models require a smaller number of poorly constrained parameters, such as roughness length, snow grain shape and size, and computational demands are lower (Keenan et al., 2021). This is especially advantageous for data-scarce regions and large-scale studies. Therefore, semi-empirical firn models have been used widely for correction of satellite altimetry products (e.g., Adusumilli et al., 2018; Smith et al., 2020; Willen et al., 2021) and to assess the meltwater buffering capacity of the Antarctic firn layer (Ligtenberg et al., 2014; Kuipers Munneke et al., 2014). Several studies have found that the performance of semi-empirical models is comparable to that of physics-based models (Steger et al., 2017; Vandecrux et al., 2020; Keenan et al., 2021).

Differences in the formulations of, e.g., firn densification and fresh snow density can lead to a substantial spread in modelled firn thickness and air content (Lundin et al., 2017; Verjans et al., 2021). Using statistical emulation of firn densification models, Verjans et al. (2021) quantified how different sources of uncertainty contribute to the total uncertainty in modelled firn thickness change of the East Antarctic ice sheet. They show that, at the basin scale, the ensemble uncertainty in firn thickness changes ranges from 0.2 to 1.0 $\mathrm{cm\,yr^{-1}}$, which amounts to a 15-300 % relative uncertainty. While differences in climate forcing have

the largest influence on the spread, in basins with high snowfall and a large spatial variability of climatic conditions, firn densification formulation also has a substantial contribution to the spread (up to 46 %). In basins with recent increases in snowfall rates the contribution of fresh snow density to the total uncertainty is also substantial (up to 28 %).

    In this study, we have improved the semi-empirical IMAU Firn Densification model (IMAU-FDM) (Ligtenberg et al., 2011) by updating (i) the fresh snow density and (ii) the firn densification parameterizations, and (iii) by using updated atmospheric

forcing fields. In addition, the model has been calibrated and evaluated using 112 in situ firn core density measurements across the AIS. This version 1.2A of IMAU-FDM supersedes the previous version (Ligtenberg et al., 2011). Here, we present the simulated contemporary characteristics of the Antarctic firn layer for the period 1979-2020. We focus on spatial, seasonal and decadal variability in firn thickness and firn air content (FAC). To evaluate the decadal surface elevation changes and seasonal cycles we compare the simulations to a multi-mission remote sensing surface height change product for the period 1992-2015

(Schröder et al., 2019). Finally, we test how sensitive the modelled firn layer is to uncertainties in model variables and to the spin-up procedure.

## 2    Materials and methods

### 2.1    IMAU-FDM

IMAU-FDM is a semi-empirical 1D firn densification model that simulates the time evolution of firn density, temperature,

liquid water content and surface height changes due to firn and SMB processes. The model employs up to 3000 layers of 3 to 15 $\mathrm{cm}$ thickness, which represent the firn properties in a Lagrangian way. Abbrevations of all versions of IMAU-FDM used in this study are listed in Table 1. The model was originally developed by Helsen et al. (2008), and updated by Ligtenberg et al. (2011) to a version abbreviated here to FDM v1.1. Brils et al. (2022) recently improved IMAU-FDM applied to the Greenland ice sheet by including a refined parameterization of the thermal conductivity to version FDM V1.2G. The various versions

of IMAU-FDM have been extensively evaluated against firn density and temperature observations from both Greenland and Antarctica (Ligtenberg et al., 2011; Kuipers Munneke et al., 2015; Brils et al., 2022). We further improved the model to FDM v1.2A by updating (i) the fresh snow density and (ii) the firn densification parameterizations, as discussed below. The model is applied to the AIS, and we use this term to broadly include the East Antarctic, West Antarctic and Antarctic Peninsula ice sheets, as well as the ice shelves and disconnected glaciers and ice caps on the continent.

**Table 1.** Abbreviations and characteristics of IMAU-FDM versions used in this study.

| Abbreviation | IMAU-FDM version | Forcing | Fresh snow density | $MO_{830*}$ fit |
|---|---|---|---|---|
| FDM v1.2A | FDM v1.2 Antarctica | RACMO2.3p2, ERA-5 | Eq. (2); this study | Power (Eq. 5) |
| FDM FS-K | FDM v1.2 Antarctica | RACMO2.3p2, ERA-5 | Eq. (1); Kaspers et al. (2004) | Power (Eq. 5) |
| FDM FS-L | FDM v1.2 Antarctica | RACMO2.3p2, ERA-5 | Eq. (2); Lenaerts et al. (2012) | Power (Eq. 5) |
| FDM v1.2A-log | FDM v1.2 Antarctica | RACMO2.3p2, ERA-5 | Eq. (2); this study | Log (Eq. 4) |
| FDM v1.1p1 | FDM v1.1 | RACMO2.3p1, ERA-Interim | Eq. (1); Kaspers et al. (2004) | Log (Eq. 4) |
| FDM v1.1p2 | FDM v1.1 | RACMO2.3p2, ERA-5 | Eq. (1); Kaspers et al. (2004) | Log (Eq. 4) |
| FDM v1.2G | FDM v1.2 Greenland | RACMO2.3p2, ERA-5/Interim | Fausto et al. (2018) | Log (Eq. 4) |

### 2.1.1 Fresh snow density

The fresh snow density ($\rho_s$; $\mathrm{kg\,m^{-3}}$) is an important, but poorly constrained upper boundary condition of firn models. FDM v1.1 used a fresh snow density parameterization based on Kaspers et al. (2004) with a correction by Helsen et al. (2008), which has been calibrated over Antarctica and yields density values that typically represent the first $0.5\,\mathrm{m}$ of the snowpack. It varies as a function of annual average surface temperature ($\bar{T}_s$; K), 10-m wind speed ($\bar{V}_{10}$; $\mathrm{m\,s^{-1}}$) and accumulation ($\dot{b}$; $\mathrm{mm\,w.e.\,yr^{-1}}$):

$$\rho_s = A + B\bar{T}_s + C\bar{V}_{10} + E\dot{b} \qquad (1)$$

in which $A$, $B$, $C$ and $E$ are fit coefficients. Snow crystal type and degree of riming are indeed temperature dependent, and previous work shows that fresh snow density increases with increasing temperature (Judson and Doesken, 2000). Also, strong winds during snowfall cause crystal breaking, thereby reducing the snow effective grain size (Sato et al., 2008), which results in more efficient packing and increased fresh snow densities. However, at least two concerns can be raised against Eq. (1). First, due to the partly similar spatial patterns of accumulation and temperature in Antarctica, a statistical relationship between fresh snow density and accumulation may exist, however this relationship has no obvious physical grounds. Other studies have not found a dependency between annual accumulation and fresh snow density in Antarctica, Greenland and the Alps (Lehning et al., 2002; Fausto et al., 2018; Lenaerts et al., 2012). Second, Eq. (1) neglects the impact of the meteorological conditions at the time of deposition, which can be highly variable in time. Therefore, we tested the more recent parameterization of Lenaerts et al. (2012), which has been calibrated over Antarctica, in which $\rho_s$ is a function of instantaneous surface temperature ($T_s$; K) and 10-m wind speed ($V_{10}$; $\mathrm{m\,s^{-1}}$):

$$\rho_s = A + BT_s + CV_{10} \qquad (2)$$

in which $A$, $B$ and $C$ are fit coefficients. The resulting density values are representative of the fresh snow layer density, which typically represents the upper few cm of the snowpack. Within days after deposition, the crystal structure of freshly deposited

snow breaks down due to wind and redistribution of drifting snow, which increases the surface snow density (Groot Zwaaftink et al., 2013). IMAU-FDM does not calculate this densification by wind-packing and redistribution of drifting snow. The fit coefficients $A$, $B$ and $C$ in Eq. (2) for the fresh snow density are recalibrated to improve the fit of the simulated with observed surface snow densities, defined as the top 0.5 m of the firn column, as this best matches the thickness of the sampled layer. These fit coefficients are then used to simulate the fresh snow density, i.e. the density of the fresh snow that is added on top of the firn column. This recalibration is described in Section 3.1. As we only use 10 additional measurements in the evaluation dataset compared to the calibration dataset, we also perform a 10-fold cross evaluation.

### 2.1.2 Dry snow densification rate

Over time, the freshly deposited snow is buried and becomes denser. The rate of the dry firn densification ($d\rho/dt$) is calculated using the semi-empirical equation of Arthern et al. (2010):

$$\frac{d\rho}{dt} = D\dot{b}g(\rho_i - \rho)e^{(-\frac{E_c}{RT} + \frac{E_g}{RT_{ave}})} \tag{3}$$

where $D$ is a constant, $\dot{b}$ is the long-term average accumulation rate ($\mathrm{kg\,m^{-2}\,y^{-1}}$), $g$ is the gravitational acceleration, $\rho_i$ is the density of bubble free ice ($917\ \mathrm{kg\,m^{-3}}$), $R$ is the gas constant, $\rho$ is the layer density ($\mathrm{kg\,m^{-3}}$), $T$ is the instantaneous layer temperature (K), $T_{ave}$ is the long-term average surface skin temperature (K), and $E_c$ and $E_g$ are the activation energies for creep ($60\ \mathrm{kJ\,mol^{-1}}$) and grain-growth ($42.4\ \mathrm{kJ\,mol^{-1}}$), respectively. $\dot{b}$ is used as a proxy of the overburden pressure, thereby approximating the accumulation rate as constant. Using FDM v1.2G for Greenland, Brils et al. (2022) show that assuming a constant $\dot{b}$ introduces a minor error in the load experienced by a layer of firn (e.g. <3.2 % at Summit and <1.9 % at Dye-2). The constant $D$ has different values above (0.03) and below (0.07) the critical density level $\rho = 550\ \mathrm{kg\,m^{-3}}$ to represent the distinct densification mechanisms: for $\rho < 550\ \mathrm{kg\,m^{-3}}$, densification mainly occurs by settling and sliding of grains, and for $\rho > 550\ \mathrm{kg\,m^{-3}}$, it mainly occurs by deformation, recrystallization and molecular diffusion (Herron and Langway, 1980).

By comparing simulated and observed depths of the 550 and 830 $\mathrm{kg\,m^{-3}}$ density levels ($z_{550}$ and $z_{830}$, respectively), Ligtenberg et al. (2011) found that Eq. (3) requires dimensionless correction terms $MO_{550}$ for $\rho < 550\ \mathrm{kg\,m^{-3}}$ and $MO_{830*}$ for $550 < \rho < 830\ \mathrm{kg\,m^{-3}}$, which are defined as the ratio of modelled and observed values of $z_{550}$ and $z_{830*}$, where $z_{830*} = z_{830} - z_{550}$. The correction terms $MO_{550}$ and $MO_{830*}$ are added as a multiplier to Eq. (3); MO values below one reduce the densification rate, and values above one enhance the densification rate. $MO_{550}$ and $MO_{830*}$ are chosen as functions of the long-term mean accumulation rate. Ligtenberg et al. (2011) and Brils et al. (2022) used logarithmic correction functions, thus:

$$MO = \alpha - \beta \ln(\dot{b}) \tag{4}$$

in which MO is either $MO_{550}$ or $MO_{830*}$, and $\alpha$ and $\beta$ are fit coefficients which differ for $MO_{550}$ and $MO_{830*}$. In FDM v1.1 a minimum value of 0.25 is imposed on $MO_{550}$ and $MO_{830*}$. For Greenland and Antarctica, different values for $\alpha$ and $\beta$ have been used. Here, we also tested a power-law function for $MO_{830*}$:

$$MO_{830*} = \delta \dot{b}^{-\epsilon} + \phi \tag{5}$$

in which $\delta$, and $\epsilon$ and $\phi$ are fit coefficients. The fit coefficients in Eq. (4) and (5) are calibrated, which is described in Section 3.2. As we only use 10 additional measurements in the evaluation dataset compared to the calibration dataset, we also perform a 10-fold cross evaluation.

### 2.1.3 Heat conduction, meltwater percolation and refreezing

The conduction of heat is simulated by using a one-dimensional heat transfer equation, which couples vertical heat conduction to temperature gradients through the thermal conductivity of firn. In FDM v1.1 the thermal conductivity is a function of firn density only (Anderson, 1976), and in FDM v1.2G this has been extended to a function of density and temperature (Calonne et al., 2019), which is also included in FDM v1.2A. The lower boundary condition assumes a constant heat flux across the lowest layer, where the deep temperature is allowed to change along with long-term changes in surface temperature and/or internal heat release. Meltwater percolation is simulated using the bucket method, whereby each firn layer has a maximum irreducible water content that decreases with increasing density (Coléou et al., 1999). The meltwater can percolate through all layers in a single timestep. The meltwater (partly) refreezes when it reaches a layer with a temperature below the freezing point. To capture the percolation and refreezing of meltwater more accurately, the model timestep is reduced from 3600 to 300 s for locations with melt. Further details and evaluation of heat conduction, percolation and refreezing in IMAU-FDM are provided by Brils et al. (2022).

## 2.2 Model strategy

An initial firn layer is obtained by spinning up the model over a reference period until the firn layer is in equilibrium with the surface climate. The equilibrium is roughly reached when the entire firn column, defined here as the depth where the density $\rho$ reaches $830 \mathrm{~kg\,m^{-3}}$, is fully refreshed by accumulation. The required spin-up time in years is therefore calculated as the total mass of the modelled firn layer divided by the annual accumulation in w.e. (Kuipers Munneke et al., 2015). As no obvious strong long-term trends have been detected in Antarctica's surface climate and SMB during the period 1979-2020 (e.g., Favier et al., 2017; Mottram et al., 2021), and no reliable surface climate fields are available prior to 1979, the spin-up forcing is obtained by looping over the 1979-2020 forcing data. Thereby, we make the rather strong assumption that this 42-year period is representative for the past 100-1000 years and that the firn layer is in equilibrium with that climatic period. In the actual simulation following the spin-up, a minor trend ($<0.6 \mathrm{~mm\,yr^{-1}}$ averaged over the AIS) in total FAC remains, because at the bottom of the column, ice with a density between $830 - 917 \mathrm{~kg\,m^{-3}}$ slowly replaces bubble free ice ($\rho = 917 \mathrm{~kg\,m^{-3}}$), which originates from the initialization of the firn column prior to the spin-up. This trend is removed before further analysis of the results.

## 2.3  Surface elevation change

IMAU-FDM quantifies the impact of firn and SMB processes on the depth of the firn layer. The resulting vertical velocity of the firn top surface ($v_{tot}$) is defined as the sum of these components:

$$v_{tot} = v_{snow} + v_{sub} + v_{snd} + v_{melt} + v_{fc} + v_{by} + v_{ice} \tag{6}$$

where $v_{snow}$ represents the vertical velocity component as a result of snowfall, $v_{sub}$ of sublimation/riming, $v_{snd}$ of snowdrift erosion/deposition, $v_{melt}$ of snowmelt, $v_{fc}$ of firn compaction and $v_{by}$ represents the vertical motion associated with the changing ice shelf draft when the mass of the firn layer changes. $v_{ice}$ represents the downward/upward movement of the surface by the local divergence/convergence of the ice flow, driven by the long-term vertical mass flux through the lower boundary of the firn column. In a steady-state firn layer, this equals the mass flux at the upper boundary. $v_{ice}$ is therefore equal to the long-term average annual SMB ($\mathrm{kg\,m^{-2}\,yr^{-1}}$) over the reference period divided by $\rho_i$, but of opposite sign. All components are derived from run-time output of IMAU-FDM. The minor residual trend in d$H$/d$t$ due to increasing air content in the ice below the firn layer is added to $v_{ice}$. $v_{by}$, only relevant over ice shelves, is equal to the negative change in firn mass divided by the density of sea water. The contributions of ice-dynamical imbalance and bedrock motion to changes in the surface elevation are not included in the current model.

## 2.4  RACMO2.3p2 forcing

IMAU-FDM is forced at the upper boundary with 3-hourly fields of snowfall, sublimation, snowdrift erosion, 10-m wind speed, surface temperature, snowmelt and rainfall. In this study, FDM v1.2A is driven with ERA-5 reanalysis data (Hersbach et al., 2020), dynamically downscaled with the regional atmospheric climate model RACMO2.3p2 (Van Wessem et al., 2018). RACMO2.3p2 is run with a horizontal resolution of 27x27 km and the simulation covers the period 1979-2020. RACMO2.3p2 has been thoroughly evaluated for the Antarctic and Greenland ice sheets by Van Wessem et al. (2018) and Noël et al. (2018), respectively. In comparison to RACMO2.3p1, RACMO2.3p2 employs upper-air relaxation, has updated topography, and a retuned cloud scheme and modified snow properties. Upper-air relaxation is the indiscriminate nudging applied to the upper part of the atmosphere only. The most important effects of these changes are (i) increased snowfall in the ice sheet interior, (ii) reduced snowdrift sublimation and (iii) increased snowmelt (Van Wessem et al., 2018). Several recent studies use output from FDM v1.1 forced with RACMO2.3p2 over the AIS (e.g., Schröder et al., 2019; Keenan et al., 2021), but the impact of the improved forcing has not yet been described. As snowfall and surface melt are defining drivers of the state of the firn, we also assess the impact of the updated forcing in this paper. In addition to the updates of RACMO2.3p2, previous FDM simulations for Antarctica used dynamically downscaled ERA-Interim reanalysis data, which has been replaced by ERA-5 in 2019. This results in an improved forcing. For example, surface mass balance anomalies in Dronning Maud Land from 2006 to 2018 are better captured in ERA-5 compared to ERA-Interim (Gossart et al., 2019). We expect that forcing differences have a larger impact on the temporal evolution of the surface height compared to adjustments in the fresh snow density and dry snow densification parameterizations (Verjans et al., 2021).

## 2.5 Observational data

To calibrate and evaluate the firn model, we collected published firn density profiles from widely varying locations across the AIS (Fig. 1). We used 112 density profiles from firn cores and 8 density profiles from neutron density probe measurements, combining multiple datasets (van den Broeke, 2008; Schwanck et al., 2016; Bréant et al., 2017; Fernandoy et al., 2010; Montgomery et al., 2018; Fourteau et al., 2019; Olmi et al., 2021; Winstrup et al., 2019). The firn cores and neutron density probe measurements are mostly obtained in summer months between 1980 and 2020. For calibrating the fresh snow density parameterization, observations of the density of surface snow, defined as the top 0.5 m, from 61 firn cores could be used. Eight additional surface snow density values from neutron density probe measurements and two from firn cores (Montgomery et al., 2018) were added after calibration, and are thus only used for evaluation. We used densities of the top 0.5 m, as this depth is representative of many in situ surface snow density measurements, it generally consists of snow from multiple snowfall events, and it captures the densification by wind and redistribution of drifting snow (Brun et al., 1997). For calibrating the MO fits, only dry firn cores are used, as Eq. (3) describes dry snow densification only. A firn core is considered dry if its location experiences on average less melt than 3 % of the average annual accumulation. For the MO fits, 93 dry firn cores could be used, which improves upon Ligtenberg et al. (2011), who used data from 48 dry firn cores. In addition to the 93 dry cores used for calibration of the MO ratios, another 11 wet cores and four additional dry cores that were later added were used to evaluate the $z_{550}$ and $z_{830}$ model output. For evaluating FAC output, 31 firn cores could be used. Table S1 in the Supplement lists all measurements that have been used, the corresponding coordinates, methods and citations.

In order to verify whether the modelled surface elevation trends and seasonal variability align with observations, we used a multi-mission satellite AIS altimetry product covering the period January 1992 to December 2015 as presented by Schröder et al. (2019). In this product, data from the ERS-1, ERS-2, Envisat, ICESat, and CryoSat-2 missions are combined into a monthly time series on a 10x10 km grid, with the polar gap, floating ice shelves and steep terrain being excluded. Validation of this dataset with in situ and airborne observations showed that the products are successfully combined (Schröder et al., 2019). The altimetry product has been resampled to the FDM grid using bilinear resampling. The measurement precision over flat terrain ranges from 5 to 15 cm, depending on the satellite. However, as the precision decreases with increasing surface slope, the uncertainty can reach several meters in the coastal regions with steep topography. Offsets due to varying penetration depth of the radar signals (thus relevant for ERS-1, ERS-2, Envisat and CryoSat-2), caused by variations in electromagnetic properties of the ice sheet, are not included in this uncertainty estimate. These additional offsets can reach up to several centimeters per year, and can for instance increase the seasonal amplitude at continent-wide scales by up to several centimeters (Nilsson et al., 2022). While corrections are applied to account for variations in penetration depths, these corrections are in many cases unable to fully correct the artificial signals, which can e.g. be seen from the intermission variation in seasonal amplitude (Schröder et al., 2019; Nilsson et al., 2022). On the other hand, signal penetration is negligible for laser measurements (such as ICESat, ICESat-2, Operation IceBridge). However, ICESat and Operation IceBridge datasets have relatively poor spatial and temporal coverage, and we believe that the record of ICESat-2 is currently too short (only two years and two months overlap with FDM v1.2A, and no overlap with FDM v1.1p1) to provide a viable evaluation. Therefore, we used the multi-mission

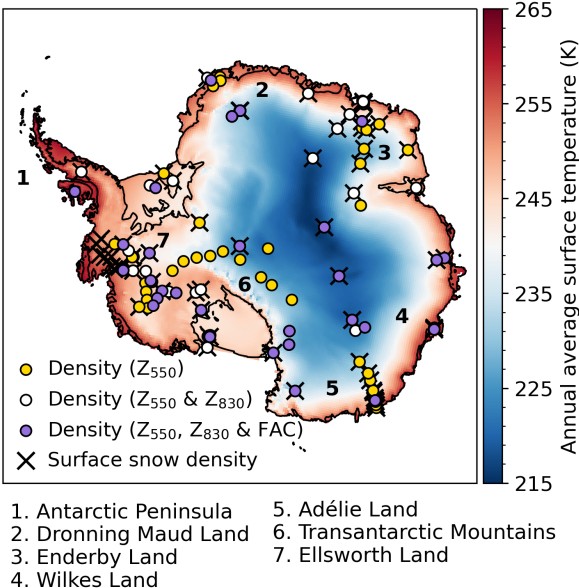

**Figure 1.** Map of Antarctica showing annual average surface temperature from RACMO2.3p2 (1979-2020) and the locations of the in situ density observations. The yellow circles indicate observational locations of the depths of the critical density level of $\rho = 550\,\mathrm{kg\,m^{-3}}$ ($z_{550}$), the white circles of the depths of the critical density levels of both $\rho = 550\,\mathrm{kg\,m^{-3}}$ ($z_{550}$) and $\rho = 830\,\mathrm{kg\,m^{-3}}$ ($z_{830}$), the purple circles of the depths of the critical density levels of both $\rho = 550\,\mathrm{kg\,m^{-3}}$ ($z_{550}$) and $\rho = 830\,\mathrm{kg\,m^{-3}}$ ($z_{830}$), and of the firn air content (FAC), and the black crosses of the surface snow densities (top 0.5 m of the firn column). The numbers indicate regions mentioned throughout the manuscript.

satellite altimetry product as presented by Schröder et al. (2019), which has a good spatial and temporal coverage. Our firn simulations only provide elevation change due to anomalies in firn thickness caused by variations in e.g. snow accumulation, surface temperature, firn densification and melt. The difference between the observed and simulated elevation change therefore represents a combination of elevation changes due to ice-dynamical imbalance, bedrock motion, errors in the firn model and its climate forcing and errors in the altimetry observations. For this comparison, we therefore exclude regions with known ice-dynamical imbalance, by using the ice-dynamical imbalance mask of Shepherd et al. (2019), which mainly consists of Pine Island, Thwaites, Totten and Getz glacier basins and the Kamb ice stream.

## 2.6 Sensitivity analysis

To study the model sensitivity to input and parameter uncertainty we performed a sensitivity analysis for the 106 locations shown in Figure S1a. These locations include 95 locations from firn core sites (the firn cores initially used for MO calibration and $z_{550}$ and $z_{830}$ evaluation are located at 95 unique grid points). To improve the representativeness of the climatic conditions over the AIS, we used 11 additional locations in this analysis that are located in high-accumulation or low-temperature regions, as these areas are underrepresented in the 95 firn core sites (Figs. S1b,c).

All the performed experiments are listed in Table 2. Verjans et al. (2021) and Lundin et al. (2017) showed that differences in accumulation, temperature, firn densification formulation and fresh snow density can lead to a substantial spread in modelled firn thickness and air content. For each of these components, we separately performed a range of model sensitivity tests, which includes the spin-up as well as the main run. The fresh snow density was uniformly varied with the RMSE from the evaluation with in situ measurements (+/- 30 $\mathrm{kg\,m^{-3}}$; Section 3.1). To test the sensitivity to uncertainty of the dry snow densification rate, we perform a simulation where we use the 95 % confidence interval boundaries of the MO fits (Eqs. 4 and 5, Fig. 3). To test the sensitivity to uncertainty in accumulation and temperature forcing, we use the standard deviation of snowfall and temperature among regional climate models and reanalysis products (Carter et al., 2022) (Fig. S2). This spread varies spatially but is assumed to be constant through time. The accumulation is varied with the ratio of the ensemble standard deviation to the ensemble mean, and the temperature with the ensemble standard deviation.

Another source of forcing uncertainty is the climate forcing during the spin-up period. As explained in Section 2.2, the spin-up forcing is obtained by looping over the 1979-2020 forcing data. However, firn core (Thomas et al., 2015, 2017; Medley and Thomas, 2019) and isotope (Stenni et al., 2017) studies show that in the Antarctic Peninsula and Ellsworth Land, the accumulation and temperature were typically about 10 % and 1 K lower during the last centuries. In addition, over the remainder of the AIS, the accumulation and temperature were approximately 5 % and 0.5 K higher or lower during the last centuries (Thomas et al., 2017; Stenni et al., 2017). To investigate the typical effect this has on the IMAU-FDM results, we include schematic sensitivity tests in which we adjusted the accumulation and temperature forcing with these values only during the spin-up period. To mimic a gradual increase in precipitation over the Antarctic Peninsula and Ellsworth Land, we performed a spin-up experiment in which the precipitation up until the third-to-last loop (the 42 year long reference period) is decreased by 10 %, in the second-to-last by 6.66 %, and in the last by 3.33 % with respect to the mean precipitation of 1979-2020. To mimic a gradual increase in temperature over the Antarctic Peninsula and Ellsworth Land, we create a spin-up experiment in which the temperature up until the third-to-last is decreased by 1 K, in the second-to-last by 0.66 K, and in the last by 0.33 K. Over the other regions of the AIS, accumulation and temperature were both decreased and increased with 5 % and 0.5 K during the spin-up period. As the uncertainties between temperature and accumulation during the spin-up period are dependent, we also performed two tests in which we simultaneously adjusted the temperature and precipitation.

For all our experiments, except for the experiments where we perturbed the coefficients of the MO fits, we re-ran our model calibration procedure to get the optimal MO fits. We then ran the model with these updated MO fits and compared the outputs to our reference FDM v1.2A run. We investigate the impact of the uncertainties on the FAC and surface elevation change. As the model set up of FDM v1.2A assumes a firn layer in steady-state, there is no net surface elevation trend in these simulations over the period 1979-2020, except for sensitivity tests in which the accumulation and/or temperature during the spin-up period are adjusted. In the remainder of the sensitivity experiments, we therefore compare surface elevation change over the period 2015-2020, as most locations experienced clear trends in surface elevation over that period. By correlating the sensitivities of FAC and surface elevation change of each experiment to climatic variables at the sample locations, we can expand the uncertainty estimates over the entire ice sheet. The empirical relations for each sensitivity experiment are shown in Figure S3.

The uncertainties of each test were combined quadratically to obtain a value for the total uncertainty, where for the spin-up experiments only the combined temperature and accumulation experiments are included.

**Table 2.** Overview of the sensitivity experiments. The inputs and parameters are adjusted by subtracting and adding the uncertainties either during the entire run or only during the spin-up. The values used are based on the references provided or on analysis within this work.

| Experiment name | Variable | Variation | When | Reference |
|---|---|---|---|---|
| MO fits | MO fits | + and -95 % confidence interval | entire run | Figs. 3a,b |
| $\rho_s$ | $\rho_s$ | + and -30 kg m$^{-3}$ | entire run | RMSE evaluation Section 3.1 |
| $T_s$ | $T_s$ | + and - spatial variable K | entire run | $\sigma$ from Carter et al. (2022) |
| $\dot{b}$ | $\dot{b}$ | + and - spatial variable % | entire run | $\sigma$ from Carter et al. (2022) |
| spin-up $T_s$ | $T_s$ | +0/+1 K and -1/-0.5 K[a] | spin-up | Stenni et al. (2017) |
| spin-up $\dot{b}$ | $\dot{b}$ | +0/+10% and -10/-5 %[a] | spin-up | Thomas et al. (2017) |
| spin-up $T_s$, $\dot{b}$ | $\dot{b}$ and $T_s$ | +0/+5 %, +0/+0.5 K and -10/-5 %, -1/-0.5 K[a] | spin-up | Stenni et al. (2017); Thomas et al. (2017) |

[a] The values on the left side of the slash indicate the variation for the Antarctic Peninsula and Ellsworth Land, and the values on the right side of the slash for the remainder of the AIS. Temperature and accumulation over the Antarctic Peninsula and Ellsworth Land are the second-to-last loop of spin-up reduced by 6.66 % and 0.66 K and in the last loop with 3.33 % and 0.33 K, see main text.

## 3  Calibration and model performance

### 3.1  Fresh snow density

To calibrate and evaluate the fresh snow density parametrization, we performed simulations with FDM v1.2A using the parameterizations of Kaspers et al. (2004) (Eq. 1), Lenaerts et al. (2012) (Eq. 2), the latter with updated constants, and of FDM v1.1p1. These simulations are abbreviated as FDM FS-K, FDM FS-L, FDM v1.2A and FDM v1.1p1, respectively. The fit coefficients and statistics of the simulations are listed in Table 3. Figure 2a shows the modelled against observed surface snow densities. FDM FS-K somewhat overestimates the surface snow densities (bias = 17.6 kg m$^{-3}$), especially for high surface density, and FDM v1.1p1 to a lesser extent (bias = 10.3 kg m$^{-3}$). It is the update in the atmospheric forcing that causes a poorer performance of FDM FS-K compared to FDM v1.1p1. FDM FS-L substantially underestimates surface snow densities (bias = -45.1 kg m$^{-3}$), but has a higher correlation with the observations than FDM FS-K. The degree of underestimation increases with increasing wind speed, which suggests that a higher value of fit coefficient C is needed to properly capture the impact of wind compaction. This aligns with the fact that IMAU-FDM does not include densification by wind packing. In the optimized fit coefficients (FDM v1.2A, Table 3) the value of C is 2.6 times higher than the value reported by Lenaerts et al. (2012). The wind dependency is also 2.4 times stronger than proposed by Kaspers et al. (2004), which is in agreement with Sugiyama et al. (2012) who found a factor 2.8. The RMSE of the surface snow density of FDM v1.2A is respectively 28 % and 23 % lower compared to FDM FS-K and FDM v1.1p1, due to both minimizing the bias and improving the representation of the variability. Compared to FDM FS-K, the surface snow density is reduced by 18 kg m$^{-3}$ on average, especially in the

**Table 3.** Fit coefficients and evaluation of the fresh snow density parameterizations (Eqs. 1,2). The fit coefficients of FDM FS-K and FDM FS-L are taken from Kaspers et al. (2004) and Lenaerts et al. (2012), respectively.

| Version | A | B | C | E | $R^2$ | RMSE | Bias |
|---|---|---|---|---|---|---|---|
| | (kg m$^{-3}$) | (kg m$^{-3}$ K$^{-1}$) | (kg s m$^{-4}$) | (kg yr m$^{-3}$ mm w.e.$^{-1}$) | | (kg m$^{-3}$) | (kg m$^{-3}$) |
| FDM v1.2A[b] | 83 | 0.77 | 11.67 | | 0.46 | 28.8 | -1.3 |
| FDM FS-K[a] | -77 | 1.5 | 6.8 | 0.075 | 0.35 | 40.2 | 17.6 |
| FDM FS-L[b] | 97.5 | 0.77 | 4.49 | | 0.37 | 55.3 | -45.1 |
| FDM v1.1p1[a] | -77 | 1.5 | 6.8 | 0.075 | 0.34 | 37.3 | 10.3 |

[a] using annual average accumulation, temperature and wind speed. [b] using instantaneous temperature and wind speed.

high-accumulation margins, and to a lesser extent in windy escarpment regions (Fig. 2b). With the 10-fold cross evaluation for FDM v1.2A, we found that the RMSE for the surface snow density slightly increases to 30.4 kg m$^{-3}$ (compared to 28.8 kg m$^{-3}$ for FDM v1.2A). The fresh snow density parameterization for Greenland used in FDM v1.2G, is only a function of annual average temperature (Brils et al., 2022), owing to a lack of co-located surface snow density, temperature and wind speed observations on the GrIS. For all sensitivity tests presented here, we calculated the associated MO fits. However, the impact of the different MO fits on the surface snow densities is small (<0.1 %).

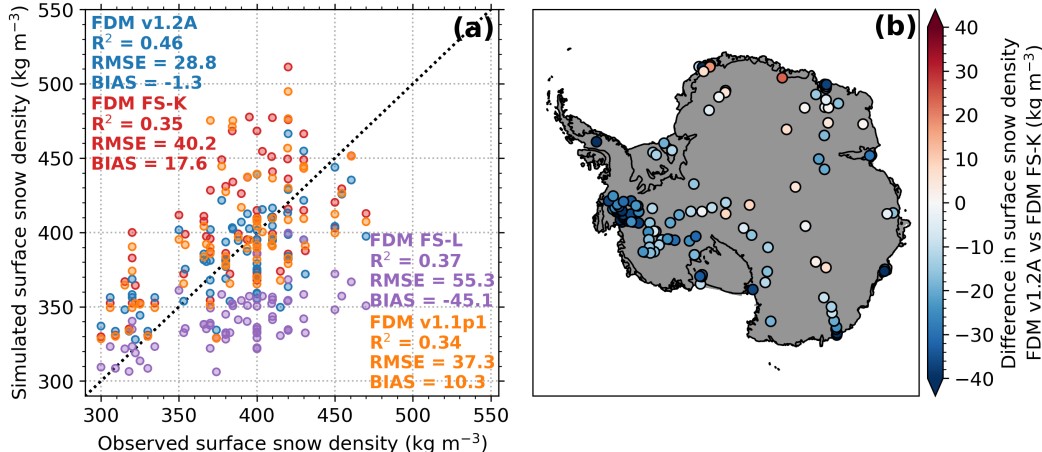

**Figure 2. (a)** Scatter plot of simulated against observed surface snow density using FDM v1.2A, FDM FS-K (Kaspers et al., 2004), FDM FS-L (Lenaerts et al., 2012) and FDM v1.1p1. **(b)** Map of average difference between the simulated surface snow densities of FDM v1.2A and FDM FS-K for the locations with observations.

## 3.2 Dry snow densification rate

To calibrate the dry snow densification rate we first performed a simulation of FDM v1.2A without MO corrections, thus only using Eq. 3. The resulting MO fits and statistics are listed in Table 4 and shown in Figures 3a,b. The optimal $MO_{550}$ fit is less steep with respect to accumulation ($\beta = 0.117$ vs 0.151) than in FDM v1.1p1. The fresh snow density is now independent of accumulation and generally lower, and hence more densification is needed at high-accumulation sites to match the density measurements. This might explain the slightly deteriorated correlation for $MO_{550}$. For the $MO_{830*}$ regression, we found that a power fit (FDM v1.2A) compared to a logarithmic fit (FDM v1.2A-log) improves correlation ($R^2 = 0.88$ compared to 0.83). Furthermore, a minimum value is now explicitly derived with the power fit, so that a rather arbitrary minimum value no longer needs to be prescribed. When the locations are rerun using the new MO values with the power fit for $MO_{830*}$, the resulting RMSE of the modelled $z_{830*}$, $z_{830}$ (firn thickness) and FAC are respectively 41, 39 and 48 % lower compared to the logarithmic fit (Figs. 3c,d). The update in atmospheric forcing without updated MO calibration does not result in an improvement in FAC (shown by the similar RMSE of FDM v1.1p1 and FDM v1.1p2, which is FDM v1.1 forced with the updated forcing RACMO2.3p2). Also, the RMSE of FAC simulated with FDM v1.2A is respectively 18 and 20 % lower compared to FDM v1.1p1 and FDM v1.1p2, which indicates a smaller improvement compared to FDM v1.2A-log (48 %). The poor performance of FDM 1.2A-log shows that the additional observations necessitated the introduction of the power fit. When we only include the seven observations that were used to calibrate FDM v1.1, and two wet firn cores, the RMSE of FAC simulated by FDM v1.2a is 2.79 m, which is 19.5 % and 9.0 % lower compared to FDM v1.1p1 and FDM v.1.1p2 for these observations.

Using a power compared to logaritmic fit results in higher $MO_{830*}$ values for low (<400 $\mathrm{mm\,yr^{-1}}$) and high (>1,400 $\mathrm{mm\,yr^{-1}}$) accumulation regimes, and lower $MO_{830*}$ values for intermediate accumulation regimes (400-1,400 $\mathrm{mm\,yr^{-1}}$) (Fig. 3b). Therefore, the densification rate is increased in high and low-accumulation regions and reduced in intermediate accumulation regions. This results in decreased firn thickness in high and low-accumulation regions, and in increased firn thickness in intermediate accumulation regions (Fig. 3d). Especially large differences are found in high-accumulation areas where the MO fit curve decreases less rapidly than FDM v1.2A-log. With the 10-fold cross evaluation for FDM v1.2A, we found that the RMSE for $z_{550}$ slightly increases to 2.33 m (compared to 2.26 m for FDM v1.2A), and the RMSE of $z_{830*}$ to 8.91 m (compared to 8.33 m for FDM v1.2A). For the 11 wet firn cores, we found that the RMSE of modelled $z_{550}$, $z_{830*}$ and FAC amounts to 3.17, 14.34 and 4.40 m, respectively, and thus larger than the RMSEs of the entire firn core dataset (2.26, 8.33 and 2.74 m, respectively).

In FDM v1.2G, the version for the Greenland ice sheet, a logarithmic $MO_{830}$ fit was applied. Dry firn cores from the GrIS cover a smaller range of average annual accumulation, 80-680 $\mathrm{mm\,yr^{-1}}$ compared to 20-960 $\mathrm{mm\,yr^{-1}}$ in Antarctica. If we only include Antarctic cores from the 80-680 $\mathrm{mm\,yr^{-1}}$ accumulation range, we find similar model fits to the data for the logarithmic model ($R^2 = 0.67$) and the power model ($R^2 = 0.68$). Another difference between FDM v1.2G and FDM v1.2A, is that in FDM v1.2G an almost constant value of 0.67 for the $MO_{550}$ fit was found, which implies a linear correlation between the densification rate and the accumulation rate. In FDM v1.2A we do find a moderate ($R^2=0.37$, $\beta = 0.12$) correlation of $MO_{550}$ with accumulation, i.e. AIS densification rate depends more strongly on accumulation than temperature compared to

**Table 4.** Fit coefficients and statistics of the MO fits (Eqs. 4,5). The fit coefficients and statistics of FDM v1.1p1 are taken from Ligtenberg et al. (2011). Please note that $R^2$ for v1.1p1 applies to the calibration dataset used in that study. The FDM v1.2A-log uses the settings of FDM v1.2A; only Eq. (3) is optimized using a logaritmic $MO_{830*}$ fit. The root-mean-squared deviation (RMSD) quantifies the deviation with respect to the fitted function.

| Version | Fit | $\alpha$ | $\beta$ | $\sigma$ | $\epsilon$ | $\phi$ | $R^2$ | RMSD |
|---|---|---|---|---|---|---|---|---|
| $MO_{550}$ FDM v1.1p1 | Log (Eq. 4) | 1.435 | 0.151 | | | | 0.43 | |
| $MO_{550}$ FDM v1.2A | Log (Eq. 4) | 1.288 | 0.117 | | | | 0.37 | 0.13 |
| $MO_{830*}$ FDM v1.1p1 | Log (Eq. 4) | 2.366 | 0.292 | | | | 0.71 | |
| $MO_{830*}$ FDM v1.2A-log | Log (Eq. 4) | 2.504 | 0.334 | | | | 0.83 | 0.10 |
| $MO_{830*}$ FDM v1.2A | Power (Eq. 5) | | | 6.387 | 0.477 | 0.195 | 0.88 | 0.09 |

the GrIS. Again, if we only include cores from the 80-680 $\mathrm{mm\,yr^{-1}}$ accumulation range this correlation weakens ($R^2$=0.08, $\beta$ = 0.07).

# 4   Spatial and temporal variability of AIS firn depth and density

## 4.1   Spatial density variability

Using the improved firn model FDM v1.2A, we simulate the spatial and temporal patterns of the AIS firn characteristics at 27x27 $\mathrm{km}$ horizontal resolution. The spatial patterns of the densities, including observed values, are shown in Figure 4. Figure 4a shows the surface snow density (top 0.5 m), which is on average 376 $\mathrm{kg\,m^{-3}}$. The patterns are primarily a reflection of the variation in fresh snow density (Eq. 2), with the lowest values on the cold and calm East Antarctic plateau and higher values in the windy escarpment regions and along the milder coastal margins. In addition to this, the surface snow density further increases over the ice shelves, where the average surface snow density amounts to 414 $\mathrm{kg\,m^{-3}}$, and over the coastal margins, where intermittent melting and refreezing of meltwater increases the near-surface density.

In line with observations (Winther et al., 2001), FDM v1.2A predicts that nearly all (>99 %) of the AIS is covered by a layer of firn. This represents the ice sheet fraction where average SMB is positive. The spatial pattern of the depths of the critical density levels $z_{550}$ and $z_{830}$ are shown in Figures 4b and c, which are on average 14.1 and 73.8 m. The patterns are roughly the inversed pattern of the surface snow density, which constitutes the upper boundary of the depth-density profile. For simplicity, henceforth we define the firn thickness as $z_{830}$. The patterns vary spatially across climatic regions with temperature, accumulation and surface melt as drivers. The highest $z_{550}$ and $z_{830}$ values (30.6 and 114.2 m, respectively) are found in the interior, where the densification rate is low due to low temperatures, and the surface snow density is low. Generally, $z_{550}$ and $z_{830}$ are small along the warm coastal margins due to faster densification, higher surface snow densities and melt. However, the $z_{830}$ values can be relatively high in high-accumulation regions along the coast, particularly in West Antarctica and the

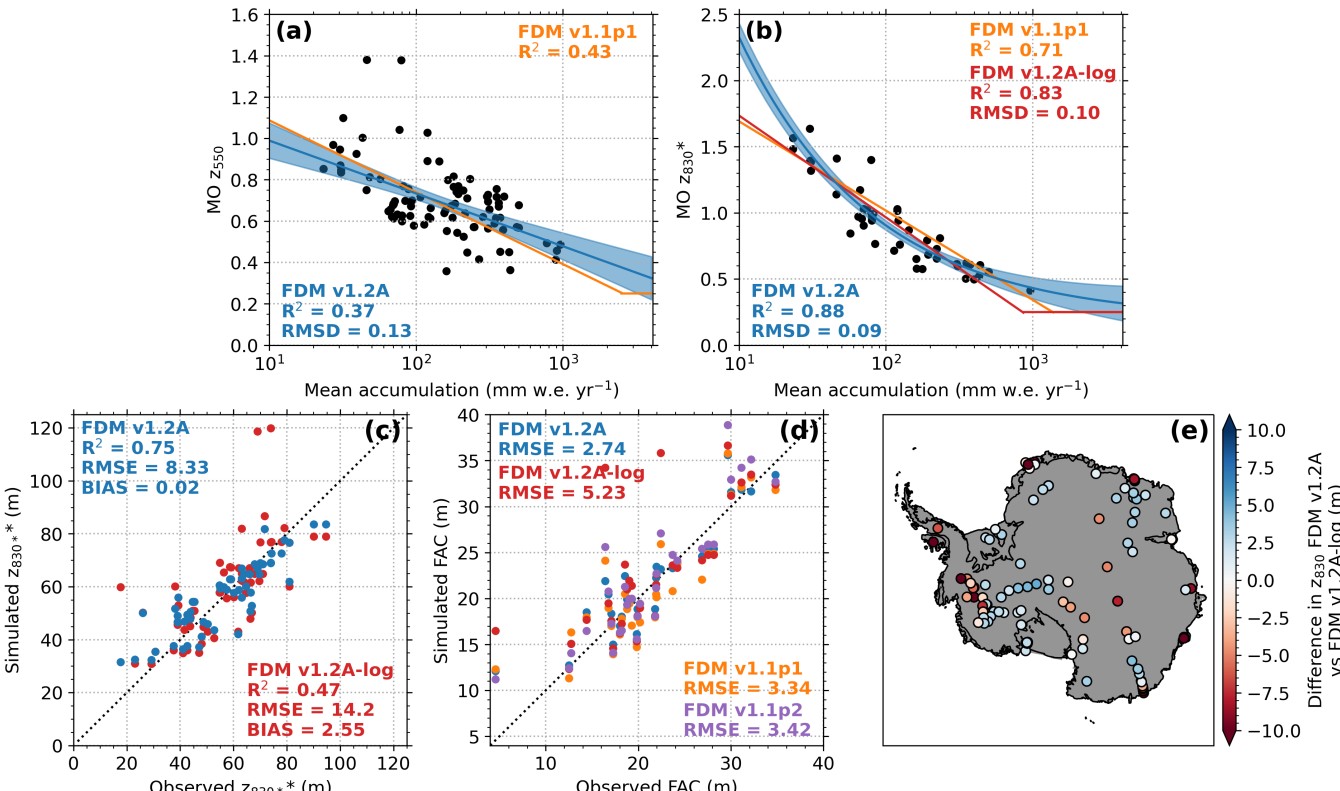

**Figure 3.** Scatter plots defining the MO ratios for (a) $z_{550}$ and (b) $z_{830}*$ as a function of the annual average accumulation including various MO fits. The blue shaded areas indicate the 95 % confidence intervals. (c) Scatter plots of the simulated against observed $z_{830*}$ ($z_{830}$-$z_{550}$) for FDM v1.2A and FDM v1.2A-log. (d) Scatter plots of the simulated against observed FAC for FDM v1.2A, FDM v1.2A-log, FDM v1.1p1 and FDM v1.1p2. (e) Map of the difference between simulated firn thickness ($z_{830}$) of the two tests (FDM v1.2A – FDM v1.2A-log) for locations with observations. The statistics and fits of the MO fits are listed in Table 4.

Antarctic Peninsula. The high accumulation rates in these regions result in younger, thus less densified snow with depth and thus thick firn layers despite relatively high temperatures. The $z_{550}$ and $z_{830}$ values are lowest in relatively warm and dry regions on and around Ross, Filchner-Ronne and Amery ice shelves, where fresh snow is buried more slowly, and the densification rates are high due to relatively high temperatures.

Figure 4d shows the age of firn at the pore close-off depth, here assumed to equal $z_{830}$, which is on average 754 yr. The combination of strongly increasing firn depth and decreasing accumulation towards the interior leads to firn ages at pore close-off depth of up to 3240 yr in central East Antarctica, whereas the pore close-off firn age in warm and wet coastal margins can be as low as 20 yr. When comparing this to firn age inferred from $\delta^{15}N$ measurements for 15 locations (Bréant et al., 2017), we find an RMSE = 231 yr, $R^2$ = 0.99. On average, the absolute errors are 25 % of the observed values. In comparison, the absolute errors of $z_{830}$ are 15 % of the observed values on average, and $R^2$ = 0.73.

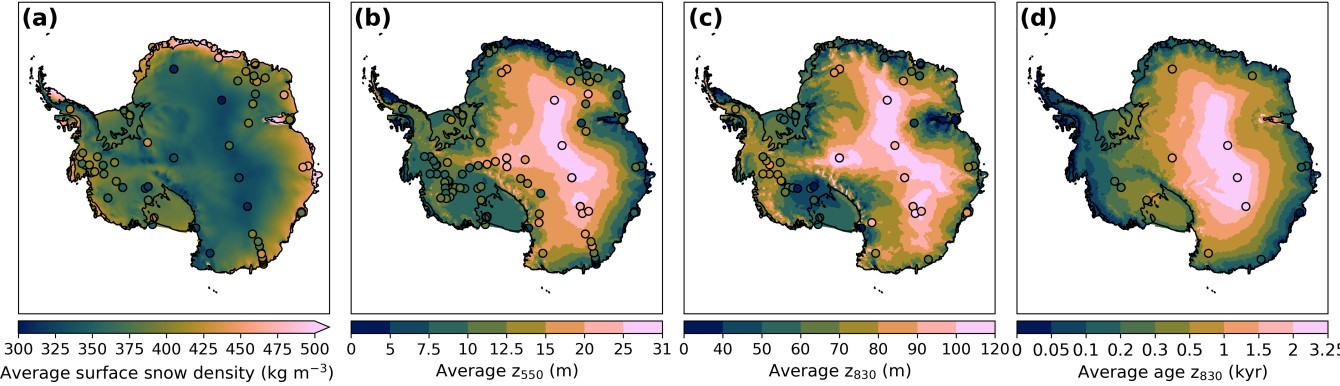

**Figure 4.** Maps from FDM v1.2A of the average **(a)** surface snow density (defined as the top 0.5 m), depths of critical density levels of **(b)** $\rho$ = 550 kg m$^{-3}$ ($z_{550}$) and of **(c)** $\rho$ = 830 kg m$^{-3}$ ($z_{830}$) (the firn thickness), and **(d)** firn age at the critical density level of (c) $\rho$ = 830 kg m$^{-3}$. Circles indicate the values found by field measurements.

### 4.2 Firn air content

Firn air content (FAC), the total (vertically integrated) pore space, is an important output of firn models as it is used to convert observed volume changes to mass changes, column thickness to ice mass, and is an indicator of the meltwater buffering capacity of the firn. FAC is expressed in m and represents the change in depth that would occur if the firn layer would be compressed to the density of glacier ice. The average FAC of the AIS according to IMAU-FDM is 22.8 m, and it varies spatially between 0 to 40 m across climatic regions (Fig. 5a). The pattern resembles the $z_{830}$ pattern (Fig. 4c); thick firn layers contain most air. High values are found in the cold interior and along parts of the coast, particularly in West Antarctica, and in areas with high accumulation. Low values are found along the warmer coastal margins, with ice shelves having an average FAC of 15.9 m. These average FAC estimates are 5 % and 7 % lower compared to values simulated with the Community Firn Model forced with MERRA-2 climate data of 24.0 and 17.0 m respectively (Medley et al., 2022).

The average FAC over the AIS simulated with FDM v1.2A is 0.6 m higher than simulated with FDM v1.1. The spatial FAC differences between FDM v1.2A and FDM v1.1 are shown Figure 5b. In the highly elevated and low-accumulation regions the updated FAC is lower, whereas in the intermediate accumulation regions the updated FAC is higher. This implies that the range of FAC is smaller and less spatially variable than previously modelled. The differences in FAC pattern can be explained by differences in (i) the MO$_{830*}$ fit, (ii) the fresh snow density, (iii) snowfall and (iv) snowmelt. FDM v1.2A has higher MO$_{830*}$ values for low- and high-accumulation regions compared to FDM v1.1p1, which results in a faster densification and therefore lower FAC values. In intermediate accumulation regions the MO$_{830*}$ values are lower, resulting in a slower densification and higher FAC values. The fresh snow densities of FDM v1.2A are on average lower (19.1 kg m$^{-3}$, Fig. 2), which contributes to FAC values being higher on average. On top of that, increased accumulation in the interior and increased snowmelt along the coastal margins in RACMO2.3p2 compared to RACMO2.3p1 (Van Wessem et al., 2018), also contribute to the general pattern of higher FAC in the interior and lower values along the coastal margins. However, in the highly elevated and lowest

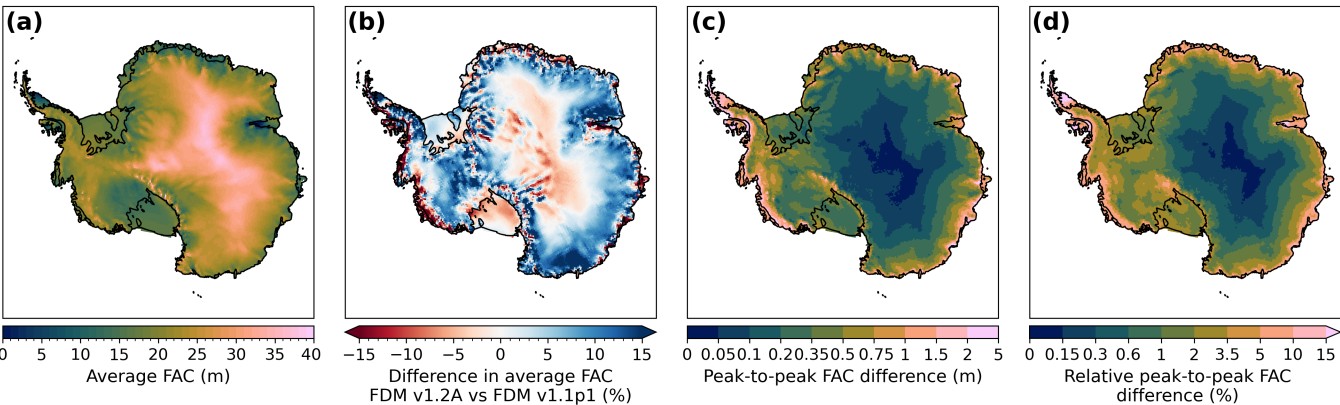

**Figure 5.** Maps from FDM v1.2A of average **(a)** firn air content (FAC), **(b)** the difference in FAC (FDM v1.2A - FDM v1.1p1), **(c)** peak-to-peak FAC difference, **(d)** relative peak-to-peak FAC difference compared to the average FAC from (a). The peak-to-peak is defined as the difference between the highest and the lowest FAC value occurring during the study period (1979-2020).

accumulation regions, we find a reduction in FAC, which implies that the $MO_{830*}$ update outweighs the effects of higher accumulation and lower fresh snow density in these regions.

Figures 5c and d focus on temporal variability in FAC. Figure 5c shows the peak-to-peak FAC difference, which represents the difference between the highest and the lowest FAC values occurring during the study period (1979-2020). The peak-to-peak FAC difference is low in the interior, less than 0.1 m, despite the high FAC. Along the coastal margins, the peak-to-peak FAC difference is higher, up to 5 m, even though the FAC is generally lower. Therefore, the relative peak-to-peak FAC difference is low in the interior (<1 %) and high near the ice sheet margins (>5 %) (Fig. 5d). The average relative peak-to-peak FAC

difference for the ice shelves, excluding the two largest ones, the Ross and Filchner-Ronne ice shelves, amounts to 13 %. This high peak-to-peak variability is caused by a combination of substantial temporal variability in accumulation and melt. The temporal FAC variability and the driving processes are discussed further in Sections 4.3 and 4.4 below.

### 4.3    Seasonal firn air content and firn depth variability

In order to better understand the FAC temporal variability and its drivers, we decompose the signal into seasonal and interannual

contributions. Figure 6a shows the average seasonal cycle of the vertical firn surface velocity ($v_{tot}$) including its components from Eq. (6). Four main processes drive the seasonality of the firn thickness: snowfall ($v_{snow}$), firn densification ($v_{fc}$), melt ($v_{melt}$) and sublimation ($v_{sub}$). Averaged over the ice sheet, snowfall is highest in winter, while firn densification, melt and sublimation peak in summer. The net result is that the firn thickness and FAC steadily increase over the winter, due to enhanced accumulation and reduced firn densification, melt and sublimation (Fig. 6b). This is followed by a sharp decrease over the

summer due to reduced accumulation and increased firn densification, melt and sublimation. The firn height change by ice flux divergence ($v_{ice}$) is constant by definition, while the AIS average impacts of snowdrift ($v_{snd}$) and buoyancy ($v_{by}$) are small (<1

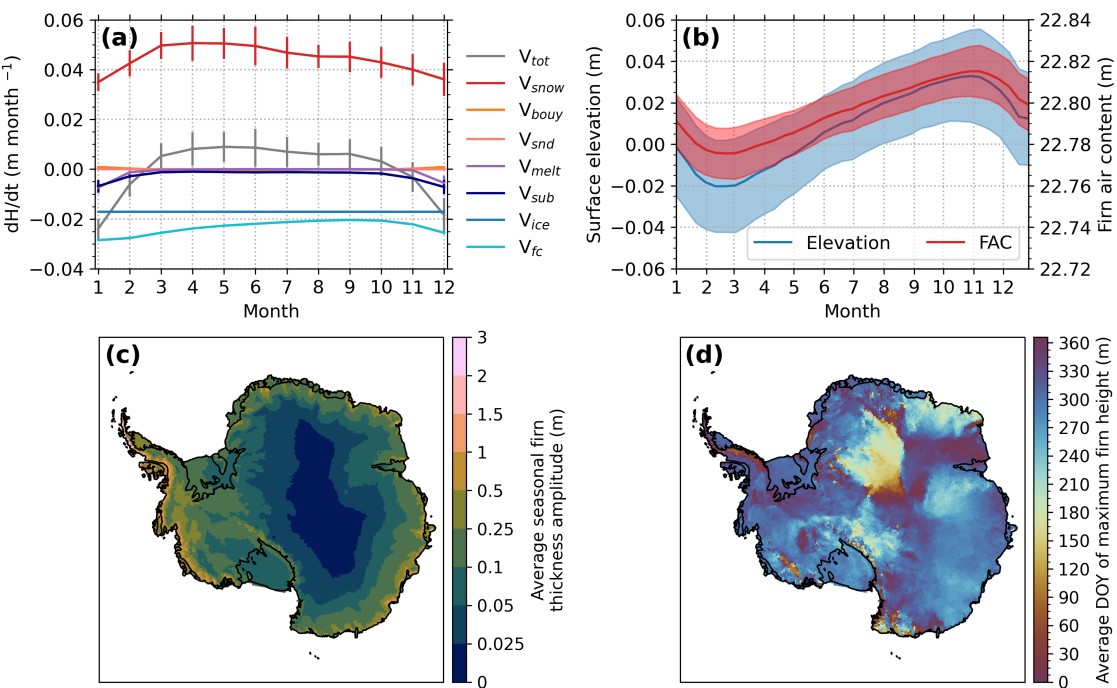

**Figure 6.** Ice sheet averaged seasonal cycle from FDM v1.2A of the **(a)** vertical firn surface velocity including its separate components from Eq. 6, and of **(b)** surface elevation and firn air content (FAC). The bars and colored areas indicate the interannual standard deviations. Maps from FDM v1.2A **(c)** of the average seasonal firn thickness amplitude and **(d)** of average day of the year of maximum firn height.

%). The vertical bars in Figure 6a indicate the interannual variability, which is driven solely by snowfall except for the two summer months, which also have contributions from sublimation and melt.

The seasonal amplitudes of the ice-sheet-wide firn thickness and FAC are 3.1 and 2.1 cm, which amounts to Antarctic wide integrated volumes of 428 and 307 km$^3$, respectively. This agrees with the modelled firn thickness amplitude of 3 cm found by Medley et al. (2022). The averages of all local seasonal amplitudes are considerably higher, 6.8 and 4.2 cm, amounting to volume changes of 892 and 562 km$^3$, respectively (Fig. 6c). The reason is that there is a spatial variation in timing of the seasonal maximum and minimum (Fig. 6d). The difference between the firn thickness and FAC seasonal cycles (6.8 vs 4.2 cm) suggests that around 62 % of the seasonal surface elevation fluctuations are caused by a change in air content rather than actual mass change. This result implies that not including the seasonal cycles of FAC in altimetry studies can generate significant errors, especially when investigating changes in subannual periods and in regions with large seasonal SMB cycles.

Figure 6c shows that the seasonal firn thickness amplitude can be up to 3 m in the western Antarctic Peninsula, caused by a rare combination of high snowfall and strong melt. Maps of mean annual precipitation, melt and sublimation are shown in Figure S9. In regions with little melt, such as coastal West Antarctica, the values are typically 1 m. In interior East Antarctica, values of less than 2.5 cm are found, as the annual average accumulation is low and melt absent. The average day of the year

of the maximum firn height generally occurs during late winter or spring, before the summer decrease in accumulation and increase in temperature (Fig. 6d). In some regions accumulation and firn depth peak in late summer or early winter, e.g., along the Weddell Sea coast, in Enderby Land, Adélie Land and parts of the interior.

## 4.4 Interannual firn air content and firn depth variability

Figure 7 shows time series of integrated FAC, the cumulative surface temperature anomalies (against the long-term mean) and the vertical firn surface velocity anomalies (against the long-term mean) for the entire AIS (Fig. 7a) and above/below 2,000 m a.s.l. (Figs. 7b,c). In addition, the surface velocity anomalies are broken down into their components (Eq. 6). The figure confirms that the seasonal firn thickness and FAC variability are driven by snowfall, densification, melt and sublimation resulting in a firn thickness peak in late winter. The figure also reveals decadal firn thickness and FAC variability mainly

resulting from fluctuations in snowfall (46 % of the total cumulative dH anomaly), and that firn densification, despite the slower response timescale, reduces these snowfall-induced fluctuations by about 20 % (calculated from difference in standard deviation (sd) between the cumulative anomalies of $v_{snow}$ and $v_{snow+fc}$). We also find interannual variability in melt, but snowfall variability dominates. Below 2,000 m a.s.l. the variability in firn thickness and FAC are substantially larger, which can be attributed to the higher accumulation (70 % of the total accumulation falls on only 40 % of the ice sheet), in combination with

larger temporal variability in snowmelt and firn densification. Above 2,000 m a.s.l. the relative contribution of firn densification to the total cumulative dH anomaly is larger (32 %) than below 2,000 m a.s.l., where this is only 14 %. This difference can partly be explained by the larger interannual temperature variability above 2,000 m a.s.l (sd in annual means = 0.78 K compared to 0.48 K), the high sensitivity of firn compaction to temperature variability for firn at low temperatures, and the absence of melt. The interannual variabilities in snowfall and temperature are related to low and mid-latitude climate oscillations such as

the El Niño-Southern Oscillation and the Southern Annular Mode (Kwok and Comiso, 2002; Marshall et al., 2017). Figure 7 shows that the magnitude of the decadal FAC variability (sd = 1.8 cm) constitutes a large fraction (67 %) of the total surface elevation change variability (sd = 2.7 cm). This reiterates the necessity of the common practice to remove decadal variability in FAC from altimetry signals to avoid errors in mass change estimates.

## 5 Comparison with altimetry and FDM v1.1 surface elevation change

In sections 3.1, 3.2 and 4 we presented and evaluated the spatial variation in firn thickness and density. To also evaluate the temporal variation, we compare our simulated elevation trends and seasonal variability to multi-mission satellite altimetry observations reported by Schröder et al. (2019) for the period 1992-2015.

### 5.1 Seasonal amplitude

The altimetry observations prior to 2003 exhibit a larger short-term variability as a result of the lower measurement precision

of the satellites during that period (Schröder et al., 2019; Nilsson et al., 2022), and are excluded from the seasonal amplitude comparison here. To enable comparison with the FDM v1.1p1 simulation, we focus on the period 2003-2015. To reduce the

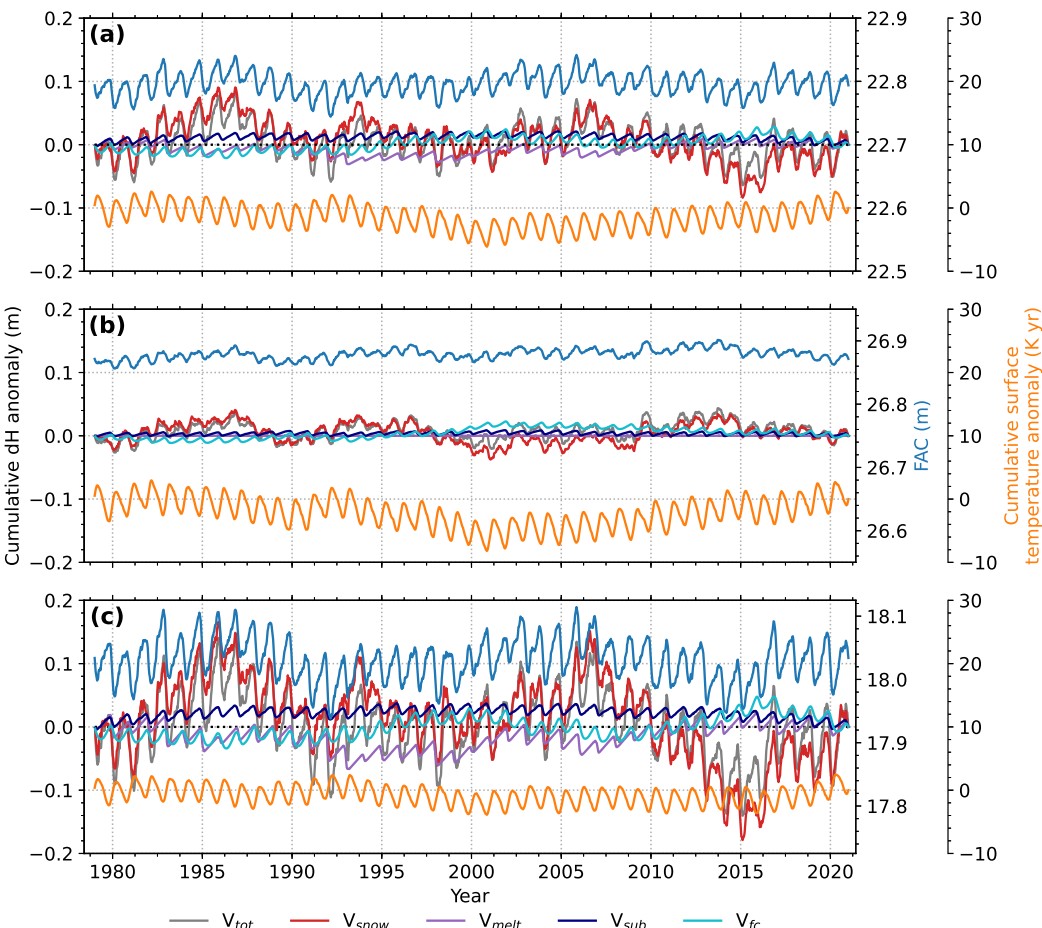

**Figure 7.** Time series from FDM v1.2A of FAC, of the cumulative anomalies (against long-term mean) of surface temperature, of the vertical firn surface velocity, and of its separate components from Eq. (6). Time series are shown for **(a)** the entire ice sheet, **(b)** the part of the ice sheet situated above 2,000 m a.s.l. and **(c)** the part of the ice sheet situated below 2,000 m a.s.l.

impact of the long-term trends we remove a linear trend from all datasets. After doing so, we find that the seasonal amplitude averaged over the ice sheet of the altimetry observations amounts to 5.2 cm, of FDM v1.2A to 3.1 cm (60 % of the observed amplitude) and of FDM v1.1p1 to 2.5 cm (48 % of the observed amplitude). The higher seasonal amplitude of FDM v1.2A
results from increased snowmelt along the coastal margins in RACMO2.3p2 compared to RACMO2.3p1 in combination with a lower fresh snow density (Fig. 2). Using updated corrections and a normalization of seasonal amplitudes from different missions, Nilsson et al. (2022) found that seasonal amplitudes during 1992-2016 are 47 % smaller than the ones from Schröder et al. (2019). This indicates that our AIS-wide seasonal amplitude lie within the range of observational uncertainty.

## 5.2 Decadal variability

In Figure 8 we compare observed (Fig. 8a) and simulated 2003-2015 surface elevation trends. To obtain these numbers, a linear regression is fitted to monthly values for the period 2003-2015. The FDM v1.2A surface elevation trend over this period is positive in Dronning Maud Land, Enderby Land, and negative in most other regions (Fig. 8b). FDM v1.1p1 (Fig. 8c) has roughly similar patterns as FDM v1.2A, but the regional magnitudes differ. Figure 8d shows the residual surface elevation trend of FDM v1.2A, which is calculated by subtracting the FDM v1.2A elevation trend from the altimetry trend, and thus

combines unresolved trends and the errors in the firn model and altimetry. In Figure 8e we show the difference in altimetry agreement between FDM v1.2A and FDM v1.1p1, which is calculated by subtracting the absolute residual surface elevation trend compared to altimetry (Fig. 8d) of FDM v1.1p1 from the similary derived absolute residual of FDM v1.2A. The blue areas indicate an improvement in altimetry agreement of FDM v1.2A compared to FDM v1.1p1. The average absolute residual trend of FDM v1.2A compared to altimetry has been reduced by 17 % compared to FDM v1.1p1 (from 2.6 to 2.1 $\mathrm{cm\,yr^{-1}}$).

The improvement is most notable in Dronning Maud Land, Wilkes Land and Adélie Land. In Dronning Maud Land the FDM v1.2A trend has become more positive, and in Wilkes Land and Adélie Land the trend has become less negative (Fig. 8b,c, for region names see Fig. 1). In the Antarctic Peninsula and Ellsworth Land the altimetry agreement of FDM v1.2A has reduced compared to FDM v1.1p1. In these regions, a substantial residual positive trend remains (>10 $\mathrm{cm\,yr^{-1}}$) (Fig. 8d), which could be due to a long-term increase in accumulation and/or ice-dynamical thickening. Smith et al. (2020) used ICESat data to show

that the mass gain is mainly located along the steep slopes of the Antarctic Peninsula and decreases with distance from the ocean, which is indicative of an increase in snow accumulation. The increase in accumulation on centennial time scales in these regions (e.g., Thomas et al., 2015, 2017; Medley and Thomas, 2019) implies that the actual firn column is not in balance with the 1979-2020 climate, as assumed in the FDM spin-up procedure. As a result, the vertical downward ice flow (Eq. 6) is overestimated, which results in underestimated surface elevation change. This is further discussed in the next paragraph.

In Figure 9 we compare time series of altimetry observed elevation change over the period 1992-2015 to simulated elevation change for eleven locations across the AIS. These locations were selected as they cover the main distinct patterns from Figure 8 and have continuous observations. The time series have been smoothed using a 6-monthly moving average window, in order to reduce the impact of seasonal variations in radar penetration depth. The red shaded area is the aggregated uncertainty range from the sensitivity analysis, which is primarily determined by uncertainty in the climate forcing during the spin-up period

(see next section for details of the sensitivity analysis). The agreement of FDM v1.2A compared to FDM v1.1p1 has improved ($R^2$ = 0.58 compared to 0.36), which is mainly due to changes in the RACMO2.3p2 forcing, given the strong similarity between elevation change of FDM v1.2A and FDM v1.1p2. The altimetry observations prior to 2003 exhibit stronger short-term variability (+9 % sd compared to after 2003), in line with the smaller measurement precision (Schröder et al., 2019). The agreement with FDM v1.2A increases after 2003 ($R^2$ = 0.93 compared to 0.64, RMSE reduced by 49 %), however the

altimetry variability remains higher than the simulated variability (+13 % sd compared to FDM v1.2A), as these more recent observations still contain noise (<11 cm over flat terrain) and have variations in radar penetration depth (Schröder et al., 2019). The uncertainty bands in Figure 9 show that a large part (73 %) of the difference between FDM v1.2A and altimetry falls

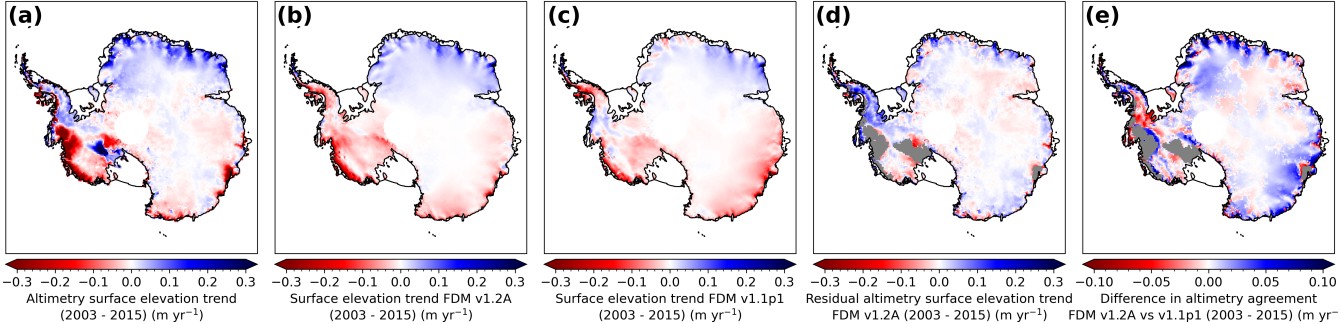

**Figure 8.** Maps of trends in surface elevation change over the period 2003-2015 from **(a)** multi-mission altimetry (Schröder et al., 2019), **(b)** FDM v1.2A and **(c)** FDM v1.1p1 obtained using linear regression. **(d)** Residual trend of altimetry minus FDM v1.2A. **(e)** Difference in altimetry agreement between FDM v1.2A and FDM v1.1p1. (absolute residual of FDM v1.1p1 compared to altimetry minus absolute residual of FDM v1.2A compared to altimetry). Regions within the ice-dynamical imbalance mask reported by Shepherd et al. (2019) are excluded in **(d)** and **(e)** and indicated by the grey shaded area.

within the uncertainty range of the firn model. As in Figure 8d, we find substantial positive residual altimetry trends ($\sim 14$ and $5 \, \mathrm{cm \, yr^{-1}}$) at locations 5 (Antarctic Peninsula) and 6 (Ellsworth Land). The upper bound of the uncertainty for those two locations is determined by the sensitivity tests in which the climate during the spin-up period was colder and drier (spin-up $T_s$, $\dot{b}$ experiment), indicated by the red dotted lines in Figure 9. With this assumption, the trend difference for location 5 greatly reduces and largely disappears for location 6.

## 6  Sensitivity analysis

We investigated the impacts of uncertainties in the model densification rate expression, fresh snow density, climate forcing, and climate forcing during the spin-up period, on the final FAC and surface elevation change results for (i) the 106 sample locations and (ii) expanded to the entire ice sheet (Table 5). For the sample locations, the sensitivities with updated MO fits are lower than with the FDM v1.2A fits (i.e. without updated MO fits), especially the FAC uncertainties, which are on average 41 % lower. The results for the sample locations with updated MO fits are then expanded to the entire ice sheet. Some differences between the ice-sheet-wide and sample locations exist, which can be explained by the sample locations being biased towards high accumulation, which we used to obtain optimal empirical relations for expanding the sample locations uncertainties (Fig. S1). The average simulated FAC, for the sample locations and entire ice sheet (both with updated MO fits), is most sensitive to the uncertainties in the MO fits (5.2 and 5.1 %), temperature forcing (2.3 and 1.8 %) and accumulation forcing (3.7 and 4.0 %), and least sensitive to uncertainties in fresh snow density (0.7 and 1.2 %), and the spin-up forcing (ranging from 0.5 to 0.7 %). The simulated surface elevation change, for the sample locations and the entire ice sheet, is most sensitive to uncertainties in the accumulation during the spin-up (22.4 and $19.0 \, \mathrm{mm \, yr^{-1}}$), followed by the accumulation forcing (7.8 and $5.6 \, \mathrm{mm \, yr^{-1}}$), and temperature during the spin-up forcing (5.2 and $4.4 \, \mathrm{mm \, yr^{-1}}$). As the uncertainties in temperature and accumulation during

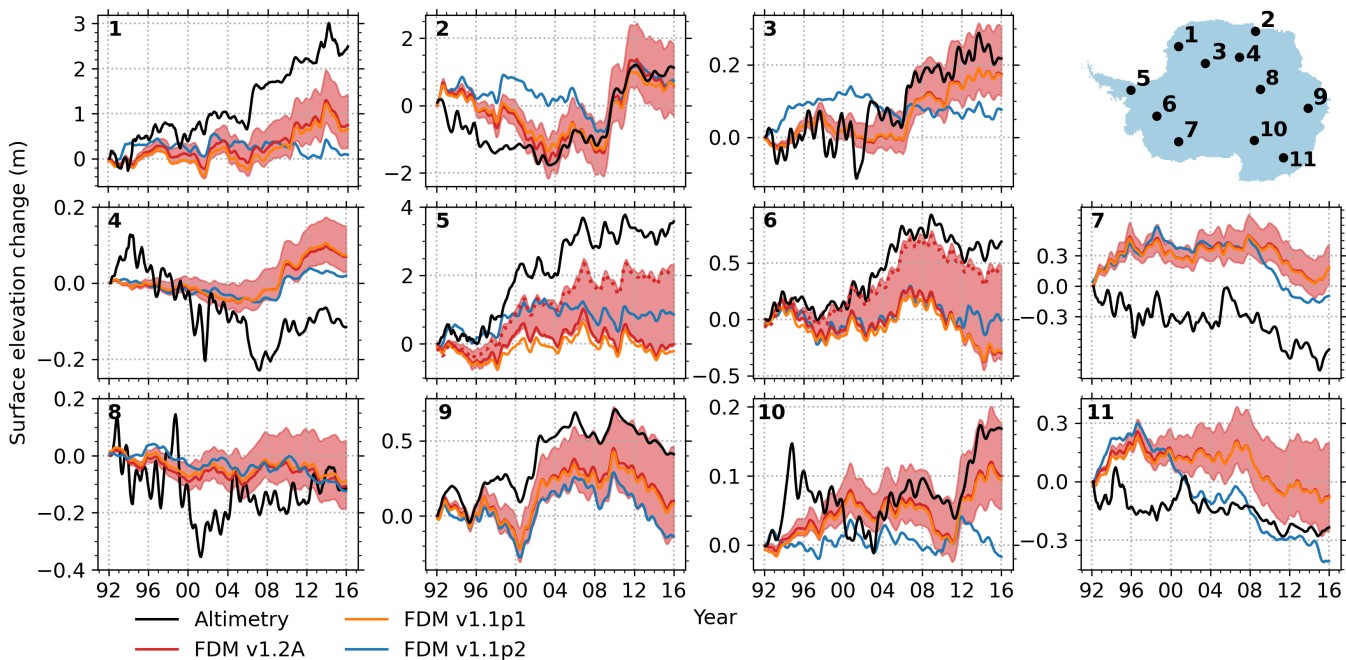

**Figure 9.** Time series of the altimetry observations and modelled elevation change for eleven locations over the period 1992-2015. The time series have been smoothed using a 6-monthly moving average window. The modelled elevation change includes FDM v1.2A, FDM v1.1p1 and FDM v1.1p2. The number of each panel corresponds to the numbered locations on the maps. The red shaded areas indicate the total uncertainty from the sensitivity analysis. The red dotted lines in panels (5) and (6) indicate the surface elevation change from the spin-up $T_s$, $\dot{b}$ experiment.

the spin-up period have opposite effects, the tests in which accumulation and temperature were adjusted simultaneously yield smaller surface elevation trend differences of 17.4 and 14.9 $\mathrm{mm\,yr^{-1}}$. The simulated surface elevation change is least sensitive to uncertainties in the MO fits (0.7 and 0.5 $\mathrm{mm\,yr^{-1}}$), fresh snow density (2.1 and 1.5 $\mathrm{mm\,yr^{-1}}$) and temperature forcing (0.7 $\mathrm{mm\,yr^{-1}}$).

520

The spatial patterns of the total FAC and surface elevation change uncertainties are shown in Figure 10, and for all experiments individually in Figures S4 and S5. The total FAC uncertainty, dominated by the MO fits, accumulation and temperature uncertainties, is high (>10 %) along the high-accumulation coastal margins and in part of the interior, especially near the Transantarctic Mountains. The surface elevation change uncertainty is dominated by the spin-up and accumulation uncertainty,

525 and mainly follows the accumulation pattern, with high values (>25 $\mathrm{mm\,yr^{-1}}$) along the high-accumulation coastal margins, especially in the Antarctic Peninsula, and in the Transantarctic Mountains.

We show time series of the uncertainty in surface elevation change and FAC at the sample locations in Figure S7a for all experiments, and in Figure S7b for the non-spin-up experiments, i.e. the MO fits, fresh snow density, temperature and accumulation experiments. First, we discuss uncertainties in surface elevation change for non-spin-up and spin-up experiments.

**Table 5.** Overview of the relative change in annual average firn air content (FAC) and surface elevation change (dH/dt) of the ice-sheet-wide estimates and sample locations experiments. The sample location experiments also include the experiments without adjusted MO fits. The FAC uncertainty for all experiments and surface elevation change uncertainty for the spin-up experiments are calculated over 1979-2020, and the surface elevation change uncertainty for the MO fits, $\rho_s$, $T_s$ and $\dot{b}$ experiments are calculated over 2015-2020. An overview of the experiments including the prescribed uncertainties are listed in Table 2.

| Experiment name | Ice-sheet-wide estimates | | Sample locations (without adjusted MO fits)[a] | |
| --- | --- | --- | --- | --- |
| | FAC (%) | dH/dt (mm yr$^{-1}$) | FAC (%) | dH/dt (mm yr$^{-1}$) |
| MO fits | 5.1 | 0.5 | 5.2 | 0.7 |
| $\rho_s$ | 1.2 | 1.5 | 0.7 (5.8) | 2.1 (2.4) |
| $T_s$ | 1.8 | 0.7 | 2.3 (4.9) | 0.7 (1.5) |
| $\dot{b}$ | 4.0 | 5.6 | 3.7 (6.8) | 7.8 (8.0) |
| spin-up $T_s$ | 0.5 | 4.4 | 0.5 (1.5) | 5.2 (5.2) |
| spin-up $\dot{b}$ | 0.7 | 19.0 | 0.7 (1.1) | 22.4 (22.4) |
| spin-up $T_s$, $\dot{b}$ | 0.7 | 14.9 | 0.6 (0.8) | 17.4 (17.4) |
| total[b] | 6.9 | 16.0 | 6.8 (11.5) | 19.2 (19.4) |

[a] The values between brackets indicate the uncertainties of the experiments without updated MO fits. [b] For the spin-up experiments only the combined temperature and accumulation experiments are included.

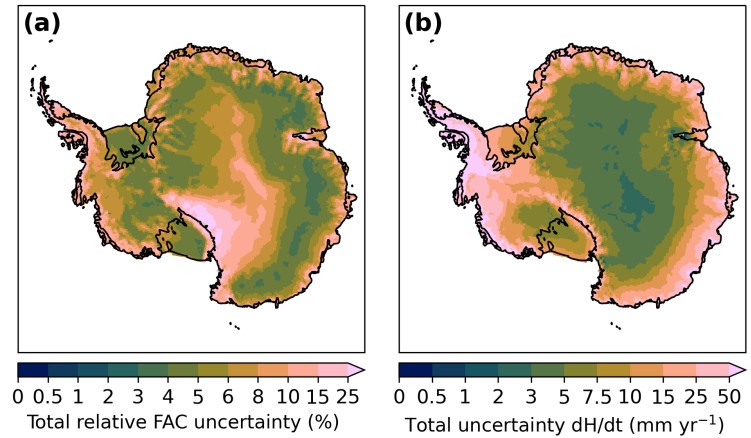

**Figure 10.** Maps from FDM v1.2A of the estimated ice-sheet-wide total uncertainty in **(a)** firn air content (FAC) and **(b)** surface elevation change (dH/dt) from the sensitivity analysis. The FAC uncertainty for all experiments and surface elevation change uncertainty for the spin-up experiments are calculated over 1979-2020, and the surface elevation change uncertainty for the MO fits, $\rho_s$, $T_s$ and b experiments are calculated over 2015-2020.

Next, we look at uncertainties in FAC. The uncertainty of the non-spin-up experiments (Fig. S7b) is only 4 % of the total surface elevation change uncertainty (Fig. S7a). This uncertainty is smaller than the 40 % that can be derived from Table 5.

This is because an imposed uncertainty has opposite effects across the AIS and in time. For example, lowering the fresh snow density increases the rate of a surface elevation increase/decrease. Table 5 lists the absolute uncertainties in surface elevation change over a short 5-year period. However, spatial and temporal variability in sign of elevation change tend to cancel out the uncertainties. In contrast, the uncertainty in the spin-up experiments gives a spatially and temporally uniform response. This enables us to expand ice-sheet-wide estimates of uncertainty of the spin-up experiments into time series (Fig. S7c), by adding a uniform uncertainty for each timestep.

Uncertainty in ice-sheet-wide FAC change is only 19 % of the uncertainty in ice-sheet-wide surface elevation change, similar to the 23 % for the suite of sample locations (Figs. S7a,c). The spatial pattern in FAC change uncertainty for the spin-up experiments is in general similar to the surface elevation change uncertainty, apart from a decreasing uncertainty in high-accumulation regions (>600 $\mathrm{mm\,w.e.\,yr^{-1}}$) (Fig. S6). We attribute this to the fact that in high-accumulation regions, a large part of the upper firn layer does not originate from the spin-up period. Here, the main impact is on deeper high-density firn, meaning that the impact on a volume to mass conversion is limited compared to the large uncertainty in surface elevation change. However, this large uncertainty in the surface elevation change compromises our ability to separate ice-dynamical from SMB and firn-related surface elevation changes. In contrast, in the non-spin-up experiments, the magnitude of the FAC uncertainty is similar (∼85 %) to the uncertainty of surface elevation change.

For the Antarctic Peninsula and Ellsworth Land, accumulation and temperature have increased during the 20th century (Thomas et al., 2017). To evaluate our results from the spin-up sensitivity tests in these two regions, we used surface elevation trends from altimetry observations (Schröder et al., 2019) (Fig. S8). We found that the absolute residual trend of altimetry minus the spin-up sensitivity test was reduced by 25 % compared to the reference FDM v1.2A (from 6.1 to 4.5 $\mathrm{cm\,yr^{-1}}$). When we leave out pixels with a $z_{830}$ age < 42 years (6% of the pixels), where we assume the underlying ice to have responded to the changing climatic conditions, the residual trend was reduced by 38 % (from 5.6 to 3.5 $\mathrm{cm\,yr^{-1}}$). This suggests that the uncertainty for those locations ($z_{830}$ age < 42 years) is overestimated, as $v_{ice}$ can not be assumed constant. Over the remainder of the AIS, some locations experienced an increase in accumulation and temperature whereas others experienced a decrease in accumulation and temperature over the last centuries (Thomas et al., 2017). This has opposite effects on the surface elevation and FAC change, which suggests that the surface elevation and FAC change uncertainties averaged over the AIS are likely lower than the results of the spin-up temperature and accumulation uncertainty experiments (Fig. S7c).

## 7 Implications, remaining limitations, and outlook

In this study, the fresh snow density and firn compaction parameterizations of IMAU-FDM have been improved, and an updated atmospheric forcing has been used over the AIS. The resulting differences in firn changes over time impact e.g. the conversion of volume changes to mass changes in altimetry studies. Over the period 2003-2019, Smith et al. (2020) used volume changes from altimetry to estimate that the total mass change of East Antarctica amounts to 195 $\mathrm{Gt\,yr^{-1}}$, and of West Antarctica to -245 $\mathrm{Gt\,yr^{-1}}$. When FDM v1.2A is used instead of FDM v1.1p1, the mass change over the period 2003-2015 for East Antarctica is 7.3 $\mathrm{Gt\,yr^{-1}}$ lower (187.7 $\mathrm{Gt\,yr^{-1}}$), and for West Antarctica this is 1.2 $\mathrm{Gt\,yr^{-1}}$ higher (-243.8 $\mathrm{Gt\,yr^{-1}}$). These temporal firn

differences are predominantly caused by the updated climatic forcing. In addition, updates in the fresh snow density and firn densification parameterizations have substantial impact on the time averaged firn density profiles. These outputs are required to assess the meltwater buffering capacity of the firn layer, and to convert column thickness to ice mass, to calculate solid ice discharge over the grounding line (Rignot et al., 2019).

The comparison with satellite altimetry measurements are used to asses the performance of FDM. However, especially in the interior, both the long-term trend and seasonal variability are small compared to the error range of the altimetry observations, which explains part of the remaining discrepancy that exists there, but also hampers the comparison (Verjans et al., 2021). Accurate in situ measurements of surface height evolution and firn densification are sparse. Future evaluation of firn models will therefore likely continue to depend on satellite altimetry comparison. The ICESat-2 ATL15 (Smith et al., 2021) is a precise Antarctic-wide 3-month height change laser product that is available from May 2018 onwards, thereby providing an opportunity to assess seasonal and interannual variations, especially in the coming years when more data will become available. Smith et al. (2022) already used ICESat-2 measurements to evaluate regional climate and firn densification models in Greenland.

The sensitivity analysis shows that updating the MO fits reduces the average FAC sensitivity to uncertainties in fresh snow density and climatic forcing by 41 % (Table 5), which implies that the MO factors partly correct for biases in these boundary conditions. Thus, application of FDM v1.2A calibrated MO fits when using a different fresh snow density or climate forcing is not recommended in combination. The fresh snow density parameterization is also dependent on the climate conditions derived from RACMO2.3p2, so any biases will impact the updates of the coefficients. In addition, the simulated fresh snow density accounts for the wind-driven compaction, and the updated coefficients are thus not recommend to use in combination with a firn model that includes wind-driven densification of the upper firn layers.

Another limitation, which is evident from both the remote sensing comparison and the sensitivity analysis, is the assumption of a steady-state firn layer over the period with reliable climate forcing (1979-present). In our sensitivity analysis we show that correcting for this by estimating previous climatic conditions from ice core data, reduces the difference between simulated and altimetry observed surface elevation trends by 38 % over the Antarctic Peninsula and Ellsworth Land. Future work could focus on further accounting for pre-1979 climatic conditions.

The densification scheme used in FDM v1.2A (Eq. 3) is developed for dry snow densification. As it is used for both wet and dry locations, we assume that the densification rate of dry firn is equal to that of wet firn. Since the presence of liquid water impacts the evolution of grain size and shape, this may also impact the densification rate of firn, however due to a lack of physical understanding and available measurements, this has not been included in FDM v1.2A. The RMSE of the modelled $z_{550}$, $z_{830}$ and FAC for the wet firn cores are respectively, 3.17, 14.34 and 4.40 m, and thus larger than the RMSEs of the entire firn core dataset (2.33, 8.33 and 2.74 m, respectively). Uncertainties in the melt forcing (Carter et al., 2022) impact the simulated density profiles, which may contribute to the lower agreement and hampers this comparison.

## 8 Summary and conclusions

In this study, we present and evaluate a simulation of the contemporary characteristics of the Antarctic firn layer using the updated IMAU-FDM (Firn Densification Model), version v1.2A, for the period 1979-2020. In this new version, the fresh snow density and firn compaction parameterizations have been improved, and an updated atmospheric forcing has been used. The RMSE when compared to observations of the fresh snow density and firn thickness have been reduced by 28 and 39 %, respectively, compared to the previous parameterizations. The RMSE of FAC is 18 % lower compared to the previous model version FDM v1.1p1. In general, the modelled fresh snow densities have decreased, especially in high-accumulation areas and to a lesser extent in windy escarpment regions. Firn thickness and FAC have decreased in low- and high-accumulation areas, whereas they have increased in intermediate accumulation areas.

In accordance with observations, the firn thickness and density patterns vary spatially across climatic regions with temperature, accumulation and surface melt as drivers. Along the coastal margins, we found that the firn thickness and FAC vary strongly in time, whereas in the interior the variations are small. The temporal variations can be split into a rather stable seasonal cycle driven by snowfall, compaction, summer melt and sublimation, and more irregular decadal variations mainly driven by slow snowfall variations. Variations in FAC and firn thickness have a similar phase, and 62 % of the seasonal and 67 % of decadal surface height variability are due to variations in FAC rather than firn mass, which emphasizes the importance of correcting for seasonal and decadal FAC variations in altimetry studies.

Comparison of simulated surface elevation change with altimetry confirms that the performance of the updated model has improved (a 17 % reduction in error), notably in Dronning Maud Land and Wilkes Land, which is predominantly due to updated climate forcing. However, a substantial trend difference ($>10 \, \mathrm{cm \, yr^{-1}}$) remains in the Antarctic Peninsula and Ellsworth Land, which is likely caused by increasing accumulation over the past centuries, violating the model assumption of a steady state over the simulated period (1979-2020). Correcting for this over the Antarctic Peninsula and Ellsworth Land, by estimating previous climatic conditions from ice core data, substantially improves this (an error reduction of 38 %). In general, uncertainties in the model formulation, climate forcing or climate forcing during the spin-up cause rather small changes in simulated FAC (<5.2 %), while uncertainties in the accumulation forcing, and in the accumulation and temperature forcing during the spin-up period can lead to substantial differences in simulated surface elevation change (4 to 19 $\mathrm{mm \, yr^{-1}}$).

*Code availability.* The code of IMAU-FDM v1.2A is available on GitHub at https://github.com/brils001/IMAU-FDM (last access: 23 November 2022).

*Data availability.* A list of all firn cores used and corresponding references can be found in the Table S1 in the Supplement. IMAU-FDM data can be obtained from the authors without conditions.

*Author contributions.* SV, WJvdB, MB, PKM and MRvdB defined the research goals and designed the study. SV improved the model, performed the simulations and analyzed the results. All authors contributed to discussions on the manuscript.

*Competing interests.* MRvdB is a member of the editorial board of journal The Cryosphere

*Acknowledgements.* This study is part of the HiRISE project which is funded by the Netherlands Organization for Scientific Research (grant no. OCENW.GROOT.2019.091). MRvdB acknowledges support from the Netherlands Earth System Science Centre (NESCC). We acknowledge ECMWF for computational time on their supercomputers.

625

630

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
