# Peer review of "Characteristics of the 1979-2020 Antarctic firn layer simulated with IMAU-FDM v1.2A"

_The Cryosphere, 2022_

## Referee Comment (RC1)

Review of
*Characteristics of the contemporary Antarctic firn layer simulated with IMAU-FDM v1.2A (1979-2020)*
by Veldhuisen et al.

Reviewer: Vincent Verjans

This study presents a modeling approach to simulate the evolution of the Antarctic firn layer. It presents the re-calibration of a firn model (IMAU-FDM), with forcing from an updated regional climate model (RACMO2.3p2). The calibration mostly follows the same approach as in the initial model release but with a more extensive dataset of firn cores and surface snow density measurements. Based on a 1979-2020 model simulation, the study presents an analysis of spatio-temporal features in firn thickness, firn air content (FAC) and surface elevation change over the entire Antarctic ice sheet. I welcome this contribution to firn model improvement. The modeling capabilities and the full coupling between climate and firn models are two remarkable aspects of this study. The model pair IMAU-FDM – RACMO is often used for evaluating the firn height change component in altimetry studies, and it is thus paramount to accurate estimates of ice sheet mass balance change by the glaciological community. But this study often gives the feeling to be a simple update of the work of Ligtenberg et al. (2011). For this reason, I find this study slightly too superficial, and several points can be evaluated and analyzed in more details. Nevertheless, I believe that, by building upon the work already achieved for this first version of the manuscript, a future revised and updated version will be a good contribution to *The Cryosphere*, and to firn model development in general.

This review is separated in Major, Minor, and Specific comments. My Major comments highlight current weaknesses of the studies that require a change in the methods or more in-depth work. My Minor comments require more clarity, small modifications, and/or strong justifications from the authors. The Specific comments are remarks concerning specific statements in the manuscript, and mostly relate to the structure of the text.

Major comments

1) Uncertainty quantification
As mentioned in the introduction, IMAU-FDM is commonly used in ice sheet mass balance assessments. In addition to this, the firn component of elevation changes is often cited as a major component of uncertainty in altimetry-based mass balance assessments. Combining these two aspects together, it is obvious that our community needs better uncertainty estimates from firn models themselves. The uncertainty quantification in this study is, in my view, not sufficient. The authors present many components of uncertainty in Section 2.6, as well as ways to estimate their contribution to IMAU-FDM output uncertainty. However, it is not clear how these points highlighted for a sensitivity analysis are subsequently used for uncertainty quantification in the results.

1a) Results of the sensitivity analysis
Many of the possible experiments presented in Section 2.6 are not even mentioned in the results section about uncertainty (Section 6). In Section 2.6, the authors mention various scenarios of accumulation and temperature reduction for the spin-up period (three for each variable). However, they only discuss a single test per variable in Section 6, and it is not specified which one it is. This obviously requires more clarity.
Similarly, the sensitivity tests using "*the 95 % confidence intervals of the MO fits*", and the "*uncertainty of the fresh snow density*" are not discussed. I strongly recommend to clarify the links between Sections 2.6 and 6. For example, each sensitivity test should be given a name. And there should be a Table that specifies the sensitivity tests, their corresponding variable adjustments, and results.
The Table 4 needs some further adjustments. I suppose that the FAC values in Table 4 are calculated only over the dataset of firn cores, and not over the entire ice sheet, which should be specified in the caption. Table 4 should also include the results from the spin-up perturbation experiments: 3 per variable (see Section 2.6), and the experiments combining temperature and accumulation perturbations (not explained in Section 2.6).
Finally, the sensitivity experiments use ice sheet wide averages of precipitation spread (+/- 8 %) and of temperature uncertainty (+/- 1.5 K). However, local uncertainties can differ strongly from the ice sheet averages. Where possible, I recommend using spatially variable uncertainties in the sensitivity analysis.

1b) Ice-sheet wide method for uncertainty quantification
The authors have investigated the results of the sensitivity experiments only at the firn core locations. However, this is of little interest to the community. A method to compute uncertainty estimates over the entire ice sheet in the different components of Eq. (6), and thus ultimately on dh/dt, should be developed. A straightforward approach could be to sample perturbations in all uncertainty sources mentioned in Section 2.6, compute simulations at key locations, and regress uncertainty estimates against climatic variables. However, I leave the choice to the authors of how to best estimate uncertainty from their model, and I note that they have already worked on similar issues (e.g., Kuipers Munneke et al., 2015). The outcome of a more thorough uncertainty analysis should be:
- uncertainty bands in time series of Figure 7 and Figure 9
- maps of total uncertainty in modeled surface elevation change (and possibly of the uncertainty components)
- a quantification and discussion of the different components to uncertainty in dh/dt across Antarctica

2) Statistical procedure
In statistical calibration of parameters such as the MOs, it is well-known that validation should be, to some degree, independent of calibration. Here, the authors use the same dataset to calibrate their new parameters (MOs and surface density) as to evaluate their calibration. As such, the validation is not meaningful. I realize that firn data is sparse, and it can be argued that excluding a part of the dataset from calibration might be detrimental to model fitting. But that does not exclude a form of k-fold cross validation to better evaluate the fits. Furthermore, this approach would provide more robust uncertainty estimates on the MO and surface density values.
For the evaluation, the authors use only 10 additional measurements in the evaluation dataset, in addition to the ones used in the calibration dataset. This modest introduction of independent data in the evaluation results in (line 243): "*a slightly deteriorated correlation for the $MO_{550}$.*" when compared to the FDMv1.1p1 parameterization. Furthermore, in Section 6, the authors demonstrate that the calibration of the MOs is sensitive to realistic uncertainty in climate forcing, and that this sensitivity strongly impacts results in FAC, $z_{550}$, and $z_{830}$ (Table 4). These two aspects thus show evidence that the statistical fits performed in this study are probably sensitive to noisy features inherent to the observations used in the calibration.
In Table 3, I recommend providing the RMSE values in all the rows, to give the bias values in addition to the RMSEs, and to compute these fit statistics also with respect to the FAC values of the dataset.
Finally, if I understand the process correctly, the comparison of fits between FDMv1.1p1 and FDM v1.2A for z550, and z830 is unfair. FDMv1.1p1 was calibrated with climate output of a previous RACMO version and another surface density parameterization. As such, it is obvious that it performs worse than FDM v1.2A when it is used with RACMO2.3p2 forcing and another equation for surface density. Furthermore, it is unclear how much of the improved fit to observations is due to the re-calibration of IMAU-FDM versus the updated climatic forcing and the updated surface density parameterization. All these aspects should be addressed in the manuscript, and identified as caveats in the comparison. The same holds when comparing FDMv1.1p1 and FDM v1.2A to the altimetry product (Figure 8): it remains unknown what part of the improvement is due to changes in the densification equation, changes in the climatic forcing, and changes in the surface density parameterization. As pointed out by the authors, Figure 9 suggests that most of the improvements are due to the update of RACMO, which raises questions concerning the improved performance of FDM v1.2A in simulating firn processes. This should be discussed more in depth.

3) Neglect of melt areas
The MO parameterization is constrained only in dry firn areas, but the model is used in wet firn areas also. One can reasonably expect errors to be much larger in modeling wet firn densification. While this topic is not the focus of the study, any ice-sheet wide study of firn evolution should at least discuss this limitation. Ideally, I would encourage the authors to evaluate IMAU-FDM in the wet firn areas also, by comparing modeled FAC to observed FAC from firn cores. They could also provide uncertainty estimates that are valid for melt areas.

Minor comments

1) There is a general lack of quantification. I encourage the authors to identify all the uses of words such as "reasonably", "somewhat", "roughly", "improved", and "substantially". These should be complemented by quantitative values.

2) In the densification equation (Eq. (3)), it seems to me that the mean long-term accumulation rate is used. An alternative approach is to use the mean accumulation rate over the lifetime of each specific firn layer, which is more representative of the effect of overburden stress (Li and Zwally, 2011). This aspect could be important given that decadal snowfall variability can be large on the Antarctic ice sheet. And there is no physical reason for the densification of a firn layer to be a function of past accumulation rates. Why did the authors choose to use the mean long-term accumulation rate?

3) Lack of clarity about the firn core dataset
Depsite re-reading several time Section 2.5, it is still unclear to me how the authors selected their dataset.
- "*We used 125 density profiles from firn cores and 8 density profiles from neutron density probe measurements*": but the wet firn cores were discarded for the calibration, so how were they used in this work?
- "*For the MO fits, 104 dry firn cores could be used (…) To evaluate the firn density profiles from the simulation using the derived MO fits, 122 firn cores could be used.*": what explains this difference of 18 cores?
- in Section 2.6 "*105 observational locations shown in Figure 1*": 105 is not even mentioned in Section 2.5.
- in caption of Figure 1 "*The grey circles indicate ten additional locations that were included in the sensitivity analysis*": give more details about these 10 additional firn cores in Section 2.5.
- Concerning access to the firn core data, please see the *Data policy* section of *The Cryosphere*: "*Authors are required to provide a statement on how their underlying research data can be accessed. This must be placed as the section "Data availability" at the end of the manuscript.*"

4) Interpretation of FAC change versus mass change
I believe that there is a confusion when interpreting the role of FAC change for conversion of elevation changes to mass changes. For example, on line 328: "*63 to 68 % of the seasonal surface elevations fluctuations are caused by a change in air content rather than actual mass change*". However, FAC changes are principally caused by changes in snowfall, which implies a corresponding mass change. Thus, there is an underlying mass-related component to fluctuations in elevation caused by FAC changes.

5) I find that there is a general lack of explanation about some model results. I give some examples here below.
- In Figure 7, FAC variability is smaller than snowfall variability. What explains the dampened response of FAC?
- Figure 6d shows that there is a lot of spatial variability in the phase of firn height. This is an interesting result, and some explanation should be provided for why such patterns appear.
- The seasonal amplitude comparison with altimetry (Section 5.1) is averaged across the ice sheet. Are there no interesting patterns that appear at finer scales? Furthermore, the authors state that (line 374): "*The performance of the new model thus appears to represent an improvement*" when comparing FDMv1.1p1 and FDM v1.2A. But because they only analyze the amplitude averaged over the entire ice sheet, the better performance could be due to error compensations in different areas. A finer-scaled analysis is therefore required.
- The evaluation against the surface altimetry data is not sufficient. There should be more focus on regional patterns and on relations between discrepancies and climate among other things. And especially, the comparison should be much more quantified (see Minor comment 1).
- On line 323: "*This agrees with the modelled firn thickness amplitude of 3.5 cm by Medley et al. (2020).*". This is despite the lower accumulation rates than in the study of Medley et al. (2020). Thus, how is that compensated? Is IMAU-FDM more sensitive to seasonal temperature variability?

Specific comments

-l4
"*improved*": specify that this refers to a previous model version.
-l5
"*observations*": change to "firn core observations".
-l15
The reference cited shows that firn thickness can exceed 100 m.
-l17 – 19
"*Firstly, firn depth and density estimates are required to convert altimetry observed volume-to-mass changes, which remains a major source of uncertainty in mass balance studies*": please also reference the work of Morris and Wingham (2015).
-l27 – 29
Please reference the work of Arthern and Wingham (1998).
-l27
"*are used as measures of its dynamics and mass balance*": clarify and rephrase.
-l30
Change "*in a mass and density component*" to "in a mass- and density-change component".
-l32 – l37
Is this paragraph only about ice shelves. If yes, this should be specified in the first sentence. If not, the reference to ice shelf hydrofracturing is confusing.
-l33
I do not believe that there is evidence of "*reduced accumulation*" for a warmer future climate in Antarctica.
-l33
Use "potentially lead to".
-l40
Please reference the work of Medley et al. (2015).
-l44
"*Firn models can roughly be divided in two classes: physically based and semi-empirical models.*": please reference the work of Morris and Wingham (2011).
-l46
It is not clear to me what "*which*" refers to.
-l47
Change "*less poorly known parameters*" to "a smaller number of poorly constrained parameters".
-l49
Please reference the work of
-l56
Start the sentence as: "This study shows that, at the basin scale, (…)".
-l58
Add comma: "climatic conditions, firn densification".
-l63
Specify the sort of "*field measurements*".
-Table 1
In the column "*Other*", please refer to Equation numbers.
-l75
"*FDM V1.2G*" has not been defined yet. Maybe, refer earlier to Table 1.
-l77
"*comparable*": with respect to what metric?
-Section 2.1.1 and in the remainder of the manuscript
Please do not use the same symbols for different parameters. If the authors want to preserve a connection between related parameters, I recommend the use of subscripts.
-l83

Change "*average*" to "averages of".

-l86

Cite a reference for "*Snow crystal size and therefore fresh snow density indeed increase with increasing temperature.*".

-l94

Specify the frequency of "*instantaneous*": hourly/daily/…

-Equation 2

I believe that there should be no overbar on $T_s$ and $V_{10}$ in this equation.

-l101

"*defined as the top 0.5 m*": am I correct that the authors calibrate the density of the modeled upper 0.5 m to the density of the observed upper 0.5 m, but that the calibrated surface density is then used only for the top layer of IMAU-FDM? If so, please clarify the approach as well as the slight discrepancy between calibration and usage of the surface density parameterization.

-l102

Specify the model time step here or elsewhere.

-Section 2.1.2

All equations should be specified as two different cases for rho<550 and rho>550 kg m$^{-3}$.

-l110

I suggest replacing "*processes*" by "mechanisms".

-l111 - 112

A citation is needed for this sentence.

-l114

Change "*turn out to depend on the accumulation rate*" to "are chosen as functions of the long-term mean accumulation rate".

-l120

Why was the power-law function not tested for $MO_{550}$?

-l124

Provide formulation of the thermal conductivity.

-l128

Provide formulation of the irreducible water content.

-l129

The description of the refreezing algorithm is unclear and confusing. It suggests that no meltwater refreezes if a layer cannot accommodate all the incoming meltwater.

-l131

Typo: remove "*the*".

-l131

Specify the amount of melt.

-l131

Do not use "*significant*" because it does not refer to statistical significance here.

-l134 – 135

What is the depth of the model domain? And what are the boundary conditions at the lower boundary?

-l137

I believe that "*total thickness*" should be replaced by "total mass".

-l142

Please quantify the "*minor trend*".

-l142

Change "*ice*" to "material with ρ>830 kg m$^{-3}$".

-Equation 6

Format of the variables in the equation does not correspond to format of the variables in the main text.

-l152

SMB units are wrong.

-l160

Change ";" to "and".

-l161

Explain briefly the notion of "*upper-air relaxation*".

-l168

Split this sentence in two: "This results in an improved forcing. For example, (…)".

-l169

Remove "*e.g.*".

-l178

Typo: "describes".

-Figure 1

As I understand it, all the firn cores are used for the sensitivity analysis. For this reason, I recommend changing the label for the grey dots in the legend to "Sensitivity analysis only".

-Figure 1 caption

Replace "*on top of*" by "in addition to".

-Section 2.6

Provide a name for each sensitivity experiment (see Major comment 1).

-l197

"*To improve the representation*": I do not see the causal link between improvement and the rest of the sentence.

-l203 – 204

Change "+-" to +/-".

-l204

Does +/- correspond to the RACMO2.3p2 RMSE? If so, please specify this.

-l206

Specify that 30 kg m$^{-3}$ corresponds to the RMSE in surface density.

-l206

"*Section 3.2*" should be "Section 3.1".

-Table 2

There is no "*D*" parameter in Eq. (2) for FDM FS-L and FDM v1.2A.

-l218 – 219

Refer to Equation numbers.

-l231

Change "*reduced with*" to "reduced by".

-l240

"*a simulation of FDM v1.2A without MO fits*": does that mean a simulation with the FDMv1.1 MO values?

-l241

"less steep": specify that this is with respect to accumulation (bdot).

-l253

Replace "*approaches zero*" by "decreases asymptotically towards zero".

-l255

Provide references for the accumulation rate values.

-Table 3

In the column "*Version*", I believe that $MO_{550}$ and $MO_{830}$ should be replaced by $z_{550}$ and $z_{830*}$. In the column "*Fit*", please refer to Equation numbers. Add a Bias column (see Major comment 2). Provide RMSE values for all models (see Major comment 2). Add FAC rows (see Major comment 2).

-l258

Specify "27 km horizontal resolution".

-l259

Typo: "in Figure 4".

-l261

What does "*calm*" mean?

-l262

Change "*On top of this*" to "In addition to this".

-l264 – 265

*"The spatial pattern of the depths of the critical density levels z550 and z830 are shown in Figures 4b and c, and are roughly the inversed pattern of the surface snow density"*: not in melt areas, please discuss (see Major comment 3).

-l266 – 267

*"The patterns vary spatially across climatic regions with temperature as a primary driver
and accumulation as a secondary driver."*: please discuss the impact of surface melt (see Major comment 3).

-l270 – 271

Please rephrase this sentence with more formal language.

-Figure 3

I believe that there is a mismatch between the numbers of points shown in the subplots, and the number of cores mentioned in Section 2.5. For example, it seems to me that there are less than 100 data points shown in Figure 3a. Also, please show scatter plots of the match between modeled and observed $z_{830*}$, as well as between modeled and observed FAC.

-l281 – 282

Change *"can be used"* to "must be used".

-Figure 4 caption

Change *"firn age of the crtical density level"* to " firn age at the crtical density level".

-Figure 5

Define *"peak-to-peak"* in the caption. Please also include a map of average seasonal amplitude.

-l286

Change *"most"* to "more".

-l286

Typo: "parts of the coast".

-l292

If statistical significance has not been tested for, please do not use *"significant"*.

-l294

Replace *"closer to the mean"* by "less spatially variable".

-l303

Please remind the reader about the study period.

-l303 – 304

*"Large values indicate that seasonal and interannual climate variability cause large temporal variations in FAC."*: this statement is not supported by the map of peak-to-peak variability, which depends only on two single values in the entire time series. In my view, peak-to-peak does not characterizes temporal variability well.

-l313

$V_{tot}$ does not appear in Equation 6. Maybe simply replace dh/dt by $V_{tot}$ in Equation 6.

-l320

Specify that dh/dt in summer months has contributions from sublimation and melt.

-Figure 6 caption

Change *"indicate the standard deviations"* to "indicate the inter-annual standard deviations"

-l329

Change *"biases"* to "errors".

-Section 4.4

In general, this section requires much more quantitative assessments (see Minor comment 1).

-l340

"cumulative surface temperature anomaly": is this the cumulative anomaly in surface temperatures from the long-term mean? Please clarify.

-l341

*"the seasonal firn thickness and FAC variability is driven by"*: please discuss why there is no one-to-one correspondence between firn thickness and FAC variability.

-l341

Typo: *"is"* should be "are".

-l343 – 344

"*firn densification, despite the long time scale, reduces these snowfall-induced fluctuations by about 15 %*": please clarify how this is calculated.

-l356

Specify: "captures the strong spatial variation in firn thickness and density observed in our firn core dataset.".

-l363 – 364

Specify: "shown in grey in Figs. 3d and 3e".

-Section 5.1

Please provide maps of seasonal amplitude and of discrepancy in seasonal amplitude.

-l368 – 369

Provide references to support that lower seasonal variability in IMAU-FDM can be explained by altimetry errors.

-Figure 7

I find the color codes in this figure confusing. I suggest to show FAC in a color other than blue.

-Figure 7 caption

Change the caption to: "Time series from FDM v1.2A of FAC, of the cumulative anomalies of surface temperature, of the vertical firn surface velocity, and of the separate components of the vertical velocity from Eq. (6). Time series are shown for (a) the entire ice sheet, (b) the part of the ice sheet situated above 2,000 m a.s.l. and (c) the part of the ice sheet situated below 2,000 m a.s.l.".

-Figure 8

Remove ice shelves from the maps, or show them in a separate color.

-Figure 8 caption

I suggest changing "*Maps of average surface elevation change*" to "Maps of trends in surface elevation".

-l384

"*which yields the improvement of FDM v1.2A compared to FDM v1.1p1*": I do not understand why the authors call this an "*improvement*". As I understand it, the residual of FDM v1.2A is calculated as altimetry minus FDM v1.2A. The residual of FDM v1.1p1 is calculated as altimetry minus FDM v1.1p1. Thus, subtracting the residuals results in FDMv1.2A minus FDMv1.1p1. In other words, it is simply the difference between both models because the altimetry term cancels out in this operation. If I misunderstand something, please clarify. Otherwise, please revise the use of "*improvement*" when referring to the difference between the residuals throughout the manuscript.

-l389

Typo: "*trends*" should be "trend".

-l394 – 395

Please provide a quantitative justification for why the 11 glaciers chose are "*representative locations*".

-l401

"*(+9 % sd)*": is that compared to post-2003 altimetry?

-l401

"*likely related to the measurement imprecision*": please provide a reference.

-l402

"*(+13 % sd)*": is that compared to FDM v1.2A?

-l402

"*the altimetry variability remains higher than the simulated variability (+13 % sd)*": please discuss possible reasons.

-Figure 9

Why do the authors use a 6-months running average? This masks out all the seasonality. I recommend using a shorter averaging window, 3 months for example.

-Figure 9 caption

Change "*altimetry observed*" to "altimetry observations".

-Section 6

This entire section should be thoroughly reworked (see Major comment 1).

-Table 4

This table should be thoroughly reworked (see Major comment 1).

-l430

"*that our results are robust*": what do the authors mean here? In contrast, I understand from the results that the MO fits are not robust to realistic uncertainties in climate forcing, and that this sensitivity induces strong discrepancies (Table 4) in FAC estimates (see Major comment 2).

-l430 – 432

Repetition: "*A difficulty of the data to model comparison is that (… ) but also hampers the comparison.*".

-l434

Typo: "*dependent*" should be "to depend".

-l440 – 441

Repetition: "*in these regions (…) in these regions*".

-l438 – 440

Please reference the work of Medley and Thomas (2019).

-Section Conclusions

Make sure to be consistent in using past or present tense.

-l446 – 448

This statement of improvement is misleading, because FDMv1.1p is forced in this study with climatic forcing that was different than the climatic forcing used for its calibration (see Major comment 2).

-l450

Typo: "*it has*" should be "they have".

-l451 – 452

"*the firn thickness and density patterns vary spatially across climatic regions with with temperature as a primary and accumulation as a secondary driver*": Here, I believe that more nuance is needed. Spatial variability in firn thickness is primarily dictated by accumulation rate patterns. Also, the sensitivity tests have shown that the trend in surface elevation change is more sensitive to accumulation uncertainty than temperature uncertainty.

-l455

"*As variations in firn air content and firn thickness align*": the meaning of this statement is not clear to me, please clarify.

-l463 – 464

"*our model in general is robust*": again, what do the authors mean here?

References used in this review

Arthern, R. J. and Wingham, D. J.: The Natural Fluctuations of Firn Densification and Their Effect on the Geodetic Determination of Ice Sheet Mass Balance, Clim. Change, 40, 605–624, https://doi.org/10.1023/A:1005320713306, 1998.

Kuipers Munneke, P., Ligtenberg, S. R. M., Noël, B. P. Y., Howat, I. M., Box, J. E., Mosley-Thompson, E., McConnell, J. R., Steffen, K., Harper, J. T., Das, S. B., and van den Broeke, M. R.: Elevation change of the Greenland Ice Sheet due to surface mass balance and firn processes, 1960–2014, The Cryosphere, 9, 2009–2025, https://doi.org/10.5194/tc-9-2009-2015, 2015.

Li, J. and Zwally, H. J.: Modeling of firn compaction for estimating ice-sheet mass change from observed ice-sheet elevation change, Ann. Glaciol., 52, 1–7, https://doi.org/10.3189/172756411799096321, 2011.

Ligtenberg, S. R. M., Helsen, M. M., and van den Broeke, M. R.: An improved semi-empirical model for the densification of Antarctic firn, The Cryosphere, 5, 809–819, https://doi.org/10.5194/tc-5-809-2011, 2011.

Medley, B., Ligtenberg, S. R. M., Joughin, I., Van Den Broeke, M. R., Gogineni, S., and Nowicki, S.: Antarctic firn compaction rates from repeat-track airborne radar data: I. Methods, Ann. Glaciol., 56, 155–166, https://doi.org/10.3189/2015AoG70A203, 2015.

Medley, B. and Thomas, E. R.: Increased snowfall over the Antarctic Ice Sheet mitigated twentieth-century sea-level rise, Nat. Clim. Change, 9, 34–39, https://doi.org/10.1038/s41558-018-0356-x, 2019.

Morris, E. M. and Wingham, D. J.: The effect of fluctuation in surface density, accumulation and compaction on elevation rates along the EGIG line, central Greenland, J. Glaciol., 57, 416–430, 2011.

Morris, E. M. and Wingham, D. J.: Uncertainty in mass-balance trends derived from altimetry: a case study along the EGIG line, central Greenland, J. Glaciol., 61, 345–356, 2015.

---

## Referee Comment (RC3)

Review of:
"Characteristics of the contemporary Antarctic firn layer simulated with IMAU-FDM v1.2A (1979-2020)"
Sanne B.M. Veldhuijsen, Willem Jan van de Berg, Max Brils, Peter Kuipers Munneke, and Michiel R. van den Broeke

Reviewer: C. Max Stevens

**Summary:**

This paper presents an updated version of the IMAU firn densification model (FDM) for Antarctica. The IMAU FDM is commonly used by other research groups that need firn model outputs. The changes to the model include an updated formulation for the densification rate, an improved surface density equation, and a different parameterization for the thermal conductivity. The new formulation is tuned using a better RCM forcing than the previous version, and it also leverage more firn core data for tuning. The authors show that in general, the updated model formulation's outputs match surface elevation change measurements better than the previous version. The paper includes an analysis of the sub-annual and interannual variability in surface elevation and its causes. The authors also performed a range of tests to understand the model's sensitivity to input and parameter uncertainties.

In general, I found this paper to be scientifically insightful and well written. Its analysis of surface height variability is timely given recent advances in satellite remote sensing. I think that the paper will be a good addition to *The Cryosphere*. Below I list general and line-by-line comments that should addressed prior to publication.

**General comments:**

1. The sections (2.6, 6) describing the sensitivity tests need to be clearer – I am still not sure exactly what you did for them. For example, what exactly are you adding and subtracting for those runs? Are you, e.g., adding 8% to the accumulation over the entire forcing, and that is one of the sensitivity runs? It would help to clarify if you wrote exactly what you did to perform these tests, e.g. "We added 8% to the accumulation forcing, and then re-ran our model calibration procedure to get the optimal MO fits (or whatever) with that forcing. We then ran the model with these MO fits and compared the outputs to our baseline model run". (Or something along those lines). I think what I just wrote is the gist of what you are doing, but this is what I need clarified. The distance between sections 2.6 and 6 made this analysis more difficult to understand: I was looking for a table describing the sensitivity tests when I read section 2.6, but did not find it until the end (Table 4). I would also appreciate if table 4 was more specific: e.g., I can figure out what Accumulation+ is from looking at section 6, but it would be nice to have all the information in the table.

   Are you only doing these sensitivity runs for the observational sites, or did you run the model for the entire ice sheet in any of the sensitivity runs? When you say "10 additional locations" (line 198), what are those in addition to? Are those 10 sites special, or were they not used in the original calibration (if so, why not)?

Finally, when doing these sensitivity tests, if you are adjusting the MO fits (section 6), with an empirical firn model, shouldn't you expect the model outputs to match the data reasonably well? This is indeed what you found, but my point is that if you are tuning a firn model using e.g. biased accumulation fields, the firn model will effectively act as a filter for that bias (or, the correction for it will be built into the model), so that the model output will match the data well. I think the upshot of my question is that I would like to see a bit more discussion contextualizing the meaning of the model sensitivity. Is the new model an improved representation of the physics compared to the previous IMAU FDM, or is it just a numerical response based on using different boundary conditions (RACMO forcing and surface density)?

2. In Figure 9 and 5.2: It seems to me that you have taken a rather qualitative approach here in looking at the elevation changes (e.g., "shows comparable patterns", "agreement seems"). I am curious how you have chosen to plot these time series: what is your zero/reference point? To me it would make most sense to plot them all zeroed at the start of the time series (1992) to see a direct comparison of how the data and model differ. The way it is plotted presently is misleading in a few cases: e.g. for site 6 (West Antarctica), the curves appear to be lined up reasonably well, but upon closer examination the data show a clear positive elevation trend while the model predicts a net decrease in elevation. This would enable a more quantitative analysis, such as fitting a trend line to each time series to get the long-term trends. Then, you could de-trend and do a time-series correlation for each site to quantify how well the models predict the shorter time-scale variability.

3. Discussion about implications: My general feeling in the paper what that it was light on discussing the implications of the research in the broader glaciological community. Given how much IMAU-FDM has been used for altimetry studies, I think adding a paragraph or a few sentences describing how the model changes affect our understanding of how the AIS is changing. Do the conclusions drawn by the users of IMAU-FDMv1.1p1 need to be updated?

**Line by line comments:**
44: Can give a sentence with a broad overview of what the 2 types are for readers not familiar with the difference?

47: "less": change to "fewer"

Table1/section 2.1.1 leaves me wondering what surface density was used for run FDM v1.2A. Perhaps change Table 1 to include a 'Fresh Snow Density' column?

Section 2.1.2: Do you use the instant accumulation rate or 'mean accumulation over the lifetime of the firn layer'? If the former, please provide detail as to why you choose this and deal with the fact that densification will be zero if there is a timestep with zero accumulation.

106/Equation (3): the Arrhenius factor is missing the $e$. Also (and this is a bit pedantic), $\dot{b}$ in Arthern et al. (2010) has units of kg m$^{-2}$ a$^{-1}$, which then gives a densification rate $\frac{d\rho}{dt}$ with units of kg m$^{-3}$ a$^{-1}$. (Arthern defines the units for the factor $D$ in their Appendix B). Your accumulation units (mm w.e a$^{-1}$) are numerically equivalent, but with them the densification rate in your Equation 3 does not end up with units of density per time.

114-122: It is not entirely clear what you mean with MO$_{550}$ and MO$_{830}$: is MO$_{550}$ the value that should be used for $\rho < 550$, and MO$_{830}$ for $550 < \rho < 830$? Also, perhaps specify that MO added as a multiplier to equation (3).

122: Add a sentence at the end of this section explaining that the parameters are tuned, and this tuning is described in Section 3.

Section 2.2: Do you set the bottom of your model domain to be the depth of the 830 horizon, or does it go deeper? In some locations in Antarctica, there is enough of a temperature gradient at the bottom of the firn to affect the temperature – if you are not modeling through the depth of the ice sheet, do you account for this heat flux?

142: not clear – why is the air content increasing?

160: change semicolon between references to "and"

181: This is confusing – at the beginning of the paragraph you state that you use 125+8 density profiles, but then here you say you used 122. I think that a bit of language clarification in this paragraph will help – something along the lines of, 'We gathered data from 125 density profiles from firn cores and 8 density profiles from neutron…'. Then, '104 of our 133 cores fit the dry-snow criteria needed for our MO-fitting routine.' Or something along those lines adding a bit more specificity.

182: This sentence has the word 'density' 3 times – can you rewrite it to make it a bit easier to read?

184: In some locations, this 0.5 m is likely snow (i.e. less than a year old), but in the interior it is several years of accumulation. Is this a concern, either for tuning or for interpreting model outputs?

203/205 (elsewhere too?): fix the +- to +/- (may be done in manuscript typesetting?)

219: Reference the equations from earlier section.

Table 2: It appears that the letters A,B,C,D do not correspond to equations 1 and 2 (e.g. the Lenaerts equation (2) does not have a D, but Table 2 lists a D). This makes the arguments in section 3.1 difficult to follow (though it appears that C is consistent).

221: The MO fit bit here is distracting – move to end of section. Also, "impact of the different MO fits on the surface snow densities is negligible" – shouldn't there be zero impact, because the surface snow density is not a function of the MO? I.e., changing your surface density scheme will change the MO values after tuning, but changing the MO values will not change the surface density.

Section 3: I assume that the $r^2$ statistic is calculated by regressing the modeled surface densities against the 1:1 line – is that true?

227: You say: "This aligns with the fact that IMAU-FDM does not include densification by wind packing." I would agree that the subsurface densification scheme does not include wind packing as a densification, but that is also true in reality, isn't it? But, if you are using any of the Kaspers, Lenaerts, or modified Lenaerts equations in IMAU-FDM to set the surface density, I would suggest that wind packing is implicitly included in IMAU-FDM because those equations do include a wind term to account for wind packing.

230: "reduced with": do you mean "reduced by"?

231: You provide a specific number that is the surface density reduction (i.e. 18 kg m$^{-3}$), which implies that the density was reduced by that everywhere; but then you point to different regions and imply that the density changes are different in those regions. I think you mean something along the lines of, "On average over the AIS, the surface density was reduced by 18 kg m$^{-3}$. The surface density decreased more in the high accumulation margins and less in windy escarpment regions." However you rephrase it, please be sure to use careful language to indicate what less and more mean, since you are dealing a with a reduction (i.e., does "less" mean reduced by 30 rather than 18, or reduced by 10 instead of 18?).

235: remove comma after v1.2G

237: The idea that the surface density in Greenland is a function of annual temperature seems rather unphysical to me – I think that our knowledge of snow science is adequate to state with confidence that in reality, local conditions on relatively short temporal and spatial scales are the determinates of snow density (which is consistent with what you are finding in Antarctica). I suggest that the discrepancy is probably more due to lack of appropriate data in Greenland, and not significant differences in climatic conditions (there are many locations in Greenland and Antarctica that have very similar climates – so why in Greenland would the density instead be determined by the previous year's climate?).

240: This is inconsistent – the first sentence of the section says you ran IMAU-FDM without MO fits (does this mean with the original Arthern equation?), and then then next sentence talks about the resulting MO fits.

Table 3: Is there an error for your updated $\alpha$ value for $MO_{550}$ FDM v1.2A? It is 1000 times the others listed.

Table 3: Why do you not get RMSE values for the first 3 rows?

243: Do you mean that more densification is needed to match density profile measurements?

246: "When the locations are rerun …" sentence is confusing. Change to something like: "When the locations are rerun using the new MO values with the power fit for $MO_{830}$, the resulting RMSE of the modelled z830* and z830 (firn thickness) are respectively 25 and 23 % lower compared to the logarithmic fit (Fig. 3c)".

Figure 3: It would be nice on panel (c) if the colors matched the same model run in the other panels (e.g. FDM v1.2A-log should be orange in all cases)

286: change to: 'contain the most air' and 'along parts of the coast'

287: change to 'areas with very high accumulation'

Figure 5: the high values on panels (c) and (d) are one the peninsula, which somewhat removes structure from much of the ice sheet. Consider changing the color bars to max out at a lower value (e.g. 2 m for panel c) and use an extended colorbar (as you do in panel b).

315: You say that snowfall is highest in winter, but I think that is only true in certain parts of the ice sheet.

325: I am confused here: you say that the average seasonal amplitudes are defined as half the peak-to-peak values of the average seasonal cycle (and for FAC that is 2.4 cm). Then, you say that the average seasonal peak-to-peak values for FAC is 8.5 cm. I am not clear why, by your definition, the average season peak-to-peak is not just double (i.e. 4.8 cm) the average seasonal amplitudes. I am guessing that I am just not interpreting this correctly, so perhaps you can change the language to make it clearer.

328: Change the word 'It' (starting the sentence) to 'This result' or similar ("It" acting as a pronoun does not have specific noun it is referring to).

Figure 6d/335: It might be simpler to just say 'average day of year of maximum firn height' rather than "phase".

Figure 7: I am not sure of the best solution, but I think this figure is trying to convey too much information. It is rather difficult to look at the behavior of 7 lines. Additionally, I think it would be nice to pick a representative ~2 year period to plot with months marked to be able to see the seasonality (which you do a good job of discussing in the text) more clearly.

382: Figure 8 caption specifies that 8d is model minus altimetry, but this is not as clear in the text. Consider change to: 'which is calculated by subtracting the altimetry elevation change from the FDMv1.2A elevation change'.

383: Likewise, what is your subtraction for Figure 8e? FDMv1.2A minus FDMv1.1p1, or vice versa? Am I interpreting this correctly that blues are where FDMv1.2A is improved vis-à-vis FDMv1.1p1? I am a bit confused on 386-7 where you mention "less negative" – any negative (red) area in 8e is where FDMv1.1p1 is better than FDMv1.2A, correct? (Because you are subtracting the absolute residuals). So, to what are you referring that demonstrates the less negative?

389: trends → trend

---

## Author Comment (AC1)

**Response to Reviewer 3:**
First of all, we would like to thank the reviewer for his time for reviewing and editing this study. We greatly appreciate the insightful comments raised by C. Max Stevens and respond to each of them below. Based on your comments, we have improved the clarity of the text, especially Sections 2.6 and 6, we have made section 5.2 more quantitative and included a discussion on implications.

Responses to the comments of the reviewers are written in red and citations of the manuscript are written in blue.

Kind regards,

Sanne Veldhuijsen

**General comments:**
1. The sections (2.6, 6) describing the sensitivity tests need to be clearer – I am still not sure exactly what you did for them. For example, what exactly are you adding and subtracting for those runs? Are you, e.g., adding 8% to the accumulation over the entire forcing, and that is one of the sensitivity runs? It would help to clarify if you wrote exactly what you did to perform these tests, e.g. "We added 8% to the accumulation forcing, and then re-ran our model calibration procedure to get the optimal MO fits (or whatever) with that forcing. We then ran the model with these MO fits and compared the outputs to our baseline model run". (Or something along those lines). I think what I just wrote is the gist of what you are doing, but this is what I need clarified. The distance between sections 2.6 and 6 made this analysis more difficult to understand: I was looking for a table describing the sensitivity tests when I read section 2.6, but did not find it until the end (Table 4). I would also appreciate if table 4 was more specific: e.g., I can figure out what Accumulation+ is from looking at section 6, but it would be nice to have all the information in the table. Are you only doing these sensitivity runs for the observational sites, or did you run the model for the entire ice sheet in any of the sensitivity runs? When you say "10 additional locations" (line 198), what are those in addition to? Are those 10 sites special, or were they not used in the original calibration (if so, why not)? Finally, when doing these sensitivity tests, if you are adjusting the MO fits (section 6), with an empirical firn model, shouldn't you expect the model outputs to match the data reasonably well? This is indeed what you found, but my point is that if you are tuning a firn model using e.g. biased accumulation fields, the firn model will effectively act as a filter for that bias (or, the correction for it will be built into the model), so that the model output will match the data well. I think the upshot of my question is that I would like to see a bit more discussion contextualizing the meaning of the model sensitivity. Is the new model an improved representation of the physics compared to the previous IMAU FDM, or is it just a numerical response based on using different boundary conditions (RACMO forcing and surface density)?

Thank you for these comments. We agree that the description of the sensitivity analysis should be improved. We suggest replacing Table 4 with Table 2 (in Section 2.6) and Table 5 (in Section 6) to improve the link between these sections (See tables shown below).

[revised manuscript text omitted]

In addition to this, we will contextualize the model sensitivity, notably the role of MO fitting in reducing the sensitivity compared to without updated MO fit. Instead of using uniformly changed precipitation (-/+ 8%) and

temperature (-/+ 1.5K) in our revised manuscript we use spatially variable perturbations in the sensitivity experiments. We find a larger sensitivity of the simulated FAC (2- 4%) than with the uniformly changed precipitation and temperature (<0.5%), but still smaller than with the original MO fits (4-7% FAC).

2. In Figure 9 and 5.2: It seems to me that you have taken a rather qualitative approach here in looking at the elevation changes (e.g., "shows comparable patterns", "agreement seems"). I am curious how you have chosen to plot these time series: what is your zero/reference point? To me it would make most sense to plot them all zeroed at the start of the time series (1992) to see a direct comparison of how the data and model differ. The way it is plotted presently is misleading in a few cases: e.g. for site 6 (West Antarctica), the curves appear to be lined up reasonably well, but upon closer examination the data show a clear positive elevation trend while the model predicts a net decrease in elevation. This would enable a more quantitative analysis, such as fitting a trend line to each time series to get the long-term trends. Then, you could de-trend and do a time-series correlation for each site to quantify how well the models predict the shorter time-scale variability.

Thank you for this recommendation. We improved the clarity of the text, thereby highlighting the quantitative part of 5.2 (See our response to comment 42). We followed your suggestions and started each time series at zero in 1992. We acknowledge that the plot is indeed rather qualitative, but it serves as an addition to Figure 8 in order to show the temporal detail. In addition,we include  results of the FDM v1.1p2 run, which was not performed for the entire ice sheet due to computational demands, and could therefore not be included in Figure 8. Including this run is important as it tells us that the difference in surface elevation change between FDM v1.2A and FDM v1.1p1 is predominantly due to the updated forcing. We have not made this figure more quantitative as these are only 11 out of ~18,000 locations, and the presented analysis is applicable to the entire ice sheet. In line with this, the trends of the time series are already covered in Figure 8. In addition, we do not focus on the short term-variability in Figure 9, which is done in Section 5.1 for the entire ice sheet, and therefore we also use a 6-month moving average window.

We agree that a more detailed temporal variability analysis (although for the entire ice sheet and not the locations in Figure 9) by separating seasonal, long-term trends, shorter time (yearly/decadal) variability would be very interesting, but this is not done in this study, as this is beyond the scope/to limit the size of the paper.

In response to another reviewer, we have added results from the sensitivity analysis in this plot.

3. Discussion about implications: My general feeling in the paper what that it was light on discussing the implications of the research in the broader glaciological community. Given how much IMAU-FDM has been used for altimetry studies, I think adding a paragraph or a few sentences describing how the model changes affect our understanding of how the AIS is changing. Do the conclusions drawn by the users of IMAU-FDMv1.1p1 need to be updated?

Thank you for this recommendation. We propose to change Section 7: Remaining limitations and outlook, into: Implications, remaining limitations and outlook. We propose to add:

"IMAU-FDM has been used to correct altimetry observations in mass balance studies (e.g., Adusumilli et al., 2018; Willen et al., 2021). The difference in firn density change over time of FDM v1.2A compared to FDM v1.1p1 has an impact on mass change estimates in altimetry studies. Over the period 2003-2019, Smith et al. (2021) found that the total mass change of East Antarctica amounts to 195 Gt yr$^{-1}$, and of West Antarctica to -245 Gt yr$^{-1}$. When FDM v1.2A is used instead of FDM v1.1p1, the mass change over the period 2003-2015 for East Antarctica is 7.3 Gt yr$^{-1}$ lower, and for West Antarctica this is 1.2 Gt yr$^{-1}$ higher."

In addition to this, we will also add a discussion about the different impact of the updated forcing (almost completely responsible for the surface elevation change) and model parameterizations (mainly an impact on average firn density profiles).

**Line by line comments:**

4. line 44: Can give a sentence with a broad overview of what the 2 types are for readers not familiar with the difference?

We suggest to add:

"Physics-based models describe densification using a constitutive relationship between stress and strain for snow, while semi-empirical models use physics based densification equations in combination with parameters that are tuned according to observational density profiles."

5. line 47: "less": change to "fewer"

Based on feedback from reviewer 1, we suggest to change:"semi-empirical models require less poorly known parameters" into: "semi-empirical models require a smaller number of poorly constrained parameters "

6. Table1/section 2.1.1 leaves me wondering what surface density was used for run FDM v1.2A. Perhaps change Table 1 to include a 'Fresh Snow Density' column?

We agree that that information was missing. We suggest to update the table as follows:

**Table 1.** Abbreviations and characteristics of IMAU-FDM versions used in this study.

| Abbreviation | IMAU-FDM version | Forcing | Fresh snow density | $MO_{830}$ fit |
|---|---|---|---|---|
| FDM v1.2A | FDM v1.2 Antarctica | RACMO2.3p2, ERA-5 | Eq. (2); this study | Power (Eq. 5) |
| FDM FS-K | FDM v1.2 Antarctica | RACMO2.3p2, ERA-5 | Eq. (1); Kaspers et al. (2004) | Power (Eq. 5) |
| FDM FS-L | FDM v1.2 Antarctica | RACMO2.3p2, ERA-5 | Eq. (2); Lenaerts et al. (2012) | Power (Eq. 5) |
| FDM v1.2A-log | FDM v1.2 Antarctica | RACMO2.3p2, ERA-5 | Eq. (2); this study | Log (Eq. 4) |
| FDM v1.1p1 | FDM v1.1 | RACMO2.3p1, ERA-Interim | Eq. (1); Kaspers et al. (2004) | Log (Eq. 4) |
| FDM v1.1p2 | FDM v1.1 | RACMO2.3p2, ERA-5 | Eq. (1); Kaspers et al. (2004) | Log (Eq. 4) |

7. Section 2.1.2: Do you use the instant accumulation rate or 'mean accumulation over the lifetime of the firn layer'? If the former, please provide detail as to why you choose this and deal with the fact that densification will be zero if there is a timestep with zero accumulation.

We have used annual average accumulation rate, hence the dot above the b, which we define in line 84. For clarity, we have repeated the definition of ḃ in this section, as explained in the comment below.

8. line 106/Equation (3): the Arrhenius factor is missing the e. Also (and this is a bit pedantic), ḃ in Arthern et al. (2010) has units of kg m$^{-2}$ a$^{-1}$ , which then gives a densification rate dp/dt with units of kg m$^{-3}$ a$^{-1}$. (Arthern defines the units for the factor D in their Appendix B). Your accumulation units (mm w.e a$^{-1}$) are numerically equivalent, but with them the densification rate in your Equation 3 does not end up with units of density per time.

Thank you for noticing this. We have added the exponent term. In addition, we suggest to add: "ḃ is the annual average accumulation rate (kg m$^{-2}$ a$^{-1}$)".

9. lines 114-122: It is not entirely clear what you mean with $MO_{550}$ and $MO_{830}$: is $MO_{550}$ the value that should be used for $\rho < 550$, and $MO_{830}$ for $550 < \rho < 830$? Also, perhaps specify that MO added as a multiplier to equation (3).

This can indeed be improved. We suggest replacing this: "By comparing simulated and observed depths (m) of the 550 and 830 kg m$^{-3}$ density levels ($z_{550}$ and $z_{830}$, respectively) Ligtenberg et al. (2011) found that Eq. (3) requires a correction function, the so-called MO fits (ratio of Modelled to Observed depths), which turn out to depend on the accumulation rate. These MO fits are defined using the ratio between modelled and observed values of $z_{550}$ and 115 $z_{830*}$, where $z_{830*} = z_{830} - z_{550}$, using a simulation in which Eq. (3) is used without correction fits. These MO fits are defined using the ratio between modelled and observed values of $z_{550}$ and 115 $z_{830*}$, where $z_{830*} = z_{830} - z_{550}$, using a simulation in which Eq. (3) is used without correction fits. Ligtenberg et al. (2011) and Brils et al. (2021) used logarithmic correction functions, thus:"

By: "By comparing simulated and observed depths (m) of the 550 and 830 kg m$^{-3}$ density levels ($z_{550}$ and $z_{830}$, respectively) Ligtenberg et al. (2011) found that Eq. (3) requires correction terms $MO_{550}$ for p < 550 and $MO_{830*}$ for $550 < \rho < 830$, which are defined as the ratio of modelled and observed values of $z_{550}$ and $z_{830*}$, where $z_{830*} = z_{830} - z_{550}$. The correction terms $MO_{550}$ and $MO_{830*}$ are added as a multiplier to Eq. (3); MO values below one reduce the densification rate, values above one enhance the densification rate. $MO_{550}$ and $MO_{830*}$ are chosen as functions of the long term mean accumulation rate. Ligtenberg et al. (2011) and Brils et al. (2022) used logarithmic correction functions, thus:"

10. line 122: Add a sentence at the end of this section explaining that the parameters are tuned, and this tuning is described in Section 3.

We suggest to add: "The fitting parameters in Eq. (3) and (4) are tuned, which is described in Section 3.2".

This comment is also relevant for section 2.1.2 (fresh snow density), therefore we suggest to add:
L102: This retuning is described in Section 3.1".

11. Section 2.2: Do you set the bottom of your model domain to be the depth of the 830 horizon, or does it go deeper? In some locations in Antarctica, there is enough of a temperature gradient at the bottom of the firn to affect the temperature – if you are not modeling through the depth of the ice sheet, do you account for this heat flux?

In IMAU-FDM we do not set the bottom of the firn layer at 830 kg/m3. The lower boundary conditions for the thermal conduction are described in Section 2.3.1 of the revised manuscript.

12. line 142: not clear – why is the air content increasing?

We explain this as follows: "In the actual simulation after the spin-up, a minor trend (of <0.6 mm/yr averaged over the AIS) in total firn air content remains, because at the bottom of the column, ice with a density between 830 - 917 kg/m3, slowly replaces dense ice ($\rho$ = 917 kg/m3), which originates from the initialization of the firn column prior to the spin-up."

13. line 160: change semicolon between references to "and"

We changed this, thank you for noticing.

14. line 181: This is confusing – at the beginning of the paragraph you state that you use 125+8 density profiles, but then here you say you used 122. I think that a bit of language clarification in this paragraph will help – something along the lines of, 'We gathered data from 125 density profiles from firn cores and 8 density profiles from neutron…'. Then, '104 of our 133 cores fit the dry snow criteria needed for our MO-fitting routine.' Or something along those lines adding a bit more specificity.

This confusion comes from the fact that some density profiles are only used for the surface snow density tuning/evaluation.

Based on feedback of all reviewers, we propose to rewrite this part as follows:

"To tune and evaluate the firn model, we collected published firn density profiles from widely varying locations across the AIS (Fig. 1). We used 115 density profiles from firn cores and 8 density profiles from neutron density probe measurements, combining multiple datasets (van den Broeke, 2008; Schwanck et al., 2016; Bréant et al., 2017; Fernandoy et al., 2010; Montgomery et al., 2018; Fourteau et al., 2019; Olmi et al., 2021; Winstrup et al., 2019). The firn cores and neutron density probe measurements are mostly obtained in summer months between 1980 and 2020. For tuning the fresh snow density parameterization, observations of the density of surface snow, defined as the top 0.5 m, from 63 firn cores could be used. Eight additional surface snow density values from density profiles of neutron density probe measurements and two from firn cores (Montgomery et al., 2018) were added after the tuning, and are thus only used for evaluation. For tuning the MO fits, only dry firn cores are used, as Eq. (3) describes dry snow densification only. A firn core is considered dry if its location experiences on average less melt than 5 % of the average annual accumulation. For the MO fits, 92 dry firn cores could be used, which improves upon Ligtenberg et al. (2011), who used data from 48 dry firn cores. In addition to the 94 dry cores used for the tuning of MO ratios, another 12 wet cores were used to evaluate the $z_{550}$ and $z_{830}$ model output. For evaluating FAC output, 31 firn cores could be used. Table S1 in the Supplement lists all measurements that have been used, the corresponding coordinates, method and citation."

15. line 182: This sentence has the word 'density' 3 times – can you rewrite it to make it a bit easier to read?
We suggest to change: "Eight additional surface snow density values from density profiles of neutron density probe measurements".
Into:
"Eight additional surface snow density values from neutron density probe measurements".

16. line 184: In some locations, this 0.5 m is likely snow (i.e. less than a year old), but in the interior it is several years of accumulation. Is this a concern, either for tuning or for interpreting model outputs?
For tuning and interpretation, we compared simulations and observations that are both of the upper 0.5 m. Furthermore, densification is still limited in the majority of the locations as either the time for densification is very short, due to the high accumulation, or the densification goes slow due to the very low temperatures. Therefore, we think this is not a concern for interpreting model outputs.

17. line 203/205 (elsewhere too?): fix the +- to +/- (may be done in manuscript typesetting?)
Thank you for noticing this, we corrected this.

18. line 219: Reference the equations from earlier section.
We added this.

19. line Table 2: It appears that the letters A,B,C,D do not correspond to equations 1 and 2 (e.g. the Lenaerts equation (2) does not have a D, but Table 2 lists a D). This makes the arguments in section 3.1 difficult to follow (though it appears that C is consistent).

Thank you for noticing. We improved, eq. (1) and (2) now have coefficients: A, B, C and E.

20. line 221: The MO fit bit here is distracting – move to end of section. Also, "impact of the different MO fits on the surface snow densities is negligible" – shouldn't there be zero impact, because the surface snow density is not a function of the MO? I.e., changing your surface density scheme will change the MO values after tuning, but changing the MO values will not change the surface density.

The fresh snow density is independent of the MO fit, however the surface snow density (top 0.5 m) has undergone some densification, so this is in fact dependent on the MO fit (although negligible), as mentioned above. We agree that this is confusing in this part of the section, so therefore we have moved this towards the end to avoid confusion.

21. line Section 3: I assume that the r2 statistic is calculated by regressing the modeled surface densities against the 1:1 line – is that true?

No, it is calculated by evaluating the scatter of the data points around a fitted regression line, because with this statistic we want to represent how well the model captures the spatial variation instead of how close it is to the 1:1 line.

22. line 227: You say: "This aligns with the fact that IMAU-FDM does not include densification by wind packing." I would agree that the subsurface densification scheme does not include wind packing as a densification, but that is also true in reality, isn't it? But, if you are using any of the Kaspers, Lenaerts, or modified Lenaerts equations in IMAU-FDM to set the surface density, I would suggest that wind packing is implicitly included in IMAU-FDM because those equations do include a wind term to account for wind packing.

For FDM with Kaspers and modified Lenaerts, this is indeed true. But the Lenearts parameterization is calibrated with fresh snow density observations of the skin layer, so wind-packing has likely not yet occurred there. This explains the underestimation with Lenaerts, and e.g. not with Kaspers.

23. line 230: "reduced with": do you mean "reduced by"?

Thank you for noticing this, we corrected this.

24. line 231: You provide a specific number that is the surface density reduction (i.e. 18 kg m$^{-3}$), which implies that the density was reduced by that everywhere; but then you point to different regions and imply that the density changes are different in those regions. I think you mean something along the lines of, "On average over the AIS, the surface density was reduced by 18 kg m$^{-3}$. The surface density decreased more in the high accumulation margins and less in windy escarpment regions." However you rephrase it, please be sure to use careful language to indicate what less and more mean, since you are dealing a with a reduction (i.e., does "less" mean reduced by 30 rather than 18, or reduced by 10 instead of 18?).

We agree that this might be confusing, therefore we suggest changing this:
"In general, the surface snow density is reduced with 18 kg m$^{-3}$, especially in the high accumulation margins, and to a lesser extent in windy escarpment regions (Fig. 2b)."

Into: "Overall the surface snow density is reduced with 18 kg m$^{-3}$, most notably in the high accumulation margins, and to a lesser extent in windy escarpment regions (Fig. 2b)."

25. line 235: remove comma after v1.2G

Done.

26. line 237: The idea that the surface density in Greenland is a function of annual temperature seems rather unphysical to me – I think that our knowledge of snow science is adequate to state with confidence that in reality, local conditions on relatively short temporal and spatial scales are the determinates of snow density (which is consistent with what you are finding in Antarctica). I suggest that the discrepancy is probably more due to lack of appropriate data in Greenland, and not significant differences in climatic conditions (there are many locations in Greenland and Antarctica that have very similar climates – so why in Greenland would the density instead be determined by the previous year's climate?).

We agree. We suggest changing this part: "The fresh snow density parameterization for Greenland used in FDM v1.2G, is only a function of yearly temperature (Brils et al. (2021). This is in contrast with Antarctica, where we found a strong dependency with instantaneous wind speed and temperature, which is in line with previous work

(Keenan et al., 2021; Lenaerts et al., 2012; van Kampenhout et al., 2017), and likely owing to the larger range in temperature and wind speed conditions during snow deposition in Antarctica."

Into: "The fresh snow density parameterization for Greenland used in FDM v1.2G, is only a function of yearly temperature (Brils et al. 2022), owing to a lack of co-located surface snow density, temperature and wind speed observations on the GrIS. This contrasts to the expression used here, which includes a dependency on instantaneous wind speed and temperature."

27. line 240: This is inconsistent – the first sentence of the section says you ran IMAU-FDM without MO fits (does this mean with the original Arthern equation?), and then then next sentence talks about the resulting MO fits.
Yes, this is indeed the original Arthern equation. To clarify, we suggest to replace this sentence by: "To tune the dry snow densification rate we first performed a simulation of FDM v1.2A without MO corrections. The resulting MO fits and statistics are listed in Table 4 and shown in Figures 3a,b."

28. line Table 3: Is there an error for your updated α value for $MO_{550}$ FDM v1.2A? It is 1000 times the others listed.
Thank you for noticing this, this should be 1.228 instead of 1288, we corrected this.

29. line Table 3: Why do you not get RMSE values for the first 3 rows?
We have not included this, since these statistics are taken from Ligtenberg et al. (2011) and based on a smaller dataset, which is therefore not fair to compare to $MO_{550}$/$MO_{830}$ FDM v1.2A. However, FDM v1.2A and FDM v1.2A-log have similar datasets, so these allow a comparison in terms of RMSE.

30. line 243: Do you mean that more densification is needed to match density profile measurements?
Yes indeed, we have added this:
"The fresh snow density is now independent of accumulation and generally lower, and hence more densification is needed at high accumulation sites to match density measurements".

31. line 246: "When the locations are rerun …" sentence is confusing. Change to something like: "When the locations are rerun using the new MO values with the power fit for $MO_{830}$, the resulting RMSE of the modelled $z_{830*}$ and $z_{830}$ (firn thickness) are respectively 25 and 23 % lower compared to the logarithmic fit (Fig. 3c)".
That is indeed clearer, we have changed this according to your suggestion.

32. line Figure 3: It would be nice on panel (c) if the colors matched the same model run in the other panels (e.g. FDM v1.2A-log should be orange in all cases).
We agree that those colors should match, the color of FDMv1.2A in all panels is changed to blue, of FDM v1.1p1 to orange, of FDM v1.2A-log to red.

33. line 286: change to: 'contain the most air' and 'along parts of the coast'
We corrected this.

34. line 287: change to 'areas with very high accumulation'
We changed this.

35. line Figure 5: the high values on panels (c) and (d) are one the peninsula, which somewhat removes structure from much of the ice sheet. Consider changing the color bars to max out at a lower value (e.g. 2 m for panel c) and use an extended colorbar (as you do in panel b).
To improve the visibility of spatial detail, we suggest to update this to:

[Figure]

36. line 315: You say that snowfall is highest in winter, but I think that is only true in certain parts of the ice sheet.

We agree that this does not hold for the entire ice sheet, but 6a shows that this holds for the mean of the ice sheet. Therefore, we suggest to change this sentence:

"Snowfall is highest in winter, while firn densification, melt and sublimation peak in summer."

Into:

"Averaged over the ice sheet, snowfall is highest in winter, while firn densification, melt and sublimation peak in summer."

37. line 325: I am confused here: you say that the average seasonal amplitudes are defined as half the peak-to-peak values of the average seasonal cycle (and for FAC that is 2.4 cm). Then, you say that the average seasonal peak-to-peak values for FAC is 8.5 cm. I am not clear why, by your definition, the average season peak-to-peak is not just double (i.e. 4.8 cm) the average seasonal amplitudes. I am guessing that I am just not interpreting this correctly, so perhaps you can change the language to make it clearer.

We found that we used the words seasonal amplitude and seasonal peak-to-peak in the wrong way. As is pointed out, seasonal peak-to-peak is indeed twice the seasonal amplitude.

What we actually want to describe here are, firstly, the amplitude of the ice-sheet wide FAC/firn height, and secondly, is the average of all local FAC/firn height amplitudes.

Therefore, we suggest to change this:

"The spatially average seasonal amplitudes of the firn thickness and FAC are 3.5 and 2.4 cm, which amounts to Antarctic wide integrated volumes of 463 and 304 km$^3$, respectively. This agrees with the modelled firn thickness amplitude of 3.5 cm by Medley et al. (2020). The average seasonal peak-to-peak values of firn thickness and FAC, defined as the average difference between the highest and the lowest value of each year, are considerably higher, 13.5 and 8.5 cm, amounting to volume changes of 1784 and 1123 km$^3$, respectively (Fig. 6c). The difference between the seasonal amplitudes and peak-to-peak values is due to interannual variations in the timing of the seasonal maximum and minimum."

Into:

"The seasonal amplitudes of the ice sheet wide firn thickness and FAC are 3.1 and 2.1 cm, which amounts to Antarctic wide integrated volumes of 428 and 307 km$^3$, respectively. This agrees with the modelled firn thickness amplitude of 3 cm found by Medley et al. (2022). The average of all local seasonal amplitudes are considerably higher, 6.8 and 4.2 cm, amounting to volume changes of 892 and 562 km$^3$, respectively (Fig. 6c). The reason is the spatial variation in timing of the seasonal maximum and minimum (Fig. 6d)."

38. line 328: Change the word 'It' (starting the sentence) to 'This result' or similar ("It" acting as a pronoun does not have specific noun it is referring to).

"It" has been changed to: "This result"

39. line Figure 6d/335: It might be simpler to just say 'average day of year of maximum firn height' rather than "phase".

Yes I agree. We have the labels and caption of Figure 6d, and we have changed this:

"Figure 6d shows the average phase of the firn thickness maximum."

Into: "Figure 6d shows the average day of the year of maximum firn height."

40. line Figure 7: I am not sure of the best solution, but I think this figure is trying to convey too much information. It is rather difficult to look at the behavior of 7 lines. Additionally, I think it would be nice to pick a representative ~2 year period to plot with months marked to be able to see the seasonality (which you do a good job of discussing in the text) more clearly.

We agree that the figure is quite full. To avoid overlapping, therefore we already had shifted the FAC and temperature anomalies time series in the vertical direction. This section is about decadal variation, selecting two representative years would not show any decadal variation, and we already discuss the seasonal variation in Section 4.3 and Figure 6a. So we decided to keep the figure as is.

Another option would be to only include timeseries with a running average window of 5 year? But then you don't see the interannual variability in seasonal cycle.

The drawback of this solution is that the current figure nicely shows that interannual variability still relates to individual events. Smoothing would remove that information. We decided to keep Figure 7 but highlight more that it focuses on interannual variability and not on decadal variability.

41. line 382: Figure 8 caption specifies that 8d is model minus altimetry, but this is not as clear in the text. Consider change to: 'which is calculated by subtracting the altimetry elevation change from the FDMv1.2A elevation change'.
We agree that this requires some clarification in the text.
We have added: "Figure 8d shows the residual surface elevation trend of FDM v1.2A, which is calculated by subtracting the FDMv1.2A elevation trend from the altimetry trend".

42. line 383: Likewise, what is your subtraction for Figure 8e? FDMv1.2A minus FDMv1.1p1, or vice versa? Am I interpreting this correctly that blues are where FDMv1.2A is improved vis-à-vis FDMv1.1p1? I am a bit confused on 386-7 where you mention "less negative" – any negative (red) area in 8e is where FDMv1.1p1 is better than FDMv1.2A, correct? (Because you are subtracting the absolute residuals). So, to what are you referring that demonstrates the less negative? This figure does also include the altimetry product, however we acknowledge that this is not clearly described. The figure is calculated as the absolute residual of FDM v1.1p1 (|altimetry-FDM v1.1p1|) minus the absolute residual of FDM v1.2A (|altimetry-FDM v1.2A|).

We changed the figure label: "Difference in altimetry agreement between FDM v1.2A and FDM v1.1p1".
And added in the figure caption: "**(e)** Difference in altimetry agreement between FDM v1.2A and FDM v1.1p1. (absolute resdiaul FDM v1.1p1 compared to altimetry minus absolute residual FDM v1.2A compared to altimetry)".

In addition, we suggest to change: "In Figure 8e we subtracted the absolute residual surface elevation change of FDM v1.2A and of FDM v1.1p1 from each other, which yields the improvement of FDM v1.2A compared to FDM v1.1p1. In Figures 8d and e, regions with ice-dynamical imbalance are excluded. The surface elevation change of FDM v1.2A has improved in most regions, especially in Dronning Maud Land, Wilkins Land and Adelie Land, where the FDM v1.2A trend has become either more positive or less negative. The residual absolute trend of FDM v1.2A has been reduced by 17 % compared to FDM v1.1p1 (from 2.6 to 2.1 cm/yr)."

Into:
"In Figure 8e we show the difference in altimetry agreement between FDM v1.2A and FDM v1.1p1, which is calculated by subtracting the absolute residual surface elevation trend compared to altimetry (Fig. 8d) of FDM v1.1p1 from the similarly derived absolute residual of FDM v1.2A. The blue areas indicate an improvement in altimetry agreement of FDM v1.2A compared to FDM v1.1p1. The average absolute residual trend of FDM v1.2A compared to altimetry has been reduced by 17 % compared to FDM v1.1p1 (from 2.6 to 2.1 cm/yr). The improvement is most notable in Dronning Maud Land, Wilkins Land and Adélie Land. In Dronning Maud Land the FDM v1.2A trend has become more positive, and in Wilkins Land and Adélie Land the trend has become less negative (Fig. 8b,c, for region names see Fig. 1)."

43. line 389: trends → trend
This mistake has been improved, thank you for pointing it out.

---

## Author Comment (AC2)

**Response to Reviewer 1:**
First of all, we would like to thank dr. Vincent Verjans for his time for thoroughly reviewing this study. We appreciate the constructive feedback we received. By following your suggestions, we have, amongst other things, revised the sensitivity analysis, model evaluation, and various parts of the text. We believe these changes have led to an improvement in the quality of the results and manuscript.

Responses to the comments of the reviewers are written in red and citations of the manuscript are written in blue.

Kind regards,

Sanne Veldhuijsen

**Major comments:**
**1a)** Many of the possible experiments presented in Section 2.6 are not even mentioned in the results section about uncertainty (Section 6). In Section 2.6, the authors mention various scenarios of accumulation and temperature reduction for the spin-up period (three for each variable). However, they only discuss a single test per variable in Section 6, and it is not specified which one it is. This obviously requires more clarity.
Thank you for pointing this out. We think there has been a misunderstanding, as our experiment description was unclear. We only tested three scenarios within the spin-up forcing uncertainty experiments, one with a lower temperature during the spin-up, one with a lower precipitation during the spin-up and one with a combination of lower temperature and precipitation during the spin-up period. In the revised manuscript, we also test the sensitivity to a higher temperature, higher precipitation and a combination of higher temperature and precipitation during the spin-up period within these experiments. We propose to clarify this as follows, in Section 2.6:

"Another important source of forcing uncertainty is the climate forcing during the spin-up period. As explained in Section 2.2, the spin-up forcing is obtained by looping over the 1979-2020 forcing data. However, firn core (Thomas et al., 2017) and isotopes (Stenni et al., 2017) studies show that in the Antarctic Peninsula and Ellsworth Land, the accumulation and temperature were typically about 10 % and 1 K lower during the last centuries. In addition, over the remainder of the AIS, the accumulation and temperature were typically about 5 % and 0.5 K higher or lower during the last centuries. To investigate the typical effect this has on the IMAU-FDM results, we include schematic sensitivity tests in which we adjusted the accumulation and temperature forcing with these values only during the spin-up period. To mimic a gradual increase in precipitation over the Antarctic Peninsula and Ellsworth Land, we create a spin-up experiment in which the precipitation up until the third-to-last loop (the 42-year reference period) is decreased by 10 %, in the second-to-last by 6.66 %, and in the last one by 3.33 % with respect to the mean precipitation of 1979-2020. To mimic a gradual increase in temperature over the Antarctic Peninsula and Ellsworth Land, we create a spin-up experiment in which the temperature up until the third-to-last loop is decreased by 1 K, in the second-to-last by 0.66 K, and in the last one by 0.33 K. Over the remainder of the AIS accumulation and temperature were both decreased and increased with 5 % and 0.5 K during the spin-up period. As the uncertainties between temperature and accumulation during the spin-up period are dependent, we also performed a test in which we simultaneously adjusted the temperature and precipitation."

To clarify this further, we added the table shown below in Section 2.6, in which we give an overview of all the experiments.

**Table 2.** Overview of the sensitivity experiments. The inputs and parameters are adjusted by substracting and adding the uncertainties either during the entire run or only during the spin up. The values used are based on the references provided or on analysis within this work.

| Experiment name | Variable | Variation | When | Reference |
|---|---|---|---|---|
| MO fits | MO fits | + and -95 % confidence interval | entire run | Figs. 3a,b |
| $\rho_s$ | $\rho_s$ | + and -30 kg m$^{-3}$ | entire run | RMSE evaluation Section 3.1 |
| $T_s$ | $T_s$ | + and - spatial variable K | entire run | $\sigma$ from Carter et al. (2022) |
| b | b | + and - spatial variable % | entire run | $\sigma$ from Carter et al. (2022) |
| spin up $T_s$ | $T_s$ | +0/+1 K and -1/-0.5 K[a] | spin up | Stenni et al. (2017) |
| spin up b | b | +0/+10% and -10/-5 %[a] | spin up | Thomas et al. (2017) |
| spin up $T_s, b$ | b and $T_s$ | +0/+5 %, +0/+0.5 K and -10/-5 %, -1/-0.5 K[a] | spin up | Stenni et al. (2017); Thomas et al. (2017) |

[a]The values on the left side of the slash indicate the variation for the Peninsula and Ellsworth Land, and the values on the right side of the slash for the remainder of the AIS. Temperature and accumulation for the Peninsula and Ellsworth Land are the second last loop of spin up varied with 6.66 % and 0.66 K and in the last loop with 3.33 % and 0.33 K.

Similarly, the sensitivity tests using "the 95 % confidence intervals of the MO fits", and the "uncertainty of the fresh snow density" are not discussed. I strongly recommend to clarify the links between Sections 2.6 and 6. For example, each sensitivity test should be given a name. And there should be a Table that specifies the sensitivity tests, their corresponding variable adjustments, and results.

Thank you for this recommendation. In the revised manuscript, we will include a discussion of the MO fits and fresh snow density results. To clarify the link between Sections 2.6 and 6, we added Table 2 in Section 2.6 and Table 5 in Section 6, which both include the names of the experiments.

**Table 5.** Overview of the change in annual average firn air content and surface elevation change of the extrapolated experiments and sample locations experiments. The sample location experiments also include the experiments without adjusted MO fits. An overview of the experiments including the prescribed uncertainties are listed in Table 2.

| Experiment name | Extrapolated to entire ice sheet | | Sample locations (without adjusted MO fits)[a] | |
|---|---|---|---|---|
| | FAC (%) | dH/dt (mm yr$^{-1}$) | FAC (%) | dH/dt (mm yr$^{-1}$) |
| MO fits | 5.1 | 0.48 | 5.2 | 0.75 |
| $\rho_s$ | 1.2 | 1.46 | 0.7 (5.8) | 2.05 (2.44) |
| $T_s$ | 1.8 | 0.66 | 2.3 (4.9) | 0.65 (1.50) |
| b | 4.0 | 5.63 | 3.7 (6.8) | 7.79 (7.99) |
| spin up $T_s$ | 0.5 | 4.40 | 0.5 (1.5) | 5.16 (5.22) |
| spin up b | 0.7 | 18.96 | 0.7 (1.1) | 22.45 (22.40) |
| spin up $T_s, b$ | 0.7 | 14.93 | 0.6 (0.8) | 17.45 (17.38) |

[a]The values between brackets indicate the uncertainties of the experiments without updated MO fits.

The Table 4 needs some further adjustments. I suppose that the FAC values in Table 4 are calculated only over the dataset of firn cores, and not over the entire ice sheet, which should be specified in the caption. Table 4 should also include the results from the spin-up perturbation experiments: 3 per variable (see Section 2.6), and the experiments combining temperature and accumulation perturbations (not explained in Section 2.6). Finally, the sensitivity experiments use ice sheet wide averages of precipitation spread (+/- 8 %) and of temperature uncertainty (+/- 1.5 K). However, local uncertainties can differ strongly from the ice sheet averages. Where possible, I recommend using spatially variable uncertainties in the sensitivity analysis.

Thank you, we agree that Table 4 (divided between Tables 2 and 5 in the revised manuscript) can be improved. We have added your suggestions. In addition, we have used spatially variable accumulation and temperature sensitivity uncertainties based on the standard deviation between RCMs and reanalysis datasets, which we obtained from Carter et al. (2022).

**1b) Ice-sheet wide method for uncertainty quantification**

The authors have investigated the results of the sensitivity experiments only at the firn core locations. However, this is of little interest to the community. A method to compute uncertainty estimates over the entire ice sheet in the different components of Eq. (6), and thus ultimately on dh/dt, should be developed. A straightforward approach could be to sample perturbations in all uncertainty sources mentioned in Section 2.6, compute

simulations at key locations, and regress uncertainty estimates against climatic variables. However, I leave the choice to the authors of how to best estimate uncertainty from their model, and I note that they have already worked on similar issues (e.g., Kuipers Munneke et al., 2015). The outcome of a more thorough uncertainty analysis should be:
- uncertainty bands in time series of Figure 7 and Figure 9
- maps of total uncertainty in modeled surface elevation change (and possibly of the uncertainty components)
- a quantification and discussion of the different components to uncertainty in dh/dt across Antarctica

Thank you for this recommendation. We agree that ice-sheet wide uncertainty estimates are of greater interest to the community. We expanded the uncertainties estimates by regressing the sensitivity of the uncertainty sources mentioned in Section 2.6 against climatic variables for the average FAC and dH/dt. We propose to include details of the sensitivity analysis in the supplementary material. As there are many interacting processes in the model, which can be variable in time, we propose to regress dH/dt instead of the different components of Eq. (6) separately. In addition, we propose to estimate the ice-sheet wide average FAC sensitivities as well in a similar manner. The empirical relations for each sensitivity experiment are included in the Figures S3. An example of these empirical relations is given in the Figure below.

[Figure]

Fig. Examples of the MO fit empirical relations for the relative FAC and surface elevation change uncertainty.

These empirical relations are used to expand the sensitivity of the sample locations to the entire ice sheet. We have added the resulting maps of the total FAC and surface elevation change uncertainty and the contribution of each uncertainty source in Figures S4.

In order to show the variation in uncertainty of FAC and surface elevation over time, we plotted time series of surface elevation change and FAC including the sensitivity for the sample locations (Figs. S5a,b). This shows that nearly all uncertainty (>98%) of the ice sheet averaged surface elevation change and FAC is caused by the spin-up experiments. As the spin-up experiments cause a relatively uniform spatial and temporal response in FAC and surface elevation change across the ice sheet, the uncertainties can be expanded over the ice sheet for each timestep. In Figure S5c we only show the dominant sensitivities of the total AIS surface elevation change and FAC time series for the spin-up experiments. To expand FAC change for the combined accumulation and temperature spin-up experiments, we obtained additional empirical relations, see Fig. S3f. We include a discussion of these new sensitivity analysis results in Section 6.

We decided not to include uncertainty bands of the components of eq. (6) in Figure 7. In the Figure below we show time series of the uncertainties of (a) yearly rates and (b) cumulative anomalies of $V_{snow}$ and $V_{fc}$ of all sensitivity experiments for the sample locations. However, in our manuscript, we rather focus on the total surface elevation change, and the contribution of each uncertainty source, as this is of greater interest to the community, instead of on the processes separately.

[Figure]

Results from our sensitivity analysis are now also included in the comparison with satellite altimetry, by including uncertainty bands of dH/dt in Figure 9. In addition, we compare dH/dt results for the Antarctic Peninsula and Ellsworth Land over the period 2003-2015 of our sensitivity analysis to satellite altimetry, and we found that the residual has been reduced by 25% compared to the reference FDM v1.2A simulation. See our response to comment 61 of Reviewer 3 for more details.

**2) Statistical procedure**
In statistical calibration of parameters such as the MOs, it is well-known that validation should be, to some degree, independent of calibration. Here, the authors use the same dataset to calibrate their new parameters (MOs and surface density) as to evaluate their calibration. As such, the validation is not meaningful. I realize that firn data is sparse, and it can be argued that excluding a part of the dataset from calibration might be detrimental to model fitting. But that does not exclude a form of k-fold cross validation to better evaluate the fits. Furthermore, this approach would provide more robust uncertainty estimates on the MO and surface density values. For the evaluation, the authors use only 10 additional measurements in the evaluation dataset, in addition to the ones used in the calibration dataset.
Thank you for this recommendation. We have included k-fold cross validation for the MO fit and surface snow density. We found that the RMSE for the surface snow density amounts to 30.4 kg/m3 (compared to 28.8 kg/m3 for the reference FDM v1.2A), the RMSE for Z830 to 9.11 m (compared to 8.83 m for reference FDM v1.2A), and the RMSE for Z550 to 2.49 m (compared to 2.44 m for the reference FDM v1.2A). This additional evaluation will be included in the manuscript.

This modest introduction of independent data in the evaluation results in (line 243): "a slightly deteriorated correlation for the MO550." when compared to the FDMv1.1p1 parameterization.
I think there is a misunderstanding here. The deteriorated correlation of the $MO_{550}$ fit is not because of the introduction of the independent surface snow density data, as the independent surface snow density data has no impact on the MOfit. The r-squared value of the FDM v1.1p1 $MO_{550}$ fit is higher (0.43) than the FDM v1.2A $MO_{550}$ fit (0.37). The reason for this could be that the previous fresh snow density parameterization was based on accumulation, which can give a higher dependency of accumulation and $MO_{550}$. We propose to rephrase Lines 242-244 as: "The optimal MO550 fit is less steep than the one found by Ligtenberg et al. (2011). The fresh snow density is now independent of accumulation and generally lower, and hence more densification is needed at high accumulation sites to match the density profiles. This might explain the slightly deteriorated correlation of the $MO_{550}$ fit."

Furthermore, in Section 6, the authors demonstrate that the calibration of the MOs is sensitive to realistic uncertainty in climate forcing, and that this sensitivity strongly impacts results in FAC, z550, and z830 (Table 4). These two aspects thus show evidence that the statistical fits performed in this study are probably sensitive to noisy features inherent to the observations used in the calibration.
This is indeed true; therefore, we include k-fold cross validation and a sensitivity analysis with the MO fits.

In Table 3, I recommend providing the RMSE values in all the rows, to give the bias values in addition to the RMSEs, and to compute these fit statistics also with respect to the FAC values of the dataset. Finally, if I understand the process correctly, the comparison of fits between FDMv1.1p1 and FDM v1.2A for z550, and z830 is unfair. FDMv1.1p1 was calibrated with climate output of a previous RACMO version and another surface density parameterization. As such, it is obvious that it performs worse than FDM v1.2A when it is used with RACMO2.3p2 forcing and another equation for surface density.

In this table we only show the statistics of the MO fits and not the evaluation of the density, which is done in Figure 3. The FDMv1.1p1 MO fit statistics are taken from Ligtenberg et al. (2011), and are thus entirely based on old calibration, parameterizations and input, hence fair. We include characteristics of all simulations used in Table 1, to avoid raising the impression that the comparison is unfair. In addition, the RMSE column is about the RMSE with the fit, and only valuable to compare for FDM v1.2A-log and FDM v1.2A, as these have the same set of data points. We have added FAC evaluation of FDM v1.2A-log, FDM v1.2A, and FDM v1.1p1 in Figure 3e (See figure below).

[Figure]

Furthermore, it is unclear how much of the improved fit to observations is due to the re-calibration of IMAU-FDM versus the updated climatic forcing and the updated surface density parameterization. All these aspects should be addressed in the manuscript, and identified as caveats in the comparison.

This is an important point. To overcome this, we now include FDM v1.1p1 in the surface snow density evaluation (Table 3 and Figure 2a).

The same holds when comparing FDMv1.1p1 and FDM v1.2A to the altimetry product (Figure 8): it remains unknown what part of the improvement is due to changes in the densification equation, changes in the climatic forcing, and changes in the surface density parameterization. As pointed out by the authors, Figure 9 suggests that most of the improvements are due to the update of RACMO, which raises questions concerning the improved performance of FDM v1.2A in simulating firn processes. This should be discussed more in depth.

Regarding the surface elevation change, the forcing indeed explains almost all of the difference as is shown by Figure 9. However, Figure 2b and 3d show that changes in updates in the fresh snow density and firn densification parameterizations have substantial impact on the time averaged firn profiles. These outputs are important when IMAU-FDM is used for instance in the input-output method to estimate AIS mass balance, and to assess the meltwater buffering capacity of the firn layer. This distinction will be discussed in Section 7 (Implications, remaining limitations, and outlook) following suggestions from Reviewer 3.

3) Neglect of melt areas

The MO parameterization is constrained only in dry firn areas, but the model is used in wet firn areas also. One can reasonably expect errors to be much larger in modeling wet firn densification. While this topic is not the focus of the study, any ice-sheet wide study of firn evolution should at least discuss this limitation. Ideally, I would encourage the authors to evaluate IMAU-FDM in the wet firn areas also, by comparing modeled FAC to observed FAC from firn cores. They could also provide uncertainty estimates that are valid for melt areas.

Thank you for pointing this out. It can indeed be expected that densification rates are different between dry and wet firn. Unfortunately, firn cores from melt areas are sparse in Antarctica, and we only found two that were of sufficient quality to calculate FAC. Especially in melt regions, forcing uncertainties have a large impact on the simulated FAC. In Greenland more cores with FAC observations are available, see Brils et al. (2022), their Fig. 5 (see below). Based on that result, the model appears quite capable of simulation wet firn densification.

[Figure]

**Figure 5.** Modelled vs. observed firn air content in metres. Dry locations are indicated with circles, whereas wet locations are indicated with triangles. A location is labelled as dry if it experiences 5 % less melt than accumulation during the spin-up period. The blue lines indicate the uncertainty in the v1.2G results.

(Brils et al. 2022).

**Minor comments**
1) There is a general lack of quantification. I encourage the authors to identify all the uses of words such as "reasonably", "somewhat", "roughly", "improved", and "substantially". These should be complemented by quantitative values.
We agree and have included quantitative values throughout the manuscript.

2) In the densification equation (Eq. (3)), it seems to me that the mean long-term accumulation rate is used. An alternative approach is to use the mean accumulation rate over the lifetime of each specific firn layer, which is more representative of the effect of overburden stress (Li and Zwally, 2011). This aspect could be important given that decadal snowfall variability can be large on the Antarctic ice sheet. And there is no physical reason for the densification of a firn layer to be a function of past accumulation rates. Why did the authors choose to use the mean long-term accumulation rate?
We acknowledge that taking ḃ as a constant is an approximation that simplifies the densification rate, but the uncertainty this introduces is minor. Brils et al. (2022) demonstrated for two locations in Greenland that the error in the load experienced by a layer of firn is <3.2% at Summit and <1.9% at Dye-2. We included this information as follows: "ḃ is used as a measure of the overburden pressure, thereby assuming that the accumulation rate is constant over time. Using FDM v1.2G for Greenland Brils et al. (2022) shows that assuming a constant ḃ introduces a minor error in the load experienced by a layer of firn (e.g. <3.2 % at Summit and <1.9 % at Dye-2)."

3) Lack of clarity about the firn core dataset
Depsite re-reading several time Section 2.5, it is still unclear to me how the authors selected their dataset.
- "We used 125 density profiles from firn cores and 8 density profiles from neutron density probe measurements": but the wet firn cores were discarded for the calibration, so how were they used in this work?
Wet firn cores were used for the evaluation and calibration of the surface snow density, and used for firn densification evaluation. We clarified this in the revised paper and have rewritten this section, see our response to comment 14 of Reviewer 3.

- "For the MO fits, 104 dry firn cores could be used (...) To evaluate the firn density profiles from the simulation using the derived MO fits, 122 firn cores could be used.": what explains this difference of 18 cores?
Some of these cores are wet cores, and others have only been used for the surface snow density. We clarified this in the revised paper.

- in Section 2.6 "105 observational locations shown in Figure 1": 105 is not even mentioned in Section 2.5. We clarified this in the revised paper, and plotted the sensitivity analysis locations separately in Figure S1 (See our response to comment 51).

- in caption of Figure 1 "The grey circles indicate ten additional locations that were included in the sensitivity analysis": give more details about these 10 additional firn cores in Section 2.5.
We have rewritten these sections by taking the comments above into account, and feedback from the other reviewers.

- Concerning access to the firn core data, please see the Data policy section of The Cryosphere: "Authors are required to provide a statement on how their underlying research data can be accessed. This must be placed as the section "Data availability" at the end of the manuscript."
References and characteristics of firn core data will be added as supplementary material. We refer to this in the data availability statement of the revised manuscript.

**Specific comments**
1. l4 "improved": specify that this refers to a previous model version. We have changed this into: "We have improved previous fresh snow density and firn densification parameterizations."

2. l5 "observations": change to "firn core observations". Thank you for noticing this, we have changed this accordingly.

3. l15 The reference cited shows that firn thickness can exceed 100 m. Thank you, this has been changed accordingly.

4. l17 – 19 "Firstly, firn depth and density estimates are required to convert altimetry observed volume-to-mass changes, which remains a major source of uncertainty in mass balance studies": please also reference the work of Morris and Wingham (2015). Thank you for noticing this, we have included the reference.

5. l27 – 29 Please reference the work of Arthern and Wingham (1998). This reference has been included as well.

6. l27 "are used as measures of its dynamics and mass balance": clarify and rephrase. Thank you for this recommendation, this has been changed to: "Changes of the AIS surface elevation are an expression of changes in the firn layer and of dynamical change of the underlying ice and bedrock."

7. l30 Change "in a mass and density component" to "in a mass- and density-change component". Thank you for this suggestion, we have changed this accordingly.

8. l32 – l37 Is this paragraph only about ice shelves. If yes, this should be specified in the first sentence. If not, the reference to ice shelf hydrofracturing is confusing. We have rephrased this into: "In a warmer future climate enhanced firn compaction, melt, and refreezing can all potentially lead to firn air depletion, thereby limiting the meltwater storage capacity of the firn. This is especially important over the ice shelves, where meltwater accumulation can lead to hydrofracturing-induced ice shelf collapse (Munneke et al. 2014; Datta et al. 2019) thereby accelerating future Antarctic mass loss and sea level rise (Gilbert et al. 2021)."

9. l33 I do not believe that there is evidence of "reduced accumulation" for a warmer future climate in Antarctica. We agree. This has been omitted.

10. l33 Use "potentially lead to". This has been changed.

11. l40 Please reference the work of Medley et al. (2015). This reference has been added.

12. l44 "Firn models can roughly be divided in two classes: physically based and semi-empirical models.": please reference the work of Morris and Wingham (2011). We do not see how this paper is related to this sentence, therefore we have not included it.

13. l46 It is not clear to me what "which" refers to. We clarified this: "use a larger degree of tuning to observations representative of the past or present climate, while these conditions might not be representative of a future warming climate."

14. l47 Change "less poorly known parameters" to "a smaller number of poorly constrained parameters". We have changed this.

15. l49 Please reference the work of. The reference is missing in the reviewers' comment.

16. l56 Start the sentence as: "This study shows that, at the basin scale, (…)". Thank you, done.

17. l58 Add comma: "climatic conditions, firn densification". The comma has been added.

18. l63 Specify the sort of "field measurements". We have specified that this is about in situ firn core measurements.

19. Table 1 In the column "Other", please refer to Equation numbers. Good idea. We have included this.

20. l75 "FDM V1.2G" has not been defined yet. Maybe, refer earlier to Table 1. Thank you for noticing this, we changed this.

21. l77 "comparable": with respect to what metric? This sentence will be omitted.

22. Section 2.1.1 and in the remainder of the manuscript. Please do not use the same symbols for different parameters. If the authors want to preserve a connection between related parameters, I recommend the use of subscripts. Thank you for noticing this. D has been changed to E throughout the manuscript.

23. l83 Change "average" to "averages of". This has been changed.

24. l86 Cite a reference for "Snow crystal size and therefore fresh snow density indeed increase with increasing temperature.". We have included the reference of Judson and Doesken (2000).

25. l94 Specify the frequency of "instantaneous": hourly/daily/… This is a single time step of RACMO, which is 6-min, this has been added.

26. Equation 2 I believe that there should be no overbar on Ts and V10 in this equation. Agreed, we have adjusted this.

27. l101 "defined as the top 0.5 m": am I correct that the authors calibrate the density of the modeled upper 0.5 m to the density of the observed upper 0.5 m, but that the calibrated surface density is then used only for the top layer of IMAU-FDM? If so, please clarify the approach as well as the slight discrepancy between calibration and usage of the surface density parameterization.
Yes indeed. We changed this into: "The fit coefficients A, B and C in Eq. (2) for the fresh snow density are retuned to improve the fit of simulated with observed surface snow densities, defined as the top 0.5 m of the firn column, as this matches the thickness of the sampled layer. These fit coefficients are than used to simulate the fresh snow density."

28. l102 Specify the model time step here or elsewhere. The model time step is described in Section 2.1.3.

29. Section 2.1.2 All equations should be specified as two different cases for rho<550 and rho>550 kg/m$^3$
This has only been done for eq. 5, because this distinction is not made in the other equations.

30. l110 I suggest replacing "processes" by "mechanisms". This has been replaced.

31. l111 – 112 A citation is needed for this sentence. We added Herron and Langway (1980) as a reference.

32. l114 Change "turn out to depend on the accumulation rate" to "are chosen as functions of the long term mean accumulation rate". We changed this accordingly.

33. l120 Why was the power-law function not tested for MO$_{550}$? This was indeed tested, resulted in poorer results, and, therefore, was not applied. In order to keep the manuscript somewhat short, this test was not mentioned.

34. l124 Provide formulation of the thermal conductivity. We refer to Brils et al. (2022), to somewhat limit the size of the manuscript.

35. l128 Provide formulation of the irreducible water content. See comments above.

36. l129 The description of the refreezing algorithm is unclear and confusing. It suggests that no meltwater refreezes if a layer cannot accommodate all the incoming meltwater. We agree and have changed this to: "The retained meltwater refreezes when it reaches a layer where the latent heat can be released."

37. l131 Typo: remove "the". Thank you for noticing this, we have removed this.

38. l131 Specify the amount of melt. There is actually no limit, we have improved this.

39. l131 Do not use "significant" because it does not refer to statistical significance here. We agree. This part has been omitted, see comment above.

40. l134 – 135 What is the depth of the model domain? And what are the boundary conditions at the lower boundary? This is defined in Section 2.1: 3000 layers of 3 to 15 cm thickness. The lower boundary conditions for the thermal conduction are described in Section 2.3.1 of the revised manuscript.

41. l137 I believe that "total thickness" should be replaced by "total mass". You are right. Done.

42. l142 Please quantify the "minor trend". Done.

43. l142 Change "ice" to "material with $\rho > 830$ kg m$^{-3}$". This has been changed.

44. Equation 6 Format of the variables in the equation does not correspond to format of the variables in the main text. Thank you for noticing this. The format in the text has been adjusted.

45. l152 SMB units are wrong. The units should indeed be kg/m2/yr, changed.

46. l160 Change ";" to "and". Done.

47. l161 Explain briefly the notion of "upper-air relaxation". We added: "Upper-air relaxation is the indiscriminate nudging applied to the upper part of the atmosphere only."

48. l168 Split this sentence in two: "This results in an improved forcing. For example, (…)". Done.

49. l169 Remove "e.g.". Done.

50. l178 Typo: "describes". This has been changed.

51. Figure 1 As I understand it, all the firn cores are used for the sensitivity analysis. For this reason, I recommend changing the label for the grey dots in the legend to "Sensitivity analysis only". We propose to add a separate figure for the sensitivity analysis locations in Figure S1:

[Figure]

52. Figure 1 caption Replace "on top of" by "in addition to". See comment above.

53. Section 2.6 Provide a name for each sensitivity experiment (see Major comment 1). This has been done in Table 2 and 5 of the revised manuscript.

54. l197 "To improve the representation": I do not see the causal link between improvement and the rest of the sentence. This might be confusing, what we mean is that we aim to have a better representation of all climatic conditions across the ice sheet within the sensitivity analysis. This has been changed: "To improve the representativeness of the climatic conditions over the AIS, we used 10 additional locations in this analysis that are located in high accumulation or low temperature regions, as these areas are underrepresented in the 95 firn core sites (Fig. S1)."

55. l203 – 204 Change "+-" to +/-". Thank you, done.

56. l204 Does +/- correspond to the RACMO2.3p2 RMSE? If so, please specify this. This comment does not apply anymore, as we have changed the temperature uncertainty experiments.

57. l206 Specify that 30 kg m-3 corresponds to the RMSE in surface density. Same as above: We added: "The uncertainty of the fresh snow density was based on the RMSE from the evaluation with in-situ measurements"

58. l206 "Section 3.2" should be "Section 3.1". Thanks.

59. Table 2 There is no "D" parameter in Eq. (2) for FDM FS-L and FDM v1.2A. We fixed this.

60. l218 – 219 Refer to Equation numbers. We fixed this.

61. l231 Change "reduced with" to "reduced by". Thank you for noticing this.

62. l240 "a simulation of FDM v1.2A without MO fits": does that mean a simulation with the FDMv1.1 MO values? This has been changed to: "To tune the dry snow densification rate we first performed a simulation of FDM v1.2A without MO corrections."

63. l241 "less steep": specify that this is with respect to accumulation (bdot). This has been changed to: "The optimal $MO_{550}$ fit is less steep with respect to accumulation than the one in FDM v1.1p1."

64. l253 Replace "approaches zero" by "decreases asymptotically towards zero". This has been changed.

65. l255 Provide references for the accumulation rate values. Thank you, a reference to Brils et al. (2021) has been added.

66. Table 3 In the column "Version", I believe that $MO_{550}$ and $MO_{830}$ should be replaced by $z_{550}$ and $z_{830*}$. In the column "Fit", please refer to Equation numbers. Add a Bias column (see Major comment 2). Provide RMSE values for all models (see Major comment 2). Add FAC rows (see Major comment 2).
See response to major comment 2. We refer to the equation number in the fit column.

67. l258 Specify "27 km horizontal resolution". Thank you. Horizontal has been added.

68. l259 Typo: "in Figure 4". We fixed this.

69. l261 What does "calm" mean? The definition of calm is the absence of winds, therefore we assume this is clear.

70. l262 Change "On top of this" to "In addition to this". Thank you, we changed this.

71. l264 – 265 "The spatial pattern of the depths of the critical density levels $z_{550}$ and $z_{830}$ are shown in Figures 4b and c, and are roughly the inversed pattern of the surface snow density": not in melt areas, please discuss (see Major comment 3). This is also the case for melt areas, we find high surface snow density and low $z_{550}$ and $z_{830}$.

72. l266 – 267 "The patterns vary spatially across climatic regions with temperature as a primary driver and accumulation as a secondary driver.": please discuss the impact of surface melt (see Major comment 3). We agree that surface melt is also an important driver, we have changed this into: "The patterns vary spatially across climatic regions with temperature as a primary driver and accumulation and surface melt as secondary drivers." In line 268-268 we discuss the impact of surface melt: "Generally, the $z_{550}$ and $z_{830}$ values are low along the warm coastal margins due to faster densification, higher surface snow densities and surface melt."

73. l270 – 271 Please rephrase this sentence with more formal language. We propose to change this into: "The high accumulation rates in these regions result in less densified snow with depth and thus thick firn layers despite relatively high temperatures."

74. Figure 3 I believe that there is a mismatch between the numbers of points shown in the subplots, and the number of cores mentioned in Section 2.5. For example, it seems to me that there are less than 100 data points shown in Figure 3a. Also, please show scatter plots of the match between modeled and observed z830*, as well as between modelled and observed FAC. See responses to minor comment 3) Lack of clarity about the firn core dataset. We will rewrite this section. We have included FAC evaluation in Figure 3 (See figure major comment 2), and we already have $z_{830*}$ in Fig 3c.

75. l281 – 282 Change "can be used" to "must be used". It can also be done with surface elevation change in combination with SMB data. However, we have changed "can be used" to "as it is used."

76. Figure 4 caption Change "firn age of the crtical density level" to "firn age at the crtical density level". Thank you, this has been changed.

77. Figure 5 Define "peak-to-peak" in the caption. Please also include a map of average seasonal amplitude. We have included the definition of the peak-to-peak FAC. We decided not to include a map of average seasonal amplitude, as we focus on seasonal variability of surface elevation change and FAC in Section 4.3. In Section 4.2 and thus Figure 5, we only introduce the temporal variability of FAC.

78. l286 Change "most" to "more". Based on feedback from reviewer 3, we have changed this into: "contain the most air."

79. l286 Typo: "parts of the coast". Thank you, this has been changed.

80. l292 If statistical significance has not been tested for, please do not use "significant". We agree, significant has been omitted.

81. l294 Replace "closer to the mean" by "less spatially variable". Thank you, that is indeed better.

82. l303 Please remind the reader about the study period. We added: ".. during the study period (1979-2020)."

83. l303 – 304 "Large values indicate that seasonal and interannual climate variability cause large temporal variations in FAC.": this statement is not supported by the map of peak-to-peak variability, which depends only on two single values in the entire time series. In my view, peak-to-peak does not characterizes temporal variability well. We agree that peak-to-peak is no direct measure of seasonal and interannual variability, therefore we have omitted this sentence.

84. l313 Vtot does not appear in Equation 6. Maybe simply replace dh/dt by Vtot in Equation 6. We added this.

85. l320 Specify that dh/dt in summer months has contributions from sublimation and melt. We added: "which is driven solely by snowfall except for the two summer months, which also have contributions from sublimation and melt."

86. Figure 6 caption Change "indicate the standard deviations" to "indicate the inter-annual standard deviations". Thank you for this recommendation, done.

87. l329 Change "biases" to "errors". Thank you for pointing this out, we changed this.

88. -Section 4.4 In general, this section requires much more quantitative assessments (see Minor comment 1). We agree and have made this section more quantitative. E.g. we added this in the part below: "Above 2,000 m a.s.l. the relative contribution of firn densification to the total cumulative dH anomaly is larger (32 %) than below 2,000 m a. s. l., where this is only 14 %. This difference can partly be explained by the larger interannual temperature variability above 2,000 m a.s.l. (sd in annual means = 0.78 K compared to 0.48 K) and the absence of melt."

89. -l340 "cumulative surface temperature anomaly": is this the cumulative anomaly in surface temperatures from the long-term mean? Please clarify. We specified this.

90. -l341 "the seasonal firn thickness and FAC variability is driven by": please discuss why there is no one-to-one correspondence between firn thickness and FAC variability. As the FAC and firn thickness are different characteristics, we also do not expect a 1:1 comparison, therefore we have not discussed this in the manuscript.

91. -l341 Typo: "is" should be "are". Thank you for noticing this, we have changed this.

92. l343 – 344 "firn densification, despite the long time scale, reduces these snowfall-induced fluctuations by about 15 %": please clarify how this is calculated. This is calculated from the difference in standard deviation between the cumulative anomalies of Vsnow and Vsnow+vfc. We have specified this.

93. l356 Specify: "captures the strong spatial variation in firn thickness and density observed in our firn core dataset.". We changed this into: "In Sections 3.1, 3.2 and 4 we presented and evaluated the spatial variation in firn thickness and density."

94. l363 – 364 Specify: "shown in grey in Figs. 3d and 3e". We moved this to the methods (Section 2.5), and therefore we do not refer to Figures 8d and e. However, it is explained in the figure caption.

95. -Section 5.1 Please provide maps of seasonal amplitude and of discrepancy in seasonal amplitude. To keep the size of the manuscript within reasonable limits, we decided not to include figures of the seasonal amplitude in the remote sensing comparison, and to focus on long-term surface elevation trends in the altimetry comparison instead, which is less impacted by the altimetry uncertainties.

96. -l368 – 369 Provide references to support that lower seasonal variability in IMAU-FDM can be explained by altimetry errors. We agree and we refer to Nilsson et al. (2022).

97. -Figure 7 I find the color codes in this figure confusing. I suggest to show FAC in a color other than blue. We assume the reviewer means Figure 8. As we dont see which other colors would lead to an improvement, we decided to keep it as is. We made sure the colors combination used is colorblind friendly.

98. -Figure 7 caption Change the caption to: "Time series from FDM v1.2A of FAC, of the cumulative anomalies of surface temperature, of the vertical firn surface velocity, and of the separate components of the vertical velocity from Eq. (6). Time series are shown for (a) the entire ice sheet, (b) the part of the ice sheet situated above 2,000 m a.s.l. and (c) the part of the ice sheet situated below 2,000 m a.s.l.". Thank you for this recommendation, that is indeed clearer, we have changed this accordingly.

99. -Figure 8 Remove ice shelves from the maps, or show them in a separate color. The altimetry product we use does not include ice shelves, steep topography regions and the polar gap. However, we still plot FDM v1.1A and FDM v1.2A (Fig. 8a,b) for these areas, as we do not see a reason why not. Ice shelves are indicated by the grounding line, and omitted from all the plots that include RS data.

100. -Figure 8 caption I suggest changing "Maps of average surface elevation change" to "Maps of trends in surface elevation". Thank you for this recommendation, we agree and have changed this.

**101. -l384** "which yields the improvement of FDM v1.2A compared to FDM v1.1p1": I do not understand why the authors call this an "improvement". As I understand it, the residual of FDM v1.2A is calculated as altimetry minus FDM v1.2A. The residual of FDM v1.1p1 is calculated as altimetry minus FDM v1.1p1. Thus, subtracting the residuals results in FDMv1.2A minus FDMv1.1p1. In other words, it is simply the difference between both models because the altimetry term cancels out in this operation. If I misunderstand something, please clarify. Otherwise, please revise the use of "improvement" when referring to the difference between the residuals throughout the manuscript. This Figure does also include the altimetry product, however we acknowledge that this is not clearly described. The figure is calculated as the absolute residual of FDM v1.1p1 (|altimetry-FDM v1.1p1|) minus the absolute residual of FDM v1.2A (|altimetry-FDM v1.2A|).

We have explained this in the text as follows: "In Figure 8e we show the difference in altimetry agreement between FDM v1.2A and FDM v1.1p1, which is calculated by subtracting the absolute residual surface

elevation trend compared to altimetry (Fig. 8d) of FDM v1.1p1 from the similarly derived absolute residual of FDM v1.2A. The blue areas indicate an improvement in altimetry agreement of FDM v1.2A compared to FDM v1.1p1."

102. l389 Typo: "trends" should be "trend". We have adjusted this.

103. l394 – 395 Please provide a quantitative justification for why the 11 glaciers chose are "representative locations". We remove representative, and explain the reason why we selected these locations. "These locations were selected, as they cover the main distinct patterns from Figure 8, and have continuous observations."

104. l401 "(+9 % sd)": is that compared to post-2003 altimetry? Yes, we have added this: "The altimetry observations prior to 2003 exhibit relatively stronger short-term variability (+9 % sd compared to after 2003)."

105. l401 "likely related to the measurement imprecision": please provide a reference. This we found in Table 1 of Schroder et al. (2019). We will add the reference.

106. l402 "(+13 % sd)": is that compared to FDM v1.2A? Yes indeed, we have added this: "The agreement with FDM v1.2A also appears to increase after 2003, however the altimetry variability remains higher than the simulated variability (+13 % sd compared to FDM v1.2A)."

107. l402 "the altimetry variability remains higher than the simulated variability (+13 % sd)": please discuss possible reasons. Thank you for this recommendation. In the revised manuscript we have discussed possible reasons: The higher simulated variability can partly be related to altimetry errors, as these newer observations still contain noise (<11 cm) and have variations in radar penetration depth, while on the other hand, it can also partly be related to errors in the firn model.

108. Figure 9 Why do the authors use a 6-months running average? This masks out all the seasonality. I recommend using a shorter averaging window, 3 months for example. The seasonality amplitude is discussed in Section 5.1, therefore we chose a 6-month average running period in this section to focus on decadal patterns and reduce the impact of seasonal variation in radar penetration depth. We motivated this in the text: "The time series have been smoothed using a 6-monthly moving average window, in order to focus on decadal patterns and reduce the impact of seasonal variation in radar penetration depth."

109. Figure 9 caption Change "altimetry observed" to "altimetry observations". Done. Thank you.

110. Section 6 This entire section should be thoroughly reworked (see Major comment 1). Done, see major comment 1.

111. Table 4 This table should be thoroughly reworked (see Major comment 1). This has been done, and is divided between Table 2 and Table 5 in the revised manuscript, see major comment 1.

112. l430 "that our results are robust": what do the authors mean here? In contrast, I understand from the results that the MO fits are not robust to realistic uncertainties in climate forcing, and that this sensitivity induces strong discrepancies (Table 4) in FAC estimates (see Major comment 2).
See our response to comment 122. Instead of using uniform changed precipitation (-/+8%) and temperature (-/+ 1.5K) our revised manuscript we will use spatially variable perturbations in the sensitivity experiments, as a result there is a larger sensitivity of the simulated FAC (2- 4%) than with the uniform changed precipitation and temperature (<0.5%). In the revised text we now discuss the role of MO fitting in reducing the sensitivity compared to without MO fit (4-7% FAC sensitivity) and contextualise the model sensitivity.

113. l430 – 432 Repetition: "A difficulty of the data to model comparison is that (… ) but also hampers the comparison.". We propose to change this into: "Especially in the interior, both the long-term trend and seasonal variability are small compared to the error range of the altimetry observations, which explains part of the remaining discrepancy that exist there, but also hampers the comparison (Verjans et al., 2021)"

114. l434 Typo: "dependent" should be "to depend". Thank you, we changed this.

115. l440 – 441 Repetition: "in these regions (…) in these regions". We omitted the second "in these regions".

116. l438 – 440 Please reference the work of Medley and Thomas (2019). We added this reference.

117. Section Conclusions Make sure to be consistent in using past or present tense.
Thank you for this recommendation, we will be consistent with this.

118. l446 – 448 This statement of improvement is misleading, because FDMv1.1p is forced in this study with climatic forcing that was different than the climatic forcing used for its calibration (see Major comment 2).
As we explained above, I think there is a misunderstanding, FDMv1.1p2 is not used as comparison to our performance, only in Fig. 9 to show that the improvement in dH/dt is mainly due to updated climatic forcing. Regarding the $Z_{830}$ and FAC: the logarithmic $MO_{830}$ fit has been retuned with the new forcing. Also, the previous fresh snow density parameterization was taken from Kaspers et al. (2004), and therefore not calibrated with the previous forcing from FDM v1.1p1. In addition to that, we now also include FDM v1.1p1 in the surface snow density evaluation and FAC.

119. l450 Typo: "it has" should be "they have". Thank you, we have adjusted this.

120. l451 – 452 "the firn thickness and density patterns vary spatially across climatic regions with with temperature as a primary and accumulation as a secondary driver": Here, I believe that more nuance is needed. Spatial variability in firn thickness is primarily dictated by accumulation rate patterns. Also, the sensitivity tests have shown that the trend in surface elevation change is more sensitive to accumulation uncertainty than temperature uncertainty.

[Figure]

The plot above shows that temperature is a primary driver and accumulation a secondary driver for the spatial variability. Indeed for the surface elevation change, accumulation uncertainty is higher, but that is something different than what we state.

121. l455 "As variations in firn air content and firn thickness align": the meaning of this statement is not clear to me, please clarify. Thank you for this recommendation. We changed this to: "As variations in firn air content and firn thickness have a similar phase."

122. -l463 – 464 "our model in general is robust": again, what do the authors mean here?

We mean that: "In general, uncertainties in the model formulation, forcing or forcing during the spin-up cause rather small changes in simulated FAC (<5.2%). On the other hand, uncertainties in the accumulation forcing, and in the accumulation and temperature forcing during the spin-up period can lead to substantial differences in simulated surface elevation change (between 4-19 mm/yr)."

We will specify this in the revised manuscript.

**References**
Brils, M., Kuipers Munneke, P., van de Berg, W. J., & van den Broeke, M. (2022). Improved representation of the contemporary Greenland ice sheet firn layer by IMAU-FDM v1. 2G. *Geoscientific Model Development*, *15*(18), 7121-7138.

Carter, J., Leeson, A., Orr, A., Kittel, C., & van Wessem, J. M. (2022). Variability in Antarctic surface climatology across regional climate models and reanalysis datasets. *The Cryosphere*, *16*(9), 3815-3841.

Herron, M. M., & Langway, C. C. (1980). Firn densification: an empirical model. *Journal of Glaciology*, *25*(93), 373-385.

Judson, A., & Doesken, N. (2000). Density of freshly fallen snow in the central Rocky Mountains. *Bulletin of the American Meteorological Society*, *81*(7), 1577-1588.

Morris, E. M., & Wingham, D. J. (2014). Densification of polar snow: Measurements, modeling, and implications for altimetry. *Journal of Geophysical Research: Earth Surface*, *119*(2), 349-365.

Nilsson, J., Gardner, A. S., & Paolo, F. S. (2022). Elevation change of the Antarctic Ice Sheet: 1985 to 2020. *Earth System Science Data*, *14*(8), 3573-3598.

Ligtenberg, S. R. M., Helsen, M. M., & Van den Broeke, M. R. (2011). An improved semi-empirical model for the densification of Antarctic firn. *The Cryosphere*, *5*(4), 809-819.

---

## Author Comment (AC3)

Response to Reviewer 2:

First of all, we would like to thank Baptiste Vandecrux for his time for reviewing and editing this study. We are happy with the constructive feedback we received. Based on the feedback we have revised the sensitivity analysis, and improved the remote sensing evaluation, and improved the clarity of the text. We believe that the given suggestions will improve the quality of the results and manuscript in many ways.

Responses to the comments of the reviewers are written in red and citations of the manuscript are written in blue.

Kind regards,

Sanne Veldhuijsen

**General comment:**
1. The authors present a new offline run from their firn model, using both updated forcing (RACMO2.3p2) and updated firn model (IMAU-FDMv1.2A). The changes in the model are clearly described. The model output is thoroughly presented, including how it differs from previous version of the IMAU-FDM model. The output is compared to a multi-mission altimetry product over 1992-2015. The model is also run under various scenarios to test its sensitivity to uncertainty in the forcing data, model parameters, or in the choices made for the spin-up procedure. The manuscript is nicely written, and the figures are of very good quality.

However I have major concerns on the science output of the study and how it increases our understanding of the firn characteristics in Antarctica. The comparison to altimetry (although necessary and much appreciated) is non-conclusive because of the uncertainty of the altimetry product, and reveal that other, more precise, datasets should be used to evaluate the model. The sensitivity analysis shows that the firn model tuning procedure allows to fit equally well observations with different forcings. This makes it hard for other research teams to use the parametrizations developed here with any other forcing than RACMO2.3.p2. Testing the sensitivity on the spinup setting can be of broader interest for regional climate modelling community, but the study does not conclude on a best practice on this matter.

Nevertheless, the dataset it produces is of great interest for the science community, and actually has more value in itself than the science output being presented in the manuscript. This makes the manuscript perfectly suited for a data-oriented journal. For the Cryosphere, I would encourage the authors to strengthen their findings. This could be done by exploring one of these three options:

- using new and hopefully more precise observation datasets to gain more insights on what the model is doing good or bad (coffee can experiments, GPS records, IceBridge laser measurements...)

- looking in more detail at regions where the model indicates changing processes.

- detailing the sensitivity analysis so that the study can conclude in best practices that can benefit other research teams.

Maybe such scientific findings are already in the manuscript, and I have just missed them. Then I much apologize for my comment. More light should then be given to few key findings and some of the side analysis should be removed to keep the focus on the main insights. The findings should be highlighted in the abstract and in the conclusion. Once the interesting science findings properly highlighted, the manuscript will be a great candidate for publication in the Cryosphere.

Thank you for these recommendations. We appreciate all the given options. To add further relevance to our scientific findings, we add more detail to the sensitivity analysis by estimating the uncertainty in the FAC and surface elevation trend for every location on the ice sheet. We have done this by extrapolating sensitivities for specific locations to the entire ice sheet by regressing them against various climatic variables. In addition, we included analysis using spatially variable uncertainties in temperature and precipitation. Notably, the spin-up sensitivity analysis reveals large uncertainties in the surface elevation trend. As differences in accumulation, and to a lesser extent temperature, during the spin-up period (compared to 1979-2020) result in large surface elevation trends over the Antarctic Peninsula and Ellsworth Land, the altimetry product provides a reliable source of evaluation here.

Detailed comments:

2. - l.5 "observations" of what? We added: "in-situ firn core observations"

3. - l.8 "with altimetry" Please replace by "with previously published multi-mission altimetry product for the X-Y period" Thank you for this recommendation, this has been changed: "with a previously published multi-mission altimetry product for the period 2003-2015".

3. - l.9 "reasonably well" Please quantify the agreement. This sentence will be omitted in the revised manuscript. Quantification in other parts of the text is improved in the revised manuscript.

4. - l.19 Where does the 98% come from? Is it on average on the firn-covered area? At some locations, significant runoff can occur from the firn (e.g. ice slabs, perched or perennial firn aquifer regions). This value comes from Medley et al. 2022, where it is calculated from a firn model, however in the revised manuscript the value has been changed to 94%.

5. - l.63 "measurements" of what? We added: "firn core observations"

6. - l.66 "remote sensing altimetry" please replace by "a multi-mission remote sensing surface height change product for the X-Y period (REF)" We changed this into: "with a previously published multi-mission altimetry product for the period 1995-2015 (Schroder et al. 2019)."

7 - l.79 "further improved" do you use v1.2G as starting point? meaning do you use the same thermal conductivity as Brils et al. ? Yes, indeed. We propose to change this into: "We further improved upon FDM 1.2G for Antarctica to FDM v1.2A by updating .."

8. - l. 83 "Kaspers et al. (2004)" How do they define the fresh snow density in their study? How long after deposition do they consider the snow to be fresh? Or if they look at surface snow, for which depth range? Was this study in Antarctica? Thank you for this recommendation. We clarified: "FDM v1.1 used the fresh snow density parameterization of (Kaspers et al. 2004), which has been calibrated over Antarctica and yields density values that typically represent the first 0.5 m of the snowpack."

9. - l. 94 Same question as above for Lenaerts et al. (2012) We added that the parameterization has been calibrated over Antarctica. The depth range was already described.

10. - l. 108 "average surface temperature" is it a fixed, long-term average or the average for the past x years? Is it surface skin temperature or near surface air temperature? It is the long-term average surface skin temperature, we have specified this in the revised text.

11. - l.115 "z_830*" Here you mention z_830* but in the next lines you only mention MO_830. Is there a MO_830* ? I don't fully understand how z_830* is used. Please explain how MO_550 and MO_830 is used in Eq3. You are right, $MO_{830}$ should actually be $MO_{830*}$.
To explain how $z_{830*}$ is used, and to explain how $MO_{550}$ and $MO_{830}$ are used in Eq. (3) we suggest to replace lines 112-116 by:
"By comparing simulated and observed depths (m) of the 550 and 830 kg m-3 density levels (z550 and z830, respectively) Ligtenberg et al. (2011) found that Eq. (3) requires correction terms MO550 for p < 550 and MO830* for 550 < p < 830, which are defined as the ratio of modelled and observed values of z550 and z830*, where z830* = z830 – z550. The correction terms MO550 and MO830* are added as a multiplier to Eq. (3). MO550 and MO830* are chosen as functions of the long term mean accumulation rate. Ligtenberg et al. (2011) and Brils et al. (2022) used logarithmic correction functions, thus:"

12. - l. 120 What do you mean "optimize densification"? To make the model match observed firn density? Please describe explicitly what is your objective function when fitting the alpha and beta parameter. Yes, we mean making the model match observed firn density. In the revised manuscript, we discuss differences between Antarctica and Greenland in Section 2.2, therefore this sentence has been omitted.

13. - l. 126 Please mention if there is any difference from v1.2G on this point. No, there are no differences between v1.2G and FDM v1.2A on this point.

14. - l. 127 "tipping-bucket" I think it is called simply "bucket". When the retention capacity of a layer (=the bucket) is full, the excess water flows to the next layer without the bucket/layer to "tip" or empty itself. It overflows, but it does not tip. Thank you for this recommendation, we have changed this to "bucket method".

15. - l. 129-130 "if the latent heat..." Please rephrase to " and that has subfreezing temperature" or something alike. Thank you, we have changed this to: "The retained meltwater refreezes when it reaches a layer with a temperature below the freezing point."

16. - l. 134 "the reference period" Since you haven't specified it so far, maybe change to "a" reference period. Thank you for this recommendation, we have changed this accordingly.

17. - l. 142 "a minor trend" Please give its magnitude to show that it is minor. Please explain briefly its origin. Is it because ice with air bubbles continue to replace the dense (917kg m-3) bottom ice prescribed at the initiation of the spinup? Later in the manuscript, you mention that there is no trend in average surface height because you assume a steady state. Is the removal of the "minor trend" your way of prescribing steady state? I am wondering how this minor trend look compared to the trend you get when using your alternative spin-up strategy.

In the alternative spin-up strategy that is used for Greenland, which was developed after our simulations were performed, the trend in FAC between 1 January 1979 and 31 December 2020 that is allowed is 0.5 mm/yr. In our simulations, the FAC trend is 0.6 mm/yr. However, as the output is stored at a 10 day interval, we compare 10 January 1979 with 30 December 2020. This means that part of the trend is caused by this time gap in summer. By removing this minor trend we make sure that the 830-917 kg/m3 density part of the firn layer is also in steady state. We have added the magnitude of the dH/dt trend to show that it is minor and added an explanation.

"In the actual simulation after the spin-up, a minor trend (of <0.6 mm/yr averaged over the AIS) in total firn air content remains, because at the bottom of the column, ice with a density between 830 - 917 kg/m3, slowly replaces dense ice ($\rho$ = 917 kg/m3), which originates from the initialization of the firn column prior to the spin-up. This trend is removed before further analysis of the results."

18. - l.147 Here I am a bit confused by the v_ice and v_by (for ice shelves only?) terms and by the frame of reference used. Is v_ice the vertical velocity of the pore close-off (PCO) depth? Do you separate the compaction of deep firn into ice (which changes the PCO depth) from dynamical thickening or thinning of the underlying ice? Where is your height reference point? at the PCO depth? at the bottom of the ice? At sea level?

We explain this as follows: "Vice represents the downward movement of the surface by the local divergence of the ice flow, driven by the long-term vertical mass flux through the lower boundary of the firn column (density = 917 kg/m3). In a steady-state firn layer, this equals the mass flux at the upper boundary. Vice is therefore equal to the mean SMB (kgm-2) times ρi of the reference period, but of opposite sign."

If it is at PCO or bottom of the ice, then the buoyancy is outside of the system and shouldn't be included. If it is at sea-level, the isostatic rebound, and bedrock movement should also be in the equation. If you indeed placed yourself in a reference system where isostatic rebound is not important and where buoyancy is, then please define how you calculate buoyancy and ice thickness. Eq. (6) **only** describes the impact of firn and SMB processes on the firn height, therefore bedrock movement, basal melt and ice dynamical imbalance are not included, but it **does** include the impact on buoyancy of the changing firn mass, because this is an impact that the firn/SMB processes have on the firn height (compared to sea level). If we want to include buoyancy, we need to define our frame of reference at sea level.

We include the buoyancy impact only of the changing firn mass, (which is only relevant for the ice shelves, as those are floating), which does not require ice thickness estimates. We clarify: "(Vbouy) represents the vertical motion associated with the changing ice shelf draft when the mass of the firn layer changes."

19. - Section 2.5: Please use the same order as above to introduce your observations: i) Surface snow density, ii) Densification, iii) Surface height. We agree, the order has been changed.
You could even consider moving the observations' description at the end of sections 2.1.1, 2.1.2 and 2.3. So the reader finds out about the data right after it read about the fitting method or the surface height definition. For clarity we decided to keep the observation description separately.

20. -l.181 "122 firn cores" I'm guessing this includes the 104 cores used for tuning MO. Please rephrase into "In addition to the dry cores used for the tuning of MO ratios, another 18 cores that could not be considered dry were used to evaluate the model output." This has been changed into: "In addition to the dry cores used for the tuning of MO ratios, another 12 cores from areas with regular melt were used to evaluate the model output."

There still 11 profiles missing from the 125 + 8 profiles mentioned in l.174. This section has been rewritten to clarify, see our answer to comment 14 of Reviewer 3.

21. - l.183 "Montgomery" Please give the original source/reference of the profiles, along with the SUMup citation. References and characteristics of firn core data will be added as supplementary material. We refer to this in the data availability statement.

22. - l.184 "top 0.5 m" This top layer thickness does not match with the 3 cm thickness of the top layer, nor the 1h time step of the model. We also calibrate with simulated 0.5 m, which is now explained in Section 2.1: "The fit coefficients A, B and C in Eq. (2) for the fresh snow density are retuned to improve the fit of the simulated with observed surface snow densities, defined as the top 0.5 m of the firn column, as this matches the thickness of the sampled layer."

23. Figure 1: Here a bit of creativity could help to avoid symbol overlapping each other. For example, z550 and z830 could be shown by half disks, and sensitivity analysis could be shown as a box surrounding the markers. Z550 and Z830 do not overlap, as the white dots indicate that both Z550 and Z830 are available. The final figure is shown below, which we think is sufficiently clear. We propose to add an additional figure for the sensitivity analysis locations in Figure S1 (See our response to comment 25).

[Figure]

24. - l. 197 "105 observations" From the legend in Figure 1 I thought that sensitivity analysis would be only done at the gray dots. See comment above.

25. - l.199 "10 additional locations" Now I am concluding that the gray dots are sensitivity analysis locations where you have no density observation available. Please change the marker of the sensitivity analysis to a different one than the observation location and remove the gray fill to show that the marker does not mask anything underneath. See comment above.

I also see that there are 105 sensitivity analysis location while you use 133 profiles for tuning and evaluation of the model. We need to see which locations have observations but are not part of the sensitivity analysis. I am also unsure how you summarize the sensitivity analysis at these 105 locations into Table 4. Are the results averaged? How do we know that the average is not biased due to over/underrepresentation of certain climate zones in the 105 sites selected? We added the 11 locations, to better represent all the climatic conditions and avoid underrepresentation of certain climatic zones. See the Figure S1 shown below for a map and temperature and precipitation histograms of the locations. In addition, in the revised manuscript, we now expand the sensitivity to the entire ice sheet, so there should be no spatial bias in the final results. We also specify this in Table 4 (Table 5 in the revised manuscript).

[Figure]

**Fig.** Locations used in the sensitivity analysis.

26. - l. 204 "-1.5K" Does this evaluate the air temperature or the surface skin temperature (which is that actual input of IMAU-FDM, I assume?)? This is evaluated with 10-m firn temperature, which is a good approximation equals annual average surface skin temperature. However, this comment is not relevant anymore, as this analysis has been changed to distributed uncertainty in surface skin temperature based on the spread in surface skin temperature between RCM's and reanalysis datasets.

27. - l. 212-213 I thought that you had no data before 1979? what are these years corresponding to? Please update to something like "To mimic this increase in precipitation, we create a spinup in which model loops three times over the 41-year-long reference period where we decrease precipitation in the first loop by 10%, by 6.66% in the second and by 3.33% in the third". Make something similar for the temperature.
Thank you for this suggestion. We have changed this into: "To mimic a gradual increase in precipitation, we performed a spin-up experiment in which the precipitation up until the third-to-last loop (the 42 year long reference period) is decreased by 10 %, in the second-to-last by 6.66 %, and in the last by 3.33 % with respect to the mean precipitation of 1979-2020. To mimic the increase in temperature, we create a spinup in which the temperature up until the third-to-last loop is decreased by 1 K in the second-to-last by 0.66 K and in the last by 0.33 K."

28. - l.224 Using different symbols and reducing the size of markers, I believe you can display this third comparison in Figure 2a. Please mention that the FDM FS-K shown in Figure 2a are calculated with RACMO2.3p2 surface climate. In IMAU-FDMv1.1, FS-K was used but the surface climate was also different, therefore we still don't know if the fresh snow density in the new model is greater or lower than in IMAU-FDMv1.1.
Thank you for this suggestion. We added FDM FS-K as well as FDM v1.1p1 in Figure 2a to also evaluate the impact of the different forcing.

[Figure]

29. - l. 234-238 Brils et al. uses yearly temperature because it is the only type of parametrization of surface snow density in Greenland. There has not been any evaluation of the impact of wind speed in Greenland because there is simply no instantaneous and collocated measurements of temperature, wind-speed and surface snow density. In turn, people used yearly temperature, because they thought it was more robust and less model dependent. Also, a surface snow density parametrization built on simulated T and WS will only be valid when using T and WS from the same model because it accounts for potential model biases in the T and WS values. This should be mentioned somewhere. Consequently, I don't think that the difference between the parameterizations in

Greenland and Antarctica tell anything about the snow or the climate and might be just arbitrary. We agree, we suggest to change Lines 234-238 into: "The fresh snow density parameterization for Greenland used in FDM v1.2G, is only a function of yearly temperature (Brils et al. 2022), owing to a lack of co-located surface snow density, temperature and wind speed observations on the GrIS. This contrasts to the expression used here, which includes a dependency on instantaneous wind speed and temperature."

30. - l.241 "the optimal MO is less steep" Please add, at least at the beginning of the paragraph, plain word explanation of what "lower/higher MO830" mean (i.e. overestimated/underestimation modeled densities for the top/deep firn). I'm still not sure if the MO correction enhances or inhibits the densification. Thank you for pointing this out, we understand that this can be confusing. We now clarify this in Section 2.1.2 as follows: "The correction terms $MO_{550}$ and $MO_{830}$ are added as a multiplier to Eq. (3); MO values below one reduce the densification rate, values above one enhance the densification rate."

31. - l. 253 "Differences between" Please mention what are these differences. In terms of MO fits but also in general terms: Is densification faster/slower in than in Greenland or less/more responsive to accumulation or temperature? Good point. We clarified as follows: "In FDM v1.2G, the version for the Greenland ice sheet, a logarithmic $MO_{830}$ fit was applied. Dry firn cores from the GrIS cover a smaller range of average annual accumulation, 80-680 mm/yr compared to 20-960 mm/yr in Antarctica. If we only include cores from the 80-690 mm/yr accumulation range, we find a similar r-squared value for the logarithmic vs power fit (0.67 vs 68). Another difference is that in FDM v1.2G, an almost constant value of 0.67 for the $MO_{550}$ fit was found, which implies a linear correlation between the densification rate and the accumulation rate. In FDM v1.2A we do find a moderate (r=0.37, b = 0.12) correlation of $MO_{550}$ with accumulation, i.e. AIS densification rate depends more strongly on accumulation than temperature compared to the GrIS. Again, if we only include cores from the 80-690 mm/yr accumulation range this correlation weakens (r=0.08, b = 0.07)."

32. - l. 275 "Figure 4d shows the age of firn" This part could be more quantitative. What is the median/maximum firn depth, what is the median/maximum age at PCO? What is the fraction of the ice sheet covered by firn? Since you mention the age of the firn at PCO and its usefulness for ice core interpretation, could you present the age of the firn at PCO depth for ice core locations and compare to the values found in literature? Since you are trying to show that the model output can be used for this purpose, you might as well show how well it does.
Otherwise this paragraph and the figure could be removed to leave the focus on the surface height and FAC discussion.
I have compared the simulated PCO age to measured 15 delta age values from Breant et al. (2017) (See figure below). Based on that favourable comparison we added more quantitative information to this section, following your suggestions as follows: "Nearly all (>99 %) of the AIS is covered by a layer of firn (i.e. the ice fraction where average SMB is positive (observations suggest 99 %; Winther et al. (2001))."
&
"The highest z500 and z830 values (30.6 and 114.2 m, respectively) are found in the interior, where the densification rate is low due to low temperatures, and the surface snow density is low."
&
"Figure 4d shows the age of firn at the pore close off depth, here assumed to equal z830, which is on average 754 yr. The combination of strongly increasing firn depth and decreasing accumulation towards the interior leads to firn ages at close-off depth up to 3240 years in central East Antarctica, whereas the pore close-off firn age in warm and wet coastal margins can be as low as 20 yr. If we compare this to observations, we find an RMSE = 231 yr, $R^2$ = 0.985. On average RMSE is 25% of the observed values. In comparison, RMSE of $z_{830}$ is 15% of the observed values and R2 = 0.73. These simulations add to our understanding of the large spread in timescales of the firn."

[Figure]

33. - l.288 "somewhat" Please avoid this word and replace by quantitative information. We replaced it by 5 % and 7 %: "These average FAC estimates are 5 % and 7 % lower compared to values simulated with the CFM forced with MERRA-2 climate data of 24.0 and 17.0 m respectively (Medley et al. 2022)."

34. - l.289-290 "higher accumulation rates" Can you quantify this difference? We decided to remove this part, and reformulate as mentioned above.

35. - l.300 "also contributes to the general pattern..." Higher melt could indeed participate to the lower FAC at low elevations. But increased snowfall in the interior does not show in the new FAC pattern (FAC decrease in the center of the ice sheet in Fig5b). This could mean that the MO update overprints the effect of higher accumulation. That is indeed a good point, we included this in our explanation as follows: "However, in the highly elevated and low accumulation region, we find a reduction in FAC, which implies that the $MO_{830}$ update outweighs the effects of higher accumulation and lower fresh snow density in this region."

You also mention that the fresh snow density is decreased since the last version. This should have enhanced the FAC increase due to increased accumulation. Yes indeed, we also include the effect of fresh snow density.

36. - l.318 "Vice is constant" Why is it constant by definition? How do you define it? The description of $V_{ice}$ in Section 2.3 (lines 164-165) has been clarified, see our response to comment 18.

37. - l.319 "Vbuoyancy is negligible" I'm still not sure how you define this and why is it negligible? Buoyancy is simply calculated with Archimede's principle, we explain in Section 2.3: "vby represents the vertical motion associated with the changing ice shelf draft when the mass of the firn layer changes."
and: "Vby, only relevant over ice shelves, is equal to the negative change in firn mass divided by the density of sea water."

As can be seen from Figure 6a, these displacements are small. We added: ", while the AIS average impacts of snowdrift ($v_{snd}$) and buoyancy ($v_{buoy}$) are small (<1 %)."

38. - l.326 "in the timing" and magnitude? This section has been rewritten based on previous comments.

39. - l.327-328 "63 to 68%" I am not sure how you calculate this number. The lines above these lines have been rewritten. This explanation will be rephrased as: "The difference between the firn thickness and FAC seasonal amplitudes (6.8 and 4.2 cm) suggests that around 62 % of the seasonal surface elevation fluctuations are caused by a change in air content rather than actual mass change."

40. - l. 332-335 Isn't it redundant to the discussion of Figure 5c? Could be moved elsewhere or removed for concision. See typo "ànd". In Fig. 5c we only include the peak-to-peak variability, "The high peak-to-peak variability (on ice shelves) is caused by a combination of temporal variability in accumulation and melt." In Fig. 6 we describe the seasonal peak-to-peak variability in more detail, which is caused by high snowfall and melt, and not necessarily temporal variability in accumulation and melt.

41. -l.335-338 Does not add information. Consider removing for concision. We decided to maintain this figure. Differences in timing of the firn height maximum contributes to our understanding about the spatial variations in seasonal cycles and hence the interpretation of satellite altimetry. We also refer to this figure to explain the difference between the ice sheet-wide seasonal amplitude and the average of all local seasonal amplitudes.

42. - l.345 "5 to 10 years" Why does it matter that it last 5-10 years? How does that relate to frequency of El-Niño or annular mode? That is an observation from Figure 7. It does not directly relate to frequency of SAM (weeks) or El-Nino (3-7 years). A lot is still unknown about decadal variability in El nino and SAM. Therefore, we decided to remove this sentence.

43. - l.340 "Above 2000 m ..." I don't fully agree with this sentence. The variability in V_fc is much lower than below 2000 m. a.s.l. "larger temperature variability" than what/where? We agree that this was not clear. We have rewritten this: "Above 2,000 m a.s.l. the relative contribution of firn densification to the total cumulative dH anomaly is larger (32 %) than below 2,000 m a. s. l., where this is only 14 %. This difference can partly be explained by the larger interannual temperature variability above 2,000 m a.s.l. (sd in annual means = 0.78 K compared to 0.48 K) and the absence of melt."

44. - l.352 "sd" Spell out what it stands for the first time you use it . This has been added.

45. - l.353 "67%" It sounds like a repetition from l.327-328, but calculated slightly differently. Maybe keep only one of the two formulations. Lines 327-328 is only about the seasonal variability. This number is about the decadal variation.

46. - l.358-364 This is methods. We agree, and moved this part to the methods (Section 2.5; Observational data).

47. - l.366 "Ligtenberg et al. ..." Please present your results first, then bring in previous simulations to put your results in perspective. Thank you for this recommendation. This is a good point. We will first present our own results.

48. - l.368 "likely explained by altimetry errors" This deserves more details. What was the motivation to say that? What altimetry errors was it?
To comply, we propose to replace "altimetry errors" by "seasonal variation in radar penetration depths, as these may amount to up to several centimeters in amplitude (Nillson et al. 2022)."

49. - l.369 "The altimetry observations prior to 2003..." This should be moved to the method and further justified. What is the measurement precision threshold that you use? The measurement precision threshold is highly variable in space, mainly depending on the surface slope, therefore we don't use a single threshold. Instead, Table 1 in Schroder et al. 2019 shows that the satellites from 2003 onwards have a higher precision, where the noise level is a function of the slope. We do include 1992-2003 in Figure 9. Therefore, we propose to add this to the method section:
"We will focus on the observations from 2003 onwards, as satellites before that period have a lower measurement precision (Schroder et al. 2019)."

50. - l.374-376 Since you mention that the improvement is due to better melt forcing in the coastal margin, can you present the same seasonal surface height amplitudes for the altimetry product and two model runs below and above 2000 masl (or any elevation that is relevant). This would provide the evidence that the improvement comes from these coastal regions. The increased melt of the forcing dataset is described by Van Wessem et al. (2018). Since melt almost exclusively occurs in the summer months, this obviously results in a higher seasonal amplitude. However, not only the increase in melt impacts the seasonal amplitude. Also, the fresh snow density over the AIS is lower, and there is increased accumulation in the interior. Therefore, we only find a slightly larger increase below 1,000 m a.s.l. than above 1,000 m a.s.l. (13% compared to 10%), which is therefore not included in the revised manuscript.

51. - l.379 "linear regression" Please give more detail: a linear regression is fitted to...annual/monthly/daily values... for the period ... Changed accordingly.

52. - l.385 "excluded" I understand that high dynamical imbalance regions should be masked out of the altimetry product. But there is no reason to remove it from the FDM to FDM comparison in Fig 8e. This Figure does also include the altimetry product, however we acknowledge that this was not clearly described. The figure is calculated as the absolute residual of FDM v1.1p1 (|altimetry-FDM v1.1p1|) minus the absolute residual of FDM v1.2A (|altimetry-FDM v1.2A|). We have explained this in the text as follows: "In Figure 8e we show the difference in altimetry agreement between FDM v1.2A and FDM v1.1p1, which is calculated by subtracting the absolute residual surface elevation trend compared to altimetry (Fig. 8d) of FDM v1.1p1 from the similarly derived absolute residual of FDM v1.2A. The blue areas indicate an improvement in altimetry agreement of FDM v1.2A compared to FDM v1.1p1."

And in the figure label: "Difference in altimetry agreement between FDM v1.2A and FDM v1.1p1".
And in the figure caption: "(e) Difference in altimetry agreement between FDM v1.2A and FDM v1.1p1. (absolute residual of FDM v1.1p1 compared to altimetry minus absolute residual of FDM v1.2A compared to altimetry)."

53. - l.385-388 Please merge these two sentences and show that the "17%" is what make you say that there is improvement.
We changed this as follows: "The average absolute residual trend of FDM v1.2A compared to altimetry has been reduced by 17 % compared to FDM v1.1p1 (from 2.6 to 2.1 cm/yr). The improvement is most notable in

Dronning Maud Land, Wilkins Land and Adélie Land. In Dronning Maud Land the FDM v1.2A trend has become more positive, and in Wilkins Land and Adélie Land the trend has become less negative (Fig. 8b,c, for region names see Fig. 1)."

"especially in Dronning Maud Land, Wilkins Land and Adelie Land, where the FDM v1.2A trend has become either more positive or less negative." The regions are not indicated on any maps and you don't mention which ones became "more positive or more negative", neither why the simulated trend became more positive or more negative in these areas. Either give a proper description or remove .
This has been improved, see comment above.

54. - l.388-393 Please explicit how you rule out long term trend in ice dynamic thickening?
That is a good point, we propose to change this into: "In these regions, a substantial residual positive trend remains (>10 cm yr-1) (Fig. 8d), which can either be due to an increase in accumulation or ice dynamical thickening. Smith et al. (2020) found that the mass gain is mainly located along the steep slopes of the Antarctic Peninsula and decreases with distance from the ocean, which is indicative of an increase in snow accumulation. The increase in accumulation on centennial time scales in these regions (e.g., Thomas et al., 2015, 2017) implies that the actual firn column is not in balance with the 1979-2020 climate, as assumed in our spinup. As a result, the vertical downward ice flow (Eq. 6) is overestimated, which results in underestimated surface elevation change. This is confirmed by the results from the spin-up sensitivity analysis, in which the precipitation during the spin-up period was reduced, as indicated by the red shaded areas in Fig. 9."

55. - l.394-402 After briefly introducing Fig9, you end up discussing residual trends (l. 399). You might as well refer to Fig8d (which should be annotated with the names of the places mentioned in the text).
We agree. When discussing the residual trends, we will refer to Figure 8d.

Eventually in this paragraph you only mention location 5, 1 and 6. Which means that most of Figure 9 is not described in the text. The only piece of information given by Figure 9 is this variability in the altimetry product. It should be described and discussed better in the previous paragraph, where this variation actually matters. In that paragraph (Section 5.1) please give a metric for the this change in inter-annual variability: please show that on the majority of the ice sheet the standard deviation of detrended altimetry height change is significantly higher before 2003 than after. I suggest removing figure 9 and merging this paragraph with the two previous ones.
We will discuss the short-term variability of the altimetry product in Section 5.1, including the quantification in terms of standard deviation. We would like to keep Figure 9, as it shows temporal detail of the surface elevation time series. It also allows us to show the similarity FDM v1.2A and FDM v1.1p2, as we only have simulations of this model version on a limited amount of locations. Based on feedback from Reviewer 1, we also decided to include uncertainty bands in Figure 9. This shows whether the offset with altimetry trend falls within the uncertainty bands of IMAU-FDM for several locations.

56. - l.401 What is the proposed explanation for this increased variability before 2003? Could it be the frequency of the altimeter that was more sensitive to change in penetration into snow? The frequency of ENVISAT and ERS-1/ERS-2 is the same. We clarify in the revised manuscript, as follows: "as satellites before that period have a lower measurement precision (Schroder et al. (2019)."

57. - l.409 "+/-6%" Is this the temperature uncertainty or the magnitude of the FAC change? See my suggestions for Table 4 to avoid misunderstanding. We have redesigned Table 4, where we also resolve this confusion. See our response to major comment 1a of Reviewer 1.

58 - l.409 Can you be more specific than "somewhat increase"? This part will be omitted.

59. - l.410 "robust" What do you mean by robust?
I would rather say that the MO fitting allows to produce realistic firn densities even if an arbitrary bias is applied to the forcing. It is also important to mention here whether the RMSEs showed here are from the measurements that are also used for the fitting of MO or if they are independent. If they are the one the MOs are being fitted to, then it is normal that they remain low after re-fitting the MOs. See comment 122 of Reviewer 1. As said above, we will focus on the sensitivity on FAC and surface elevation changes. Instead of using uniform changed precipitation (-/+8%) and temperature (-/+ 1.5K) in our revised manuscript we will use spatially variable perturbations in the sensitivity experiments, as a result there is a larger sensitivity of the simulated FAC (2- 4%) than with the uniformly changed precipitation and temperature (<0.5%). We will discuss the role of MO fitting

in reducing the sensitivity compared to without MO fit (4-7% FAC sensitivity) and contextualize the model sensitivity.

60. - l.411 This shows the limitation of using the entire period as a reference period for spinup.
Since no obvious strong long-term trends have been detected in Antarctica's surface climate and SMB during the period 1979-2020, there is no reason to choose a shorter period. The reason for this is that you want to capture the climate of an as long as possible period. If you take less than 42 years, you have a bigger chance that you are in an anomalous climatic period.

61. - l.416 "yields an average surface elevation trend" Does that help to get closer to the altimetry results in some areas? This is an interesting point, which we decided to explore further. We found that it reduces the residual surface elevation change over the Antarctic Peninsula and Ellsworth Land with 25% and with 38% if we only include firn regions whereby the age of PCO is >42 years (See figures below).

[Figure]

Locations with PCO <42 yr show no increasing surface elevation trend as the firn from before 1979 has already been refreshed. We will include this analysis in the revised manuscript.

62. - Table 4. Please add the magnitude of the change applied in each run (not just the sign).
Are the relative changes in FAC and RMSEs in m or in %?
Is the FAC for all the ice sheet or the average change at the density profile locations?
The design of this table will be improved in the revised manuscript to avoid these confusions. See our response to major comment 1a of Reviewer 1.

63. - l. 417 "linear regression" I am wondering if this linear regression is necessary, since you have access to the model run. Could it be rephrased into "We notice that this trend (+3.3 cmyr-1) is more pronounced in high accumulation areas and find that, in this scenario, the average height change of areas with accumulation >1000mm yr-1 is 5.x m over the 1979-2020 period." ? We have extended our sensitivity analysis, and this part will be removed.

Once again, would this scenario get the model closer to the altimetry observations? Do you deem it more realistic than the reference run? See our response to comment 61.

64. - l.426 "partly explains" Please be quantitative. What was the improvement given by the inclusion of accumulation and temperature trends in the spinup? How much of the observed surface height trend is left unexplained even in this scenario? See our response to comment 61.

65. - l.430 I'm not sure what you mean by "robust". Please spell out. See our response to comment 122 of Reviewer 1.

66. - l.431 "error range of altimetry observations" The high uncertainty on the altimetry product (variability before 2003, uncorrected penetration, high relative uncertainty in the interior) poses the question of the suitability of this altimetry dataset for the evaluation of the firn model. There are many other datasets available (coffee can compaction experiments, multi-year gps records, ice bridge flights and as you mention ICESat 1 and 2 which have no penetration issues).

My main concern is that since the only observations presented in the manuscript are either used to tune the model (density profiles) or are too uncertain to truly assess the quality of the model (altimetry), I am wondering what knowledge we gain about the model or the processes it tries to describe. Where is its weakness? Where should it be improved next?
To partly overcome these concerns, we included 10-fold cross evaluations for surface snow density, $z_{550}$ and $z_{830}$. Besides that, we also include FAC evaluation, and focus on the altimetry comparison on 2003-2015.

We acknowledge that the remote sensing dataset has substantial uncertainties, but it is the only dataset available that has a large spatial and temporal coverage. We have looked into other datasets, but they also have their disadvantages. ICESat-1 has poor spatial-temporal coverage over Antarctica, and ICESat-2 overlaps only 2 years and 2 months with our dataset, and it does not overlap with our FDM v1.1p1 simulation. IceBridge is mainly available in ice dynamical imbalance regions, which also makes the comparison difficult. Also, in this region the decadal and seasonal signals are relatively large (and thus less sensitive to the altimetry errors), which gives us confidence to use the dataset of Schroder et al. (2019).

We will articulate what knowledge we learn from this study in terms of model weakness. In the sensitivity analysis and remote sensing comparison we show that the spin-up can be a substantial model weakness, which provides an opportunity to improve the model next.

67. - l.438-442 This is a repetition of what was said before. Could be removed or merged with the conclusion. This section will be rewritten, to avoid repetition, and partly moved to the conclusions.

68. - l.463 please replaced "somewhat improved" by a more quantitative "improved by X%" We agree and we will include quantification throughout the manuscript. This paragraph will also be rewritten.

69. - l.464 Please spell out what you mean by robust? See our response to comment 122 of Reviewer 1.

70. - l.465 This last statement is rather loose and arbitrary. You could move your outlooks to the end of the conclusion to finish on future perspectives. We agree. We have removed this sentence, and we will rewrite this section.

71. - Code and data availability: Note that the Cryosphere is in favor of an open code and open data science: "We recommend that any data set used in your manuscript is submitted to a reliable data repository and linked from your manuscript through a DOI. Please see our data policy. Please also consider other assets like software & model code or video supplements." (TC instruction to authors) I know that it is something that has been done before by your research group and it would be great to continue in that direction. As a minimum, the scripts to reproduce the analysis and the figures should be made available to reproduce the study's results from the model output. Please also mention where to find the density profiles that are not in SumUp and the altimetry data. References and characteristics of firn core data will be added as supplementary material. We refer to this in the data availability statement. The code of FDM v1.2A will be shared on Github. IMAU-FDM data can be obtained from the authors without conditions.

**References**
Brils, M., Kuipers Munneke, P., van de Berg, W. J., & van den Broeke, M. (2022). Improved representation of the contemporary Greenland ice sheet firn layer by IMAU-FDM v1. 2G. *Geoscientific Model Development*, *15*(18), 7121-7138.

Medley, B., Neumann, T. A., Zwally, H. J., Smith, B. E., & Stevens, C. M. (2022). Simulations of firn processes over the Greenland and Antarctic ice sheets: 1980–2021. *The Cryosphere*, *16*(10), 3971-4011.

Nilsson, J., Gardner, A. S., & Paolo, F. S. (2022). Elevation change of the Antarctic Ice Sheet: 1985 to 2020. *Earth System Science Data*, *14*(8), 3573-3598.

Van Wessem, J. M., Van De Berg, W. J., Noël, B. P., Van Meijgaard, E., Amory, C., Birnbaum, G., ... & Van Den Broeke, M. R. (2018). Modelling the climate and surface mass balance of polar ice sheets using RACMO2– Part 2: Antarctica (1979–2016). *The Cryosphere*, *12*(4), 1479-1498.

---

## Referee Report (RR1)

Review of
*Characteristics of the contemporary Antarctic firn layer simulated with IMAU-FDM v1.2A (1979-2020)*
by Veldhuisen et al.

Reviewer: Vincent Verjans

This study presents a modeling approach to simulate the evolution of the Antarctic firn layer. The manuscript reviewed is the second iteration of the reviewing process.
Firstly, I commend the authors for their thorough work on addressing the comments from the other reviewers and me. I appreciate their efforts, and I find this revised version much improved compared to the initial manuscript. In particular, I find the sensitivity analysis and the ice-sheet-wide uncertainty estimates much better than previously. In my view, this study now has the scientific quality to be published in The Cryosphere. However, I still believe that some sections of the text should be clarified and/or better explained. I address these concerns in this review.

My review is separated in one Minor comment, and Specific line-by-line comments. Please note that line numbers are with respect to the updated manuscript without tracked changes.

Minor comment

1) Section 6 still needs some work, and the last two paragraphs in particular. I have read this section several times, and I still do not understand how the statements presented can be derived from the results. I specify these statements here:

-(l516-517) "*In non-spin-up experiments, the uncertainty is only 4 % of the total surface elevation change.*"
Is this 4% value calculated at the end of the time series? Or averaged over the time series? Or something else?

-(l517) "*This uncertainty is smaller than the 40 % that can be derived from Table 3.*"
I do not understand how this can be derived from Table 5 (note that the Table number is wrong in the text). Can the authors please clarify how they calculate the 40% value?

-(l524-525) "*It indicates that imposing an uncertainty in the experiments mostly has an impact on the high-density, low-FAC part of the firn column.*"
I understand that the uncertainty in FAC is lower than in dH/dt. However, I disagree that this necessarily implies that mostly high-density firn is affected by uncertainties here. For example, density could be higher at all vertical levels (due to higher temperatures) but compensated by a larger total firn column thickness (due to higher accumulation). This can result in similar FAC values (i.e., low FAC uncertainty) but with density values affected throughout the firn column. Therefore, the authors should support this statement quantitatively, and using extra figures in the supplements that show that low-density firn is unaffected by the sensitivity tests.

-(l525-526) "*The spatial pattern in FAC change uncertainty for the spin-up experiments is different from that of surface elevation uncertainty (Fig. S6)*"
This is not clear to me when I compare Fig. 10b and Fig. S6. In contrast, I find the patterns generally similar.

-(l527-528) "*We attribute this to the fact that in high-accumulation regions, the imposed uncertainty has an effect on the entire firn layer much more quickly.*" Why is that? Can the authors clarify how they reach this conclusion?

-(l533-534) "*It suggests that the ice-sheet-wide uncertainty due to uncertainty in the spin-up climate is lower than suggested in Figure S7c.*"
Do the authors mean that their uncertainty estimates provided in Table 2 overestimate the actual uncertainty? In this case, this contradicts the references that are provided in Table 2.

Specific line-by-line comments
-Title
I suggest: "Characteristics of the 1979-2020 Antarctic firn layer simulated with IMAU-FDM v1.2A"
-l2
Here and elsewhere in the manuscript, I suggest replacing "*ice thickness*" by "ice mass".
-l6-7
"*seasonal and decadal surface height variability is due to variations*": replace is by are.
-l7
Add: "firn mass, respectively".
-l34-35
Replace: "in mass- and density-change components".
-l35
Correct: "an ice-dynamical"
-l37
Add comma: "climate, enhanced"
-l52
Replace "*while*" by "but".
-l72
Replace: "characteristics".
-Table 1
Include FDM v1.2G
-l108
Replace "*depositing*" by "deposition".
-l113
Please specify: "to simulate the fresh snow density, here applied to the top-most model layer (typically xx cm thick)" OR "to simulate the fresh snow density, here applied to the top 0.5 m of the model domain", depending on the modeling procedure.
-l114
Replace "*include*" by "perform".
-l123
Replace "*measure*" by "proxy".
-l124
Replace "*assuming that the accumulation rate is constant*" by "approximating the accumulation rate as constant".
-l128
Replace "*by deformation, sublimation and diffusion*" by "by deformation, recrystallization and molecular diffusion".
-l141
Replace "*include*" by "perform".
-l151
Is it only retained meltwater that is allowed to refreeze? I believe that meltwater reaching a subfreezing layer is allowed to refreeze even if it is not retained through the irreducible water content before.
-l187
Replace "*retuned cloud scheme and snow properties*" by "retuned schemes for clouds and for snow properties".
-Section 2.5
Is it possible to describe better how the separation between calibration and evaluation cores was decided?
-l214
I think that "*method and citation*" should be plural.
-l221
Add comma: "15 cm, depending".
-l230
Typo: "is currently too short".

-l242
"*excluding five sites that were later added for evaluation*": this is not clear. Is it meant that five additional sites were later added for evaluation?
-l247
Replace "*a model sensitivity test, which includes*" by "a range of model sensitivity tests, which include".
-l248
Is the fresh snow density varied independently at each time step?
-l252-254
Similar question: are the temperature variations applied independently at each time step?
-l257
Change: "isotope".
-l258-259
A citation is needed for this statement.
-l259
Replace "*about typically*" by "approximately".
-l268
Please specify "we also performed two tests".
-l272
Replace "*a steady-state firn layer*" by "a firn layer in steady-state".
-l272
Replace "*this simulation*" by "these simulations".
-l274
Specify: "and/or temperature".
-l275
Why "*randomly*"? Please clarify.
-l278
Replace "*included*" by "shown".
-l287
Rephrase: "extent (bias = 10.3 kg m−3 ). It is the update in the atmospheric forcing that causes a poorer (…)".
-l296
If this is correct, please specify: "is reduced by 18 kg m-3 on average".
-l299-301
In my view, these two sentences can be removed.
-l302-303
How can do the MO values influence the rho0 parameterization? Is it not the other way around?
-Table 3
The R2 values do not agree with the R2 values of Fig. 2.
-Table 4
First, MO coefficients have no units, and thus the last column should be unitless. Second, RMSE should be replaced by RMSD (root mean squared deviation) because this quantifies the deviations with respect to the fitted function, and not errors. Finally, RMSD values should be given in all rows.
-l316
The comparison with FDM v1.1p1 and FDM v1.1p2 is unfair. These models were not calibrated with the data of this study, whereas FDM v1.2A was calibrated with (almost) all the data used for evaluation. This must be specified.
-l324
Correct: "asymptotically".
-l324
To avoid any confusion, please specify: "asymptotically towards zero, but less rapidly than  FDM v1.2A-log".
-l330
Replace "*68*" by "0.68".

-l348-349
Please quantify: " *The patterns vary spatially across climatic regions with temperature as a primary driver and accumulation and surface melt as secondary drivers.*"
-l353
Please specify: "in younger, thus less densified".
-l360
"*If we compare this to observations*": which observations give firn age?
-l360-361
"*On average, the RMSE is 25 %*": is this really the RMSE (i.e., the mean across all the errors) or rather the average ratio between the individual absolute errors and their corresponding observed value?
-l361
"*In comparison, RMSE of z830 is 15 % of the observed values*": same question as comment just above.
-l366
Replace "*would be*" by "was".
-l371
Specify: "Community Firn Model".
-l375
Replace both instances of "*new*" by "updated".
-l378
I think that "*FDMv1.2A*" should be replaced by "FDMv1.1".
-l383
Correct: "contribute".
-l387
Correct: "values".
-Section 4.3
Is it possible to provide maps of mean accumulation, melt, and sublimation in the Supplementary Information?
-l415
" *as the annual average accumulation is low*": I do not understand why this causes low seasonal amplitude.
-l421
Replace "*its components*" by "the components of the latter".
-l424
Replace "*long time scale*" by "slower response timescale".
-l426
Typo: vfc is all in subscript.
-l430-432
Note that this difference is also explained by the high sensitivity of firn compaction to temperature variability for firn at low temperatures.
-l432
Correct: "variabilities".
-l435
Specify: "total surface elevation change variability".
-l451
Please remove "*which compares well with the FDM seasonal amplitudes*", and replace it by a new sentence: "This suggests that our results of AIS-wide seasonal amplitude lie within the range of observational uncertainty."
-l480
Remove "*relatively*".
-l482:
Typo: "reduced".
-l484-485
Break the sentence: "depth (Schröder et al., 2019). On the other hand,".
-l487 vs. l490
Make sure to be consistent in using brackets or not around the location numbers.

-l495
If this is correct, specify: "are on average 41% lower".
-l497-498
Replace "*is needed*" by "we used".
-l498
Refer to Fig. S3.
-l505
Typo: "a surface elevation".
-l506
Typo: "*of of*".
-l507
I do not understand "*0.7/0.8*".
-Table 5
Specify periods in the caption: 2015-2020 for dH/dt, and averaged over 1979-2020 for FAC I think.
-l517
Replace "*Table 3*" by "Table 5".
-l518
Replace "*amplifying*" by "compensating".
-Figure 10
Specify periods in the caption: 2015-2020 for dH/dt, and averaged over 1979-2020 for FAC I think.
-l519
Replace "*Table 3*" by "Table 5".
-l522
Replace this sentence by "We expand ice-sheet-wide averaged estimates of uncertainty into time series, showin in Figure S7c."
-l529-530
Replace here: "The finding that imposing an uncertainty in spin-up climate has low impact on FAC uncertainty". (see my Minor comment).
-l537
Replace "*minus the spin-up sensitivity test*" by "minus the transient run spun-up with the sensitivity test".
-l539
"*we assume the underlying ice to have responded to the changing climatic conditions*": does this mean that advection if ice at the lower boundary has changed? And is this equivalent to saying that v_ice cannot be assumed constant?
-Section 7
I would appreciate adding a couple of sentences about larger uncertainties in melt-affected areas. Can the authors also provide the quantitative results of the evaluation at the 10 wet cores?
-l543
Specify: "has been used over the AIS".
-l544
Correct format of citation.
-l546
Specify: "the mass change associated with surface processes".
-l548
Correct: "updated climatic forcing, as is shown".
-l553:
Specify: " compared to the error range of the altimetry observations and model uncertainties".
-l558
Replace "*decadal*" by "inter-annual".
-l561
Remove comma after "*factor*".
-l565
Replace: "not recommended in combination".

-l577-578
See my comment on l316.
-l582
Phrasing: "as a primary, and accumulation and surface melt as secondary drivers."
-l585
Replace "*As variations in FAC and firn thickness have a similar phase, 63 to 68 %*" by "Variations in FAC and firn thickness have a similar phase, and 63 to 68 %".
-l591
Replace "*which violates*" by "violating".
-Caption of Figure S4
(h) should be for the accumulation spin-up experiments.
-Caption of Figure S5
(h) should be for the accumulation spin-up experiments.
-Caption of Figure S7
Specify: "the estimated ice-sheet-wide averaged surface elevation change".

---

## Referee Report (RR2)

Review of revised:
"Characteristics of the contemporary Antarctic firn layer simulated with IMAU-FDM v1.2A (1979-2020)"
Sanne B.M. Veldhuijsen, Willem Jan van de Berg, Max Brils, Peter Kuipers Munneke, and Michiel R. van den Broeke

Reviewer: C. Max Stevens

**General comments:**

I thank the authors for their thorough responses to my and the other reviewers' comments on the previous draft of their manuscript. The edits have made this paper substantially better. Most notably, the authors have taken care to add much more quantitative analyses of their model results and added context in which to interpret those numbers. Additionally, the clarity of the paper improved dramatically. The paper now presents both an update to the IMAU firn densification model and an impressive suite of model experiments and sensitivity tests. These results will be helpful for those looking to quantify uncertainty in firn model applications. I am happy to recommend the paper for publication in The Cryosphere after the authors address a few lingering issues. These are listed below in a line-by-line fashion, and I believe most are small changes that will improve clarity.

The largest issue that must be addressed is the definition of the "AIS" (Apologies to the authors that I did not catch this in the first review). I think the authors are using the term 'AIS' as shorthand for the entire continent, including both the ice sheets (WAIS and EAIS), the ice shelves, and distinct glaciers and ice fields on the Antarctic Peninsula. Strictly, the term AIS refers only to the ice sheets, and the authors define AIS as the Antarctic Ice Sheet in the first sentence of the paper. But, then throughout the paper it seems that the analyses are including ice shelves and the peninsula. This leads to some confusion in the paper. For example, the authors mention on line 344 that 99% of the AIS has a firn layer. Is this figure strictly for the ice sheets, or is the peninsula, which contains distinct glaciers, included in the 99%? How about the ice shelves? I think it would be good to specifically define the area(s) included in the study. For example, "We use the term AIS broadly to include the East and West Antarctic Ice sheets, as well as distinct glaciers and ice fields on the continent and the ice shelves".

**Line by line comments:**
(line numbers should refer to main manuscript, not the tracked changes version, but I may have made mistakes with that a few times.)

13: is losing → has been losing

37: comma after climate

63: change 'this study' to 'that study' to clarify that you are not talking about your own paper.

130: consider specifying 'requires dimensionless correction terms'

Section 2.3: Consider mentioning in this section that v_bedrock also contributes to ice-sheet elevation change but is not included in IMAU-FDM.

305: clarify 'without MO corrections' – does that mean that you just used the Arthern et al. (2010) equation, i.e. Equation 3 in your paper? Or that you ran the model using the updated RACMO forcing but the MO values from Ligtenberg et al. 2011? (Table 4 does not make it clearer to me).

315/Table 4: I got a bit confused here because you refer to FDM v1.1p2, which is not listed in Table 4. From Table 1, I gather that v1.1p2 is the model using RAMCO2.3p2 and the Ligtenberg MO values. To clarify this section, it may be as simple as altering the first and third lines of Table 4 to read, "MO$_{550}$FDM v1.1p1/v1.1p2" and adding a second $R^2$ value. Then, add a note in the caption explaining the notation and reminding the reader that v1.1p2 uses the RACMO2.3.p2. Alternatively, you could add 2 more lines to the table. As a reader, I was alternating between the text and Table 4 while reading this section, and so I was a bit thrown off when I had to go searching to remind me what v1.1p2 was. I realize there is a risk of redundancy (ie., you did define v1.1p2 earlier), but given the number of model variations that are included in your study I as a reader find it helpful to have reminders throughout the text of what the variations are.

325: Similar to my last 2 comments – I am again confused by your naming scheme. Which is the 'reference FDM v1.2A'? That is not included in Table 1 or Table 4. If one of the runs you are referring to is the -log run, refer to it as you have defined it previously.

329: Specify: "if we only include Antarctic cores from the…"

329: change to, "we find similar model fits to the data for the logarithmic model ($R^2$=0.67) and the power model ($R^2$=0.68)."

330: Change to 'Another difference between FDM v1.2G and v1.2A is that in …'

337: Coming from the previous section that discussed many model variations, it would be useful here to specify, "Using the improved firn model FDM v.1.2A, we simulate…" This will make it very clear to the reader what to refer to in Tables 1 and 4.

344: specify: "In line with observations, FDM v1.2A predicts that nearly all…"

352: remove word 'especially'

361-362: The sentence about the large spread of timescales in the firn is vague – either add more detail of how these add to our understanding or remove this sentence. I think the spread in firn ages in Antarctica is reasonably well documented in the ice-core literature. I am not sure

if you are implying that your findings help with process understanding (e.g., processes operating on different timescales) or implying that your statistics (25% vs 15% RMSE for firn age and depth) can provide new insights – if so, what are those?

371: Define CFM; please cite

373: Related to comment above – make sure each instance of referring to AIS is consistent or explained. Here – did the values in the previous paragraph not include ice shelves?

376: spatial → spatially

420: This sentence needs punctuation and/or to be broken into multiple sentences to clarify. Also consider referring to figure panels in the sentence. Something like: Figure 7 shows time series of the entire AIS (7a) and above/below 2000 m (7b/7c). Each panel includes integrated FAC and the cumulative surface temperature and surface velocity anomalies. The surface velocity anomalies are broken down into their components.

487: change to plural: "find substantial positive residual-altimetry trends at locations"

494: updated MO fits – perhaps change language in this section to refer to the model versions, e.g. v1.2A, v1.1p2, etc. Then the reader can refer directly to tables to remember which one you are referring to.

518: 'increases the surface elevation change rates of both signs' – please clarify what you mean here.

546: clarify that this is the mass change of the firn column, not the entire ice sheet. Also, clarify "lower" and "higher" mean for changes, e.g. for West Antarctica, does "1.2 Gt/yr higher" mean that the mass change was -3.8 Gt/yr instead of -5 Gt/yr (I made those numbers up to illustrate; in this case higher refers to position on the number line) or higher mean that the mass change was -6.2 Gt/yr instead of 5 Gt/yr (in this case 'higher' means greater mass loss)?

594: Change wording: "forcing or forcing during" reads oddly.

---

## Author Response (AR2)

**Response to Reviewer 1:**
We would like to thank dr. Vincent Verjans again for thoroughly reviewing this study. Based on these comments, we have clarified the text and added new figures to the supplementary material. Responses to the comments of the reviewer are written in blue and citations of the manuscript in green.

Kind regards,

Sanne Veldhuijsen

**Minor comment 1)**
Section 6 still needs some work, and the last two paragraphs in particular. I have read this section several times, and I still do not understand how the statements presented can be derived from the results. I specify these statements here:

-(l516-517) "In non-spin-up experiments, the uncertainty is only 4 % of the total surface elevation change." Is this 4% value calculated at the end of the time series? Or averaged over the time series? Or something else?
There was a typo in this sentence, that explains the confusion, we have improved as: "The uncertainty of the non-spin-up experiments (Fig. S7b) is only 4 % of the total surface elevation change uncertainty (Fig. S7a)."

-(l517) "This uncertainty is smaller than the 40 % that can be derived from Table 3." I do not understand how this can be derived from Table 5 (note that the Table number is wrong in the text). Can the authors please clarify how they calculate the 40% value? Indeed this should be Table 5. See comment above.

-(l524-525) "It indicates that imposing an uncertainty in the experiments mostly has an impact on the high density, low-FAC part of the firn column." I understand that the uncertainty in FAC is lower than in dH/dt. However, I disagree that this necessarily implies that mostly high-density firn is affected by uncertainties here. For example, density could be higher at all vertical levels (due to higher temperatures) but compensated by a larger total firn column thickness (due to higher accumulation). This can result in similar FAC values (i.e., low FAC uncertainty) but with density values affected throughout the firn column. Therefore, the authors should support this statement quantitatively, and using extra figures in the supplements that show that low-density firn is unaffected by the sensitivity tests. When analyzing the results, we indeed saw that also the low-density part of the firn is impacted. Therefore, we have decided to remove this sentence. However, in very high-accumulation regions, a large part of the firn does not originate from the spin-up period, therefore here its main impact is logically on deeper, high-density firn. Therefore we have reformulated the sentence: "We attribute this to the fact that in high-accumulation regions, a large part of the firn layer does not originate from the spin-up period. Here, the main impact is on deeper high-density firn, meaning that the impact on a volume to mass conversion is limited compared to the large uncertainty in surface elevation change."

-(l525-526) "The spatial pattern in FAC change uncertainty for the spin-up experiments is different from that of surface elevation uncertainty (Fig. S6)" This is not clear to me when I compare Fig. 10b and Fig. S6. In contrast, I find the patterns generally similar. Thank you for noticing. They are indeed generally similar, especially when looking at the figures. However, as we discuss there is a decreasing uncertainty for high-accumulation regions (>600 mm/yr).
We propose to rewrite this:
 "The spatial pattern in FAC change uncertainty for the spin-up experiments is in general similar to the surface elevation change uncertainty, apart from a decreasing uncertainty in high-accumulation regions (>600 mm/yr)."

-(l527-528) "We attribute this to the fact that in high-accumulation regions, the imposed uncertainty has an effect on the entire firn layer much more quickly." Why is that? Can the authors clarify how they reach this conclusion? What we mean is that in high-accumulation regions a large part of the firn does not originate from the spin-up period. We propose to clarify this as follows: "We attribute this to the fact that in high-accumulation regions, a large part of the firn layer does not originate from the spin-up period."

-(l533-534) "It suggests that the ice-sheet-wide uncertainty due to uncertainty in the spin-up climate is lower than suggested in Figure S7c." Do the authors mean that their uncertainty estimates provided in Table 2 overestimate the actual uncertainty? In this case, this contradicts the references that are provided in Table 2.
Some locations experienced an increase in accumulation whereas others experienced a decrease in accumulation over the last centuries. This has opposite effects on the surface elevation change, and therefore the actual uncertainty averaged over the AIS is lower than expected. However, due too poor spatial coverage of accumulation records over the AIS, it is not feasible to perform separate experiments for all regions. In some regions, no significant trends have been observed, but on average we estimate an uncertainty of -5/+5%. Therefore, we rephrase as follows:
"Over the remainder of the AIS, some locations experienced an increase in accumulation whereas others experienced a decrease in accumulation over the last centuries (Thomas et al. 2017). This has opposite effects on the surface elevation change, which suggests that the surface elevation and FAC change uncertainties averaged over the AIS are likely lower than the results of the spin-up temperature and accumulation uncertainty experiments (Fig. S7c)."

**Specific line-by-line comments**
-Title I suggest: "Characteristics of the 1979-2020 Antarctic firn layer simulated with IMAU-FDM v1.2A" Thank you for this recommendation, we have adjusted the title accordingly.

-l2 Here and elsewhere in the manuscript, I suggest replacing "ice thickness" by "ice mass". Done.

-l6-7 "seasonal and decadal surface height variability is due to variations": replace is by are. Done, thank you for noticing.

-l7 Add: "firn mass, respectively". To clarify, we changed this to: "We found that 62 % of the seasonal and 67 % of the decadal surface height variability are due to variations in firn air content rather than firn mass." (there was also an error in the values mentioned in the abstract/conclusion). We also changed this sentence in the conclusion.

-l34-35 Replace: "in mass- and density-change components". Done

-l35 Correct: "an ice-dynamical". Thank you for pointing this out. Done

-l37 Add comma: "climate, enhanced" Done.

-l52 Replace "while" by "but". Done.

-l72 Replace: "characteristics". Done.

-Table 1 Include FDM v1.2G. We included FDM v1.2G in Table 1 of the revised MS.

-l108 Replace "depositing" by "deposition". Done.

-l113 Please specify: "to simulate the fresh snow density, here applied to the top-most model layer (typically xx cm thick)" OR "to simulate the fresh snow density, here applied to the top 0.5 m of the model domain", depending on the modeling procedure. The fresh snow density, is the density of het fresh snow that is added on top, and not of the top most layer or 0.5 m of the model domain. We propose to clarify as: "These fit coefficients are then used to simulate the fresh snow density, i.e. the density of the fresh snow that is added on top of the firn column."

-l114 Replace "include" by "perform". Done.

-l123 Replace "measure" by "proxy". Done.

-l124 Replace "assuming that the accumulation rate is constant" by "approximating the accumulation rate as constant". We agree, done.

-l128 Replace "by deformation, sublimation and diffusion" by "by deformation, recrystallization and molecular diffusion". Thank you, done.

-l141 Replace "include" by "perform". Done.

-l151 Is it only retained meltwater that is allowed to refreeze? I believe that meltwater reaching a subfreezing layer is allowed to refreeze even if it is not retained through the irreducible water content before. That is indeed true. We propose to change this into: "The meltwater refreezes when it reaches a layer with a temperature below the freezing point."

-l187 Replace "retuned cloud scheme and snow properties" by "retuned schemes for clouds and for snow properties". This was a typo and is now fixed as follows: "RACMO2.3p2 employs upper-air relaxation, has updated topography, and a retuned cloud scheme and modified snow properties."

-Section 2.5 Is it possible to describe better how the separation between calibration and evaluation cores was decided? This was a pragmatic decision, as some were added later when calibration and computational demanding simulations of the entire AIS were already performed. This is clarified in the MS.

-l214 I think that "method and citation" should be plural. This has been improved.

-l221 Add comma: "15 cm, depending". Done.

-l230 Typo: "is currently too short". Thank you. Done.

-l242 "excluding five sites that were later added for evaluation": this is not clear. Is it meant that five additional sites were later added for evaluation? See our response to Comment 1 of Reviewer 2.

-l247 Replace "a model sensitivity test, which includes" by "a range of model sensitivity tests, which include". Done.

-l248 Is the fresh snow density varied independently at each time step? No, it is adjusted with a constant value. To clarify this, we have rephrased this as: "The fresh snow density was uniformly varied with the RMSE from the evaluation with in situ measurements (+/-30 kg/m3)".

-l252-254 Similar question: are the temperature variations applied independently at each time step? Similar as above. We specify this as: "This spread varies spatially and is assumed to be constant through time."

-l257 Change: "isotope". Done.

-l258-259 A citation is needed for this statement. We agree and added Stenni et al. 2017 and Thomas et al. 2017.

-l259 Replace "about typically" by "approximately". Done.

-l268 Please specify "we also performed two tests". Done.

-l272 Replace "a steady-state firn layer" by "a firn layer in steady-state". Done.

-l272 Replace "this simulation" by "these simulations". Done.

-l274 Specify: "and/or temperature". Done.

-l275 Why "randomly"? Please clarify. Over the period 1979-2020 there are no remaining trends in surface elevation due to the assumption of a steady-state firn layer, but sub-periods can also have limited trends. In 2015-2020 we did find trends in surface elevation change for most locations, which enables us to test the sensitivity. We propose to change this into: "In the remainder of the sensitivity experiments, we therefore compare surface elevation change over the period 2015-2020, as most locations experienced clear trends in surface elevation over that period."

-l278 Replace "included" by "shown". Done.

-l287 Rephrase: "extent (bias = 10.3 kg m−3 ). It is the update in the atmospheric forcing that causes a poorer (…)". Thank you for this recommendation. We have changed this accordingly.

-l296 If this is correct, please specify: "is reduced by 18 kg m-3 on average". Done.

-l299-301 In my view, these two sentences can be removed. We have added these sentences following a comment from the Editor. Therefore, we decided to keep the first sentence. However, we removed the last sentence, as this is already mentioned.

-l302-303 How can do the MO values influence the rho0 parameterization? Is it not the other way around? The surface snow density, is the density of the top 0.5 m. Some densification has taken place, and is thus influenced by the MO fits.

-Table 3 The R2 values do not agree with the R2 values of Fig. 2. Thank you for noticing this. This has been corrected.

-Table 4 First, MO coefficients have no units, and thus the last column should be unitless. Second, RMSE should be replaced by RMSD (root mean squared deviation) because this quantifies the deviations with respect to the fitted function, and not errors. Finally, RMSD values should be given in all rows. This should indeed be unitless and RMSD, we have changed this. Unfortunately, we do not have the required data of FDM v1.1 to calculate the RMSD of its MO fits. We added the RMSD of FDM v1.2A for MO500, which is 0.13. Furthermore, the r2 values for both FDM v1.1p1 were taken from Ligtenberg et al. 2011, hence using a different and smaller observational dataset, and not the observations dataset used for v1.2a. As this is not yet clear in the manuscript, we added the following sentence to the caption of Table 4: "… et al (2011). Please note that R2 for v1.1a applies to the calibration dataset used in that study."

-l316 The comparison with FDM v1.1p1 and FDM v1.1p2 is unfair. These models were not calibrated with the data of this study, whereas FDM v1.2A was calibrated with (almost) all the data used for evaluation. This must be specified. Therefore we also compare to FDM v1.2A-log. In order to further mitigate this unequal comparison, we have now derived these percentages using only the FAC observations that were used to calibrate FDM v1.1p1 and FDM v1.1p2, and wet cores that were not used for evaluation at all. We updated the manuscript accordingly: "When we only include the seven observations that were used to calibrate FDM v1.1 and, two wet firn cores, the RMSE of FAC simulated by FDM v1.2a is 2.79 m, which is 19.5 % and 9.0 % lower compared to FDM v1.1p1 and FDM v.1.1p2 for these observations."

-l324 Correct: "asymptotically". See comment below.

-l324 To avoid any confusion, please specify: "asymptotically towards zero, but less rapidly than FDM v1.2A-log". We clarify: "where the MO fit curve decreases less rapidly than FDM v1.2A-log."

-l330 Replace "68" by "0.68". Done.

-l348-349 Please quantify: " The patterns vary spatially across climatic regions with temperature as a primary driver and accumulation and surface melt as secondary drivers." As this is hard to quantify, we decided to not differentiate between primary/secondary drivers. This has been formulated as: "The patterns vary spatially across climatic regions with temperature, accumulation and surface melt as drivers."

-l353 Please specify: "in younger, thus less densified". Thank you for this recommendation, we have changed this accordingly.

-l360 "If we compare this to observations": which observations give firn age? We have specified this: "When comparing this to firn age inferred from δ15N measurements for 15 locations (Breant et al. 2017), we find ..."

-l360-361 "On average, the RMSE is 25 %": is this really the RMSE (i.e., the mean across all the errors) or rather the average ratio between the individual absolute errors and their corresponding observed value? This is indeed the ratio of the individual absolute errors. We have specified this : "On average, the absolute errors are 25 % of the observed values."

-l361 "In comparison, RMSE of z830 is 15 % of the observed values": same question as comment just above. See comment above. We have specified this: "In comparison, the absolute errors of z830 are 15 % of the observed values on average, and R2 = 0.73."

-l366 Replace "would be" by "was". We think this should be "would be".

-l371 Specify: "Community Firn Model". We have specified this.

-l375 Replace both instances of "new" by "updated". We agree. Done.

-l378 I think that "FDMv1.2A" should be replaced by "FDMv1.1". That is indeed true. Done.

-l383 Correct: "contribute". Done.

-l387 Correct: "values". Thank you, done.

-Section 4.3 Is it possible to provide maps of mean accumulation, melt, and sublimation in the Supplementary Information? We agree that this is a good idea. We have provided maps of mean accumulation, melt and sublimation in Figure S9. We refer to this in Section 4.3 as: "Figure 6c shows that the seasonal firn thickness amplitude can be up to 3 m in the western Antarctic Peninsula, caused by a rare combination of high snowfall and strong melt. Maps of mean annual precipitation, melt and sublimation are shown in Figure S9."

-l415 " as the annual average accumulation is low": I do not understand why this causes low seasonal amplitude. For a similar seasonality in climatic conditions, the amplitude is lower when accumulation is lower. Low accumulation generally also means low absolute seasonality in accumulation, as zero accumulation remains a hard lower boundary.

-l421 Replace "its components" by "the components of the latter". In response to a comment by Reviewer 3, we propose to change this: "Figure 7 shows time series of integrated FAC, the cumulative surface temperature anomalies (against the long-term mean) and the vertical firn surface velocity anomalies (against the long-term mean) for the entire AIS (Fig. 7a) and above/below 2,000 m a.s.l. (Fig. 7b,c). In addition, the surface velocity anomalies are broken down into their components (Eq. 6)."

-l424 Replace "long time scale" by "slower response timescale". Done.

-l426 Typo: vfc is all in subscript. Thank you for noticing, we have corrected this.

-l430-432 Note that this difference is also explained by the high sensitivity of firn compaction to temperature variability for firn at low temperatures. Thank you for this recommendation, we have added this explanation: "This difference can partly be explained by the larger interannual temperature variability above 2,000 m a.s.l (sd in annual means = 0.78 K compared to 0.48 K), the high sensitivity of firn compaction to temperature variability for firn at low temperatures, and the absence of melt."

-l432 Correct: "variabilities". Done.

-l435 Specify: "total surface elevation change variability". Thank you, done.

-l451 Please remove "which compares well with the FDM seasonal amplitudes", and replace it by a new sentence: "This suggests that our results of AIS-wide seasonal amplitude lie within the range of observational uncertainty." Thank you, we have changed this accordingly.

-l480 Remove "relatively". Done.

-l482: Typo: "reduced". Done.

-l484-485 Break the sentence: "depth (Schröder et al., 2019). On the other hand,". We have changed this.

-l487 vs. l490 Make sure to be consistent in using brackets or not around the location numbers. Done.

-l495 If this is correct, specify: "are on average 41% lower". This is correct, so we have specified this.

-l497-498 Replace "is needed" by "we used". We agree and have changed this.

-l498 Refer to Fig. S3. We assume the reviewer means Fig. S1. We include this reference in the revised MS.

 -l505 Typo: "a surface elevation". Corrected.

-l506 Typo: "of of". Corrected.

-l507 I do not understand "0.7/0.8". This should be "0.7". This has been corrected.

-Table 5 Specify periods in the caption: 2015-2020 for dH/dt, and averaged over 1979-2020 for FAC I think. The surface elevation change uncertainty for the spin-up experiments is 1979-2020, therefore we decided to only specify this as: "The FAC uncertainty for all experiments and surface elevation change uncertainty for the spin-up experiments are calculated over 1979-2020, and the surface elevation change uncertainty for the MO fits, $\rho_s$, $T_s$ and b experiments are calculated over 2015-2020."

-Figure 10 Specify periods in the caption: 2015-2020 for dH/dt, and averaged over 1979-2020 for FAC I think. See our response to the comment above.

-l517 Replace "Table 3" by "Table 5". Done.

-l518 Replace "amplifying" by "compensating". See our response to Minor comment 1.

-l519 Replace "Table 3" by "Table 5". Done.

-l522 Replace this sentence by "We expand ice-sheet-wide averaged estimates of uncertainty into time series, show in Figure S7c."  In response to Rewiever 3, we propose to clarify this, by rephrasing as:
"This enables us to expand ice-sheet-wide estimates of uncertainty of the spin-up experiments into time series (Fig. S7c), by adding this known constant uncertainty for each timestep."

-l529-530 Replace here: "The finding that imposing an uncertainty in spin-up climate has low impact on FAC uncertainty". (see my Minor comment). See our response to minor comment.

-l537 Replace "minus the spin-up sensitivity test" by "minus the transient run spun-up with the sensitivity test". We reformulated as: "We found that the absolute residual trend of altimetry minus the spin-up sensitivity test was reduced by 25 %".

-l539 "we assume the underlying ice to have responded to the changing climatic conditions": does this mean that advection if ice at the lower boundary has changed? And is this equivalent to saying that v_ice cannot be assumed constant? Yes, indeed. We have added this explanation: "This suggests that the uncertainty for those locations (age < 42 years) is overestimated, as $v_{ice}$ can not be assumed constant."

-Section 7 I would appreciate adding a couple of sentences about larger uncertainties in melt-affected areas. Can the authors also provide the quantitative results of the evaluation at the 10 wet cores? We found at the 11 wet firn core sites that the RMSE of $z_{550}$ amounts to 3.17 m, of $z_{830}$ to 8.87 m and of FAC to 4.40 m. We mention this in Section 3.2 Dry snow densification rate, as follows: "For the 11 wet firn cores we found that the RMSE of modelled $z_{550}$, $z_{830}$, FAC amounts to 3.17 m, 14.34  m and 4.40 m, respectively, and thus larger than the RMSEs of the entire firn core dataset (2.33, 8.87 and 2.63 m, respectively)."
In Section 7 we discuss this as a separate final paragraph: "The densification scheme used in FDM v1.2A (eq. 3) is developed for dry snow densification. As it is used for both wet and dry locations, we assume that the densification rate of dry firn is equal to that of wet firn. Since the presence of liquid water impacts the evolution of grain size and shape, this may also impact the densification rate of firn, however due to a lack of physical understanding and available measurements, this has not been included in FDM v1.2A. The RMSE of the modelled $z_{550}$, $z_{830}$ and FAC for the wet firn cores are respectively, 3.17 m, 14.34 m and 4.40 m, and thus larger than the RMSEs of the entire firn core dataset (2.33, 8.87 and 2.63 m, respectively). Uncertainties in the melt forcing (Carter et al. 2022) impact the simulated density profile, which may contribute to the lower agreement and hampers this comparison."

-l543 Specify: "has been used over the AIS". Done.

-l544 Correct format of citation. This has been corrected.

-l546 Specify: "the mass change associated with surface processes". This is about total mass change. See our response to comment about l546 of Reviewer 3.

-l548 Correct: "updated climatic forcing, as is shown". Done.

-l553: Specify: "compared to the error range of the altimetry observations and model uncertainties". Done.

-l558 Replace "decadal" by "inter-annual". Done.

-l561 Remove comma after "factor". Done.

-l565 Replace: "not recommended in combination". Done.

-l577-578 See my comment on l316. We added an additional evaluation of only firn cores that were used for FDM v1.1 Because FDM is a semi-empirical model, additional fit data will improve the model by definition, which is also shown by the similar value of 21% RMSE lowering from the entire firn core dataset to 19.5% RMSE lowering of the FDM v1.1 firn core dataset.

-l582 Phrasing: "as a primary, and accumulation and surface melt as secondary drivers." Not relevant in the revised MS.

-l585 Replace "As variations in FAC and firn thickness have a similar phase, 63 to 68 %" by "Variations in FAC and firn thickness have a similar phase, and 63 to 68 %". Done.

 -l591 Replace "which violates" by "violating". Done.

-Caption of Figure S4 (h) should be for the accumulation spin-up experiments. Thank you, this has been corrected.

-Caption of Figure S5 (h) should be for the accumulation spin-up experiments. Done.

-Caption of Figure S7 Specify: "the estimated ice-sheet-wide averaged surface elevation change". Done.

**Response to Reviewer 2:**

We would like to thank dr. Baptiste Vandecrux again for reviewing and editing this study. Based on your comments, we have clarified the text and Figure 2a. Responses to the comments of the reviewer are written in blue and citations of the manuscript in green.

Kind regards,

Sanne Veldhuijsen

**Comments:**

line 241-242: "(the firn firn cores used ... )". I recommend removing this sentence as it is ambiguous: did you mean "the firn cores used in the sensitivity analysis"? If yes, then that already makes sense and the sentence between brackets can be removed. If it meant "the 103 cores used to fit/evaluate MOs and z550/850 are located in 100 different model cells and you use only 95 of these in the sensitivity analysis" then the original sentence is very elusive and maybe not so important. We propose to clarify this as follows: "These locations include 95 locations from firn core sites (the firn cores initially used for MO calibration and z550 and z830 evaluation are located at 95 unique grid points)."

l. 270 "except for the MO fits" maybe spell out to "except for the experiment where we perturbate the coefficients of the MO fits". Also consider changing "fits" to "function" or something more specific. Thank you, we have spelled this out according to your feedback. We have not changed fit into function.

l.275 "randomly selected period 2015-2020", why was that period chosen randomly? you could have used the altimetry period for which you have a separate estimation for the surface height trend? Maybe I missed the explanation, but "randomly" still raises some questions. See our response to comment l275 of Reviewer 1.

Figure 2. Thanks for updating it and adding more FDM FS-L and FDM v1.1p1 points. It allows the readers to see, among other details, the negative bias of FDM FS-L. Now it has more overlapping dots. You can consider using different symbols for different runs and using semi-transparency to see overlapping points. This is very optional.
Thank you, we made the inside of the circles transparent to improve the readability.

l.360 "to observations" give number and reference. Thank you for this recommendation, we have specified this as: "When comparing this to firn age inferred from δ15N measurements for 15 locations (Bréant et al., 2017), we find …"

L.482 "recued" reduced? This has been corrected.

L.514-523 This paragraph is very hard to read. Please add short explanation (or reminder) of the different quantities you are discussing:
"The non spin up experiments" please give their name as presented in table 5. We specify the experiments now as follows: "for the non-spin-up experiments, i.e. the MOfits, fresh snow density, temperature and accumulation experiments."

"is only 4%" please consider spelling out where that number comes from, like: "for these experiments, the uncertainty in dhdt (x-y mmyr-1, table 5) , are only 4% of the total elevation change (z mmyr-1 over period Y1-Y2)". There was a typo in this sentence, that explains the confusion, we have improved this: "The uncertainty of the non-spin-up experiments (Fig. S7b) is only 4 % of the total surface elevation change uncertainty (Figs. S7a)."

"40% that can be derived from table 3" give quick reminder of how these numbers were calculated. See comments above.

"amplifying effect" what is amplified? I understood from the previous sentence that the sensitivity analysis presented in table 5 ends up to lower uncertainty than first assumed from table 3. With amplifying we mean that an uncertainty (e.g. lower fresh snow density) amplifies the rate of surface height lowering as well as a surface height increasing. To clarify we propose to change this into: "This is because an imposed uncertainty has opposite effects across the AIS and in time. For example, lowering the fresh snow density increases the rate of a surface elevation increase/decrease."

"lowering the fresh snow density increases surface elevation change rates of both signs" I am not sure how to interpret this sentence:
i) wasn't the experiment about adding either subtracting 30 kgm-3 (l.249) so why only talking about the decreased case? Does the increased fresh snow density also lead to "surface height changes of both sings". I know that it is an example but I don't understand what it illustrates.
ii) I am also unsure what "of both signs" means: do you mean that eventually the changes caused by perturbing the fresh snow density cancel out and that is why the new uncertainty is "only 4%" of the overall trend? Please write out your reasoning so that there is nothing left to guess. See comment above.

L. 522. Does "this" refer to the last preceding sentence or to the whole sensitivity analysis? If it is the former, please spell out why or how. This indeed refers to the last preceding sentence. We propose to clarify this, by rephrasing this as: "This enables us to expand ice-sheet-wide estimates of uncertainty of the spin-up experiments into time series (Fig. S7c), by adding this known constant uncertainty for each timestep."

L.534 "lower than suggested by figure S7c" please spell out what is in figure S7c (the uncertainty of surface height and FAC due to alternative temperature and accumulation during spin-up), so that the sentence can be understood without going back and forth between the supplementary material and the main text. To clarify, we propose to change this into: ", which suggests that the surface elevation and FAC change uncertainties averaged over the AIS are likely lower than the results of the spin-up temperature and accumulation uncertainty experiments (Fig. S7c).

**Response to Reviewer 3:**
We would like to thank dr. Max Stevens again for reviewing and editing this study. Based on your comments, we have further clarified the text, by amongst others, specifying how we use the term AIS. Responses to the comments of the reviewer are written in blue and citations of the manuscript in green.

Kind regards,

Sanne Veldhuijsen

**Main comment:**
The largest issue that must be addressed is the definition of the "AIS" (Apologies to the authors that I did not catch this in the first review). I think the authors are using the term 'AIS' as shorthand for the entire continent, including both the ice sheets (WAIS and EAIS), the ice shelves, and distinct glaciers and ice fields on the Antarctic Peninsula. Strictly, the term AIS refers only to the ice sheets, and the authors define AIS as the Antarctic Ice Sheet in the first sentence of the paper. But, then throughout the paper it seems that the analyses are including ice shelves and the peninsula. This leads to some confusion in the paper. For example, the authors mention on line 344 that 99% of the AIS has a firn layer. Is this figure strictly for the ice sheets, or is the peninsula, which contains distinct glaciers, included in the 99%? How about the ice shelves? I think it would be good to specifically define the area(s) included in the study. For example, "We use the term AIS broadly to include the East and West Antarctic Ice sheets, as well as distinct glaciers and ice fields on the continent and the ice shelves". Thank you for noticing this. We use the term AIS to broadly include the East Antarctic, West Antarctic and Antarctic Peninsula ice sheets, as well as disconnected glaciers and ice caps on the continent and ice shelves, and we agree that it is important to specify this. We propose to add this definition at the end of the Section 2.1 IMAU-FDM as follows: "The model is applied to the AIS, and we use this term to broadly include the East Antarctic, West Antarctic and Antarctic Peninsula ice sheets, as well as the ice shelves and disconnected glaciers and ice caps on the continent."

**Line by line comments:**
13: is losing → has been losing. Done.

37: comma after climate. Thank you, done.

63: change 'this study' to 'that study' to clarify that you are not talking about your own paper. We agree, this has been changed.

130: consider specifying 'requires dimensionless correction terms'. We have specified this accordingly.

Section 2.3: Consider mentioning in this section that v_bedrock also contributes to ice-sheet elevation change but is not included in IMAU-FDM. This section only describes the impact of firn and SMB processes; therefore, we decided not to include dynamical change of the ice and bedrock in the bedrock. However, we do discuss the bedrock contribution in Section 1 (Introduction) and Section 2.5 (Observational data). To improve clarity, we repeat this in this section: "The contributions of ice-dynamical imbalance and bedrock motion to changes in the surface elevation are not included in the current model."

305: clarify 'without MO corrections' – does that mean that you just used the Arthern et al. (2010) equation, i.e. Equation 3 in your paper? Or that you ran the model using the updated RACMO forcing but the MO values from Ligtenberg et al. 2011? (Table 4 does not make it clearer to me). Yes indeed, we just use the Arthern et al. (2010) equation. To clarify this, we have changed this into: "To calibrate the dry snow densification rate we first performed a simulation of FDM v1.2A without MO corrections, thus only using Eq. 3."

315/Table 4: I got a bit confused here because you refer to FDM v1.1p2, which is not listed in Table 4. From Table 1, I gather that v1.1p2 is the model using RAMCO2.3p2 and the Ligtenberg MO values. To clarify this section, it may be as simple as altering the first and third lines of Table 4 to read, "MO550FDM v1.1p1/v1.1p2" and adding a second R2 value. Then, add a note in the caption explaining the notation and reminding the reader that v1.1p2 uses the RACMO2.3.p2. Alternatively, you could add 2 more lines to the table. As a reader, I was alternating between the text and Table 4 while reading this section, and so I was a bit thrown off when I had to go searching to remind me what v1.1p2 was. I realize there is a risk of redundancy (ie., you did define v1.1p2 earlier), but given the number of model variations that are included in your study I as a reader find it helpful to have reminders throughout the text of what the variations are. We do not have the fit statistics of FDM v1.1p2, since we use the fit coefficients from the Ligtenberg et al. (2011) calibration. Also, there is not much added value, since FDM v1.1p2 uses fit coefficients obtained using a different forcing. Therefore, in Table 4 we only show the statistics of the versions in which the forcing matches that used to obtain the calibration parameters.

To clarify the text, we propose to change this into: "The update in atmospheric forcing without updated MO calibration does not result in an improvement in FAC (shown by the similar RMSE of FDM v1.1p1 and FDM v1.1p2, which is FDM v1.1 forced with the updated forcing RACMO2.3p2)."

325: Similar to my last 2 comments – I am again confused by your naming scheme. Which is the 'reference FDM v1.2A'? That is not included in Table 1 or Table 4. If one of the runs you are referring to is the -log run, refer to it as you have defined it previously. With reference FDM v1.2A, we mean the ones not from the 10-fold cross evaluation simulations. To avoid confusing, we removed "the reference".

329: Specify: "if we only include Antarctic cores from the…" Thank you, done.

329: change to, "we find similar model fits to the data for the logarithmic model (R2 =0.67) and the power model (R2 =0.68)."
Thank you for this recommendation, we have changed this accordingly.

330: Change to 'Another difference between FDM v1.2G and v1.2A is that in …' Done.

337: Coming from the previous section that discussed many model variations, it would be useful here to specify, "Using the improved firn model FDM v.1.2A, we simulate…" This will make it very clear to the reader what to refer to in Tables 1 and 4. Done.

344: specify: "In line with observations, FDM v1.2A predicts that nearly all…" Done.

352: remove word 'especially' Done.

361-362: The sentence about the large spread of timescales in the firn is vague – either add more detail of how these add to our understanding or remove this sentence. I think the spread in firn ages in Antarctica is reasonably well documented in the ice-core literature. I am not sure if you are implying that your findings help with process understanding (e.g., processes operating on different timescales) or implying that your statistics (25% vs 15% RMSE for firn age and depth) can provide new insights – if so, what are those? We decided to remove this sentence.

371: Define CFM; please cite We have specified: The Community Firn Model.

373: Related to comment above – make sure each instance of referring to AIS is consistent or explained. Here – did the values in the previous paragraph not include ice shelves? Based on your previous comment, we decided to specify how we use the term AIS in section 2.1. In line with how we define the term there, we change this : "the entire AIS (including the ice shelves)" in line 373 into: "the AIS" to avoid confusion.

376: spatial → spatially Done, thank you.

420: This sentence needs punctuation and/or to be broken into multiple sentences to clarify. Also consider referring to figure panels in the sentence. Something like: Figure 7 shows time series of the entire AIS (7a) and above/below 2000 m (7b/7c). Each panel includes integrated FAC and the cumulative surface temperature and surface velocity anomalies. The surface velocity anomalies are broken down into their components. To clarify, we propose to change this into: "Figure 7 shows time series of integrated FAC, the cumulative surface temperature anomalies (against the long-term mean) and the vertical firn surface velocity anomalies (against the long-term mean) for the entire AIS (Fig. 7a) and above/below 2,000 m a.s.l. (Fig. 7b,c). In addition, the surface velocity anomalies are broken down into their components (Eq. 6)."

487: change to plural: "find substantial positive residual-altimetry trends at locations" Thank you, done.

494: updated MO fits – perhaps change language in this section to refer to the model versions, e.g. v1.2A, v1.1p2, etc. Then the reader can refer directly to tables to remember which one you are referring to. This is all FDM v1.2A, but for each sensitivity analysis, we also re-estimated the MO fit coefficients now that we changed fresh snow density, accumulation or melt. This is also explained in Section 2.6: "For all our experiments, except for the MO fits, we re-ran our model calibration procedure to get the optimal MO fits. We then ran the model with these updated MO fits and compared the outputs to the FDM v1.2A run." To clarify this in Section 6, we propose to change this into: "For the sample locations, the sensitivities with updated MO fits are lower than with the FDMv1.2A MO fits (i.e. without updated MO fits), especially the FAC uncertainties, which are 41 % lower."

518: 'increases the surface elevation change rates of both signs' – please clarify what you mean here. To clarify, we change this into: "For example, lowering the fresh snow density increases the rate of surface elevation increase/decrease."

546: clarify that this is the mass change of the firn column, not the entire ice sheet. Also, clarify "lower" and "higher" mean for changes, e.g. for West Antarctica, does "1.2 Gt/yr higher" mean that the mass change was -3.8 Gt/yr instead of -5 Gt/yr (I made those numbers up to illustrate; in this case higher refers to position on the number line) or higher mean that the mass change was -6.2 Gt/yr instead of 5 Gt/yr (in this case 'higher' means greater mass loss)? These numbers are about the entire ice sheet, which is more relevant for altimetry users. Also, the difference in firn mass change is opposite to the change in

total mass loss, and would therefore be confusing to mention. As you can see in Figure 8b,c for the EAIS, the firn height has increased in FDM v1.2A compared to FDM v1.1. Thus less increase in surface elevation change is caused by ice, which has a higher density.

We have specified lower and higher as follows: "When FDM v1.2A is used instead of FDM v1.1p1, the mass change over the period 2003-2015 for East Antarctica is 7.3 Gt/yr lower (187.7 Gt/yr), and for West Antarctica this is 1.2 Gt/yr higher (-243.8 Gt/yr)."

594: Change wording: "forcing or forcing during" reads oddly. We propose to clarify this as follows: "In general, uncertainties in the model formulation, climate forcing or climate forcing during the spin-up cause rather small changes in simulated FAC (<5.2 %)."